# Interleukin-11 causes alveolar type 2 cell dysfunction and prevents alveolar regeneration

Benjamin Ng [1,2] ✉, Kevin Y. Huang [2,5], Chee Jian Pua [1,5], Sivakumar Viswanathan [2], Wei-Wen Lim [1,2], Fathima F. Kuthubudeen [2], Yu-Ning Liu [2], An An Hii [1], Benjamin L. George [2], Anissa A. Widjaja [2], Enrico Petretto [2,3] & Stuart A. Cook [1,2,4] ✉

In lung disease, persistence of KRT8-expressing aberrant basaloid cells in the alveolar epithelium is associated with impaired tissue regeneration and pathological tissue remodeling. We analyzed single cell RNA sequencing datasets of human interstitial lung disease and found the profibrotic Interleukin-11 (IL11) cytokine to be highly and specifically expressed in aberrant KRT8+ basaloid cells. IL11 is similarly expressed by KRT8+ alveolar epithelial cells lining fibrotic lesions in a mouse model of interstitial lung disease. Stimulation of alveolar epithelial cells with IL11 causes epithelial-to-mesenchymal transition and promotes a KRT8-high state, which stalls the beneficial differentiation of alveolar type 2 (AT2)-to-AT1 cells. Inhibition of IL11-signaling in AT2 cells in vivo prevents the accumulation of KRT8+ cells, enhances AT1 cell differentiation and blocks fibrogenesis, which is replicated by anti-IL11 therapy. These data show that IL11 inhibits reparative AT2-to-AT1 differentiation in the damaged lung to limit endogenous alveolar regeneration, resulting in fibrotic lung disease.

The alveolar epithelium plays a pivotal role in lung homeostasis and protects the lung from inhaled environmental insults and pathogenic infections. In the alveolus, alveolar type 2 cells (AT2 cells) become activated after injury and proliferate and trans-differentiate into alveolar type-1 cells (AT1 cells) to restore alveolar structure and lung function[1,2]. A number of human lung pathologies, including idiopathic pulmonary fibrosis (IPF), chronic obstructive pulmonary disease (COPD), and post-infective lung damage, are characterized by failure of homeostatic AT2-to-AT1 transitions[3,4].

Recent large-scale single-cell RNA sequencing (scRNA-seq) studies of human pulmonary fibrosis (PF) have identified transitional cells that exhibit a dysfunctional phenotype and have a reduced capacity to differentiate into AT1 cells[5–9]. These disease-associated transitional

AT2 cells, coined KRT5−/KRT17+ or aberrant basaloid cells, accumulate in the lungs of patients with IPF[6,7] and after severe SARS-CoV-2 infection[10–12]. An analogous population of transitional cells termed Krt8+ alveolar differentiation intermediate (KRT8 + ADI)/damage-associated transient progenitors (DATPS)/pre-alveolar type-1 transitional cell state (PATS) are similarly seen in the damaged alveolus in mouse models of lung injury[13–15].

In mice, transitional cells, herein referred to as Krt8+ transitional cells, can be derived from either AT2 cells or airway stem cells, and possess the capacity to differentiate into mature AT1 cells[13,16]. Importantly, Krt8+ transitional cells in the mouse exhibit transcriptional similarities to human disease-associated KRT5−/KRT17+ / aberrant basaloid cells, including signatures of epithelial-mesenchymal

[1]National Heart Research Institute Singapore, National Heart Center Singapore, Singapore, Singapore. [2]Cardiovascular and Metabolic Disorders Program, Duke-National University of Singapore Medical School, Singapore, Singapore. [3]Center for Computational Biology, Duke-National University of Singapore Medical School, Singapore, Singapore. [4]MRC-London Institute of Medical Sciences, Hammersmith Hospital Campus, London, United Kingdom. [5]These authors contributed equally: Kevin Y. Huang, Chee Jian Pua. ✉e-mail: benjamin.ng.w.m@nhcs.com.sg; stuart.cook@duke-nus.edu.sg

transition (EMT), p53, and cell senescence pathways and expression of KRT8 itself[13,15]. Krt8+ transitional cells are thought to contribute to fibrosis via the expression of profibrotic and proinflammatory mediators. Recent studies have shown that elevated TGFβ signaling in AT2 cells and inositol-requiring transmembrane kinase/endoribonuclease 1α (IRE1α) activity in DATPS maintain the Krt8+ cell state following lung injury in mice[17–19]. Similarly, the persistence of senescent AT2 cells promotes progressive pulmonary fibrosis[20]. However, it remains unclear whether other molecular pathways can contribute to the emergence and abnormal maintenance of alveolar transitional cells in this aberrant state.

Interleukin-11 (IL11), a member of the IL-6 family of cytokines, is upregulated in the airways following viral infections and has been associated with a range of respiratory disorders[21]. We previously reported that IL11 was increased in the lungs and fibroblasts of patients with IPF, and its expression correlates with disease severity[22]. A contemporaneous study found that *IL11* was expressed in a range of cell types in fibrotic lungs of patients with Hermansky–Pudlak syndrome (HPS) and also in *SFTPC*+ cells in IPF[23]. More recent pharmacologic studies using siRNA have further confirmed the role of IL11 in lung fibrosis[24].

In the current study, we leveraged single-cell RNA sequencing (scRNA-seq) datasets from patients with fibrotic lung disease and analyzed IL11-lineage-labeled cells in a mouse model to delineate the different lung cell types expressing IL11 in the disease. We examined whether IL11 signaling plays a role in alveolar regeneration via its specific activity in AT2 cells using conditional *Il11ra1* deletion in AT2 cells and lineage tracing in mice that were subjected to bleomycin lung injury and also in studies of primary alveolar epithelial cells and AT2 cells in vitro. We also tested whether a neutralizing anti-IL11 antibody administered to mice with lung injury could promote alveolar regeneration by enhancing AT2-to-AT1 differentiation.

In this study, we show that IL11 is uniquely expressed by aberrant basaloid and KRT5-/KRT17+ cells in human lung fibrosis and by Krt8+ transitional cells in the fibrotic lungs of mice after bleomycin injury. In alveolar epithelial and AT2 cell cultures, IL11 stimulation promotes the expression of ECM proteins and a KRT8+ state that stalls AT2-to-AT1 differentiation. In the bleomycin model of lung fibrosis, the conditional deletion of *Il11ra1* in AT2 cells prevents the accumulation of profibrotic Krt8+ transitional cells, enhances alveolar epithelial regeneration, and protects against fibrosis. We further show that therapeutic administration of anti-IL11 antibodies in the bleomycin model similarly prevents the accumulation of profibrotic Krt8+ transitional cells and enhances regeneration of the injured lung epithelium. These data identify IL11 signaling in AT2 cells for the potentiation of pathological phenotypes in aberrant transitional epithelial cells in the injured lung and reveal that anti-IL11 may have the potential to enhance alveolar epithelial repair and promote lung regeneration in severe lung diseases.

## Results

### *IL11* is expressed by KRT5⁻/KRT17⁺ cells in human PF

To characterize IL11 and IL11RA expressing cells in human PF, we re-analyzed large-scale scRNA-seq data of lung cells from patients with PF from two independent studies by Habermann et al. and Adams et al. (GSE135893 and GSE136831, respectively)[6,7]. Our analysis showed that, in health, *IL11* was expressed at very low levels in the lung, and its expression was barely detected across most lung cell types (Supplementary Figs. 1, 2). In contrast, in PF, *IL11* was elevated in mesenchymal and epithelial cell populations and rarely detected in immune and endothelial cells (Supplementary Fig. 1). Within mesenchymal cells, *IL11* was most elevated in PLIN2+ lipofibroblasts and disease-specific HAS1^high fibroblasts (Supplementary Figs. 1, 2), which supports our previous findings[22,25] and further associates *IL11* with pathological fibroblast activity in PF. *IL11RA* (which encodes for IL11 receptor

subunit alpha), was broadly expressed and more highly enriched in fibroblast and alveolar epithelial cell populations in the human lung (Supplementary Fig. 2).

Amongst the various epithelial cell types identified in the two datasets, we observed particular enrichment of *IL11* expression in disease-specific KRT5⁻/KRT17⁺ ($P = 2.0 \times 10^{-33}$) and aberrant basaloid cells ($P = 1.2 \times 10^{-25}$) but limited *IL11* expression in basal, ciliated, MUC5B⁺, SCGB3A2⁺, AT2, transitional AT2 or AT1 epithelial cells (Fig. 1a–d and Supplementary Fig. 3). In contrast, *IL6*, which was recently implicated in airway epithelial dysfunction in fibrotic lung diseases[26], was broadly expressed in AT2, Mesothelial, MUC5AC+ High, MUC5B+ and Goblet epithelial cell types (Supplementary Fig. 4 and Supplementary Data 1) but seen rarely in transitional cells in both control and PF lungs.

Since KRT5⁻/KRT17⁺/aberrant basaloid cells may arise from defective AT2-to-AT1 differentiation, we performed trajectory and pseudotime analysis on transitional AT2 cells, KRT5⁻/KRT17⁺/aberrant basaloid and AT1 cells on combined Habermann et al. and Adams et al. datasets. To do this, we first confirmed that the transcriptional profiles between aberrant basaloid and transitional AT2 and KRT5⁻/KRT17⁺ cells were highly similar (Supplementary Fig. 5a). The aberrant cells in Adams et al. dataset were then assigned using the classification from the Habermann et al. dataset (i.e., transitional AT2 or KRT5⁻/KRT17⁺) by Seurat's FindTransfer Algorithm (see Methods) to obtain a consistent nomenclature across these two datasets. Our trajectory analyses showed two distinct differentiation paths for transitional AT2 cells in PF samples: (1) transitional AT2 to AT1 trajectory and (2) a trajectory from transitional AT2 to KRT5⁻/KRT17⁺ cells; with *IL11* expressed only by KRT5⁻/KRT17⁺ cells (Fig. 1e and Supplementary Fig. 5b). Pseudotime analysis revealed that *IL11* was specifically upregulated along the differentiation trajectory towards KRT5⁻/KRT17⁺ cells but not towards AT1 cells (Fig. 1f and Supplementary Fig. 5c).

To delineate a transcriptional program co-expressed with *IL11* along the KRT5⁻/KRT17⁺ cell trajectory, we performed co-expression analysis to the trajectory using cells assigned to the combined Habermann et al., and Adams et al. datasets and found that the *IL11* co-expression module was enriched for genes involved in epithelial-to-mesenchymal transition (EMT) (such as *COL1A1*, *SERPINE1*, *COL6A1*, *PTHLH*, *GLIPR1*, and *TGFBI*), TNFa via NFκB signaling, IL-1/ STAT5 signaling and p53 pathway (Fig. 1g, h and Supplementary Data 2, 3). Furthermore, the association between *IL11* and the *IL11* co-expression module was highly specific to disease (Fig. 1i and Supplementary Fig. 5d), suggesting a unique role of IL11 in dysfunctional alveolar epithelial cells in PF.

### IL11 is expressed by alveolar KRT8⁺ cells after lung injury

To further characterize IL11-expressing cells in the injured lung, we used an *IL11^EGFP* reporter mouse[27]. We performed single-dose oropharyngeal injections of bleomycin (BLM) (Fig. 2a), a drug that causes lung epithelial damage and fibrosis, and performed preliminary characterization of lung cells 10 days post-injury by flow cytometry (Supplementary Fig. 6). Using antibodies against a range of lung cell type markers: CD31 (endothelial cells), CD45 (hematopoietic cells), CD326/ EpCAM (epithelial cells), our analysis revealed that IL11^EGFP+ cells were rarely observed in the uninjured lung. However, following BLM injury, we found elevated proportions of IL11^EGFP+ cells in hematopoietic (CD45⁺ CD31⁻; $P = 0.0136$), epithelial (CD45⁻ CD31⁻ EpCAM⁺; $P = 0.0002$), and stromal cell populations (CD45⁻ CD31⁻ EpCAM⁻; $P = 0.0200$) (Supplementary Fig. 6). IL11^EGFP was not detected in endothelial cells (CD45⁻ CD31⁺) in both injured and uninjured lungs (Supplementary Fig. 6).

Since the low detection/abundance of IL11^EGFP+ cells precludes further FACS-based analysis, we next focused on immunohistochemistry to determine the identities of IL11-expressing cells in the injured lung. To do this, we assessed the lungs of *IL11^EGFP* reporter mice at 7 or

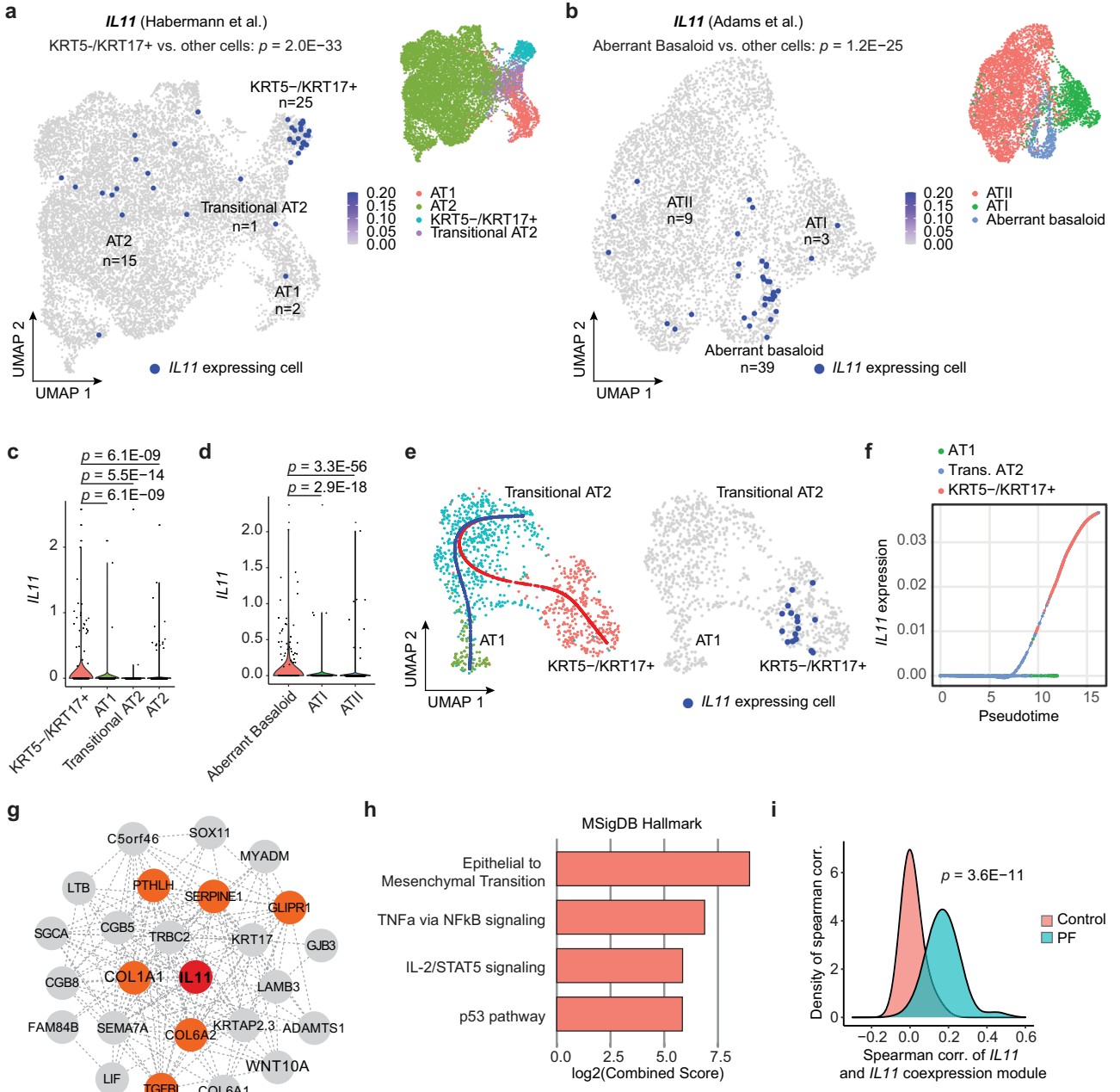

**Fig. 1 | *IL11* is specifically expressed by aberrant KRT5⁻/KRT17⁺ epithelial cells in human pulmonary fibrosis.** Uniform manifold approximation and projection (UMAP) visualization of *IL11* expressing single cells (colored in dark blue) in various alveolar epithelial cell populations (colored by cell types) in scRNA-seq data from Control and pulmonary fibrosis (PF) samples in **a** Habermann et al. (GSE135893) and **b** Adams et al. (GSE136831) datasets. The number of *IL11*-expressing cells in each cell cluster is indicated. *P* values determined by one-tailed hypergeometric test for enrichment in KRT5⁻/KRT17⁺ or aberrant basaloid versus other cell types. Violin plot indicating the expression of *IL11* in various alveolar epithelial cell populations in **c** Habermann et al. dataset and **d** Adams et al. datasets. *P* values were determined by a two-tailed Mann–Whitney test between KRT5⁻/KRT17⁺ or aberrant basaloid versus other cell types. **e** UMAP visualization of *IL11* expressing single cells colored in dark blue (right panel) and colored dots indicate cell type clustering (left panel). The blue line indicates the differentiation trajectory of transitional AT2 to AT1 cells; the red line indicates the differentiation trajectory from transitional AT2 to KRT5⁻/KRT17⁺. Data were composed of cells from PF samples in the Habermann et al. dataset. **f** Expression of *IL11* in the pseudotime trajectory from transitional AT2 to KRT5⁻/KRT17⁺ versus from transitional AT2 to AT1 cells in the Habermann et al. dataset. **g** Network of genes in the *IL11* co-expression module in the transitional AT2 to KRT5⁻/KRT17⁺ cell trajectory in combined Habermann et al. and Adams et al. datasets. *IL11* is colored in red, and genes related to epithelial to mesenchymal transition are colored in orange. **h** Pathway enrichment of genes in the *IL11* co-expression module using MSigDB Hallmark database. **i** Density plot displaying the distribution of Spearman correlation between the gene expression of *IL11* and *IL11* co-expression module in Control (salmon color) and PF (turquoise color) transitional AT2 and KRT5⁻/KRT17⁺ cells in Habermann dataset. *P* value was determined by the two-tailed Kolmogorov–Smirnov test.

21 days post-BLM injury by staining for GFP and counterstained for SFTPC (AT2 marker), PDPN (AT1 marker), PDGFRA (pan-fibroblast marker) or CD45. Consistent with the flow cytometry analysis, IL11^EGFP+ cells were very rarely observed in the lungs of uninjured *IL11*^EGFP reporter mice. In contrast, in BLM-injured lungs, IL11^EGFP expression was notably upregulated in SFTPC⁺ AT2 cells (Fig. 2b and Supplementary Fig. 7a), PDGFRA⁺ fibroblasts and a subset of CD45⁺ hematopoietic cells (Supplementary Fig. 7b, c) within injured alveolar regions

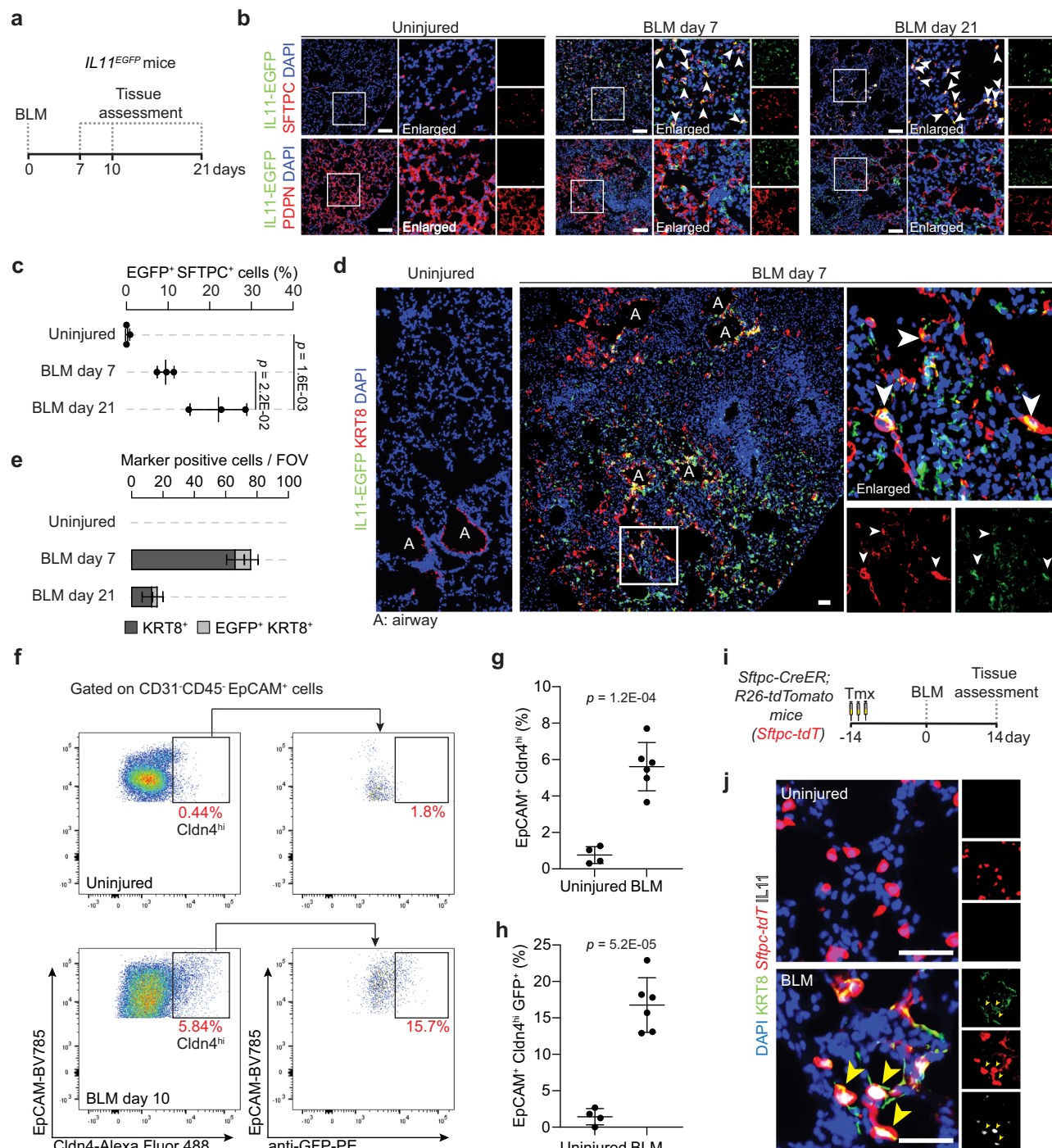

**Fig. 2 | IL11 is expressed by transitional alveolar epithelial cells after bleomycin-induced lung injury in mice. a** Schematic showing the induction of lung fibrosis via oropharyngeal injection of bleomycin (BLM) in *IL11^EGFP* reporter mice.
**b, c** Representative images and quantification of immunostaining for GFP and SFTPC or PDPN in injured regions of the lungs 7 or 21 days post-BLM challenge; *n* = 3 mice/group. **d** Representative images of immunostaining for GFP and KRT8 in the injured regions of the lung 7 days post-BLM injury. White arrowheads indicate marker-positive IL11^EGFP+ cells. **e** Quantification of alveolar KRT8+ and IL11^EGFP+ KRT8+ cells per field of view (FOV); *n* = 3 mice/group. **f**–**h** Flow cytometry analysis and

quantification of **g** EpCAM+ Cldn4^hi cells and **h** EpCAM+ Cldn4^hi GFP+ cells; *n* = 4 control and 6 injured mice/group. **i** Schematic showing the period of tamoxifen (Tmx) administration and induction of lung fibrosis in *Sftpc-CreER; R26-tdTomato* (*Sftpc-tdT*) mice. **j** Representative images of immunostaining for KRT8 and IL11 in the lungs of *Sftpc-tdT* mice post-BLM injury. DAPI for nuclei. Yellow arrowheads indicate IL11+ KRT8+ *tdT*+ cells. Scale bars: 50 μm. Data were representative of three independent experiments (**b, d, j**). Data (**c, d, g, h**) were mean ± s.d. *P* values were determined by one-way ANOVA with Tukey's multiple comparison test (**c**) or two-tailed Student's *t*-test (**g, h**).

that were marked by areas of dense consolidation of nuclei DAPI staining. IL11^EGFP was localized to numerous SFTPC+ cells adjacent to regions of tissue disruption with enlarged/elongated morphologies suggestive of transitional AT2 cells that have committed towards AT1 differentiation (Fig. 2b, c). Immunostaining for PDPN revealed that

IL11^EGFP expression was very rarely detected in mature AT1 cells in injured or uninjured lungs (Fig. 2b and Supplementary Fig. 7d).

To investigate if IL11 is expressed by Krt8+ transitional cells, we performed immunostaining for GFP and KRT8 in lung sections from BLM-treated and uninjured *IL11^EGFP* reporter mice and excluded airway

regions for quantification. In uninjured mice, KRT8 expression was limited to the airways, whereas BLM treatment resulted in the appearance of KRT8+ cells in the damaged alveolar regions (Fig. 2d and Supplementary Fig. 7e, f). There was an overlap of IL11EGFP expression in a proportion of KRT8 expressing cells in alveolar regions following BLM injury (Fig. 2e). Additionally, flow cytometry analysis of lung single cell suspension from *IL11EGFP* reporter mice for the transitional alveolar epithelial cell marker, Cldn4[14], showed that the proportions of GFP expressing Cldn4hi epithelial cells were significantly increased in the lungs after BLM-injury (Fig. 2f–h and Supplementary Fig. 8a–d), with Cldn4hi epithelial cells being the predominant IL11-expressing epithelial cell subset in the injured lung (Supplementary Fig. 8e–g).

We next sought to determine whether IL11-expressing Krt8+ transitional cells are derived from AT2 cells during lung injury. We utilized *Sftpc-CreER; R26-tdTomato* (*Sftpc-tdT*) mice to trace AT2 cells and their descendants (AT2-lineage cells) and monitored for the expression of IL11 specifically in this cell lineage after BLM injury. We exposed *Sftpc-tdT* mice to tamoxifen prior to BLM treatment and assessed the lungs 14 days post-injury (Fig. 2i). We performed immunostaining using an anti-IL11 antibody, which showed consistent overlap with anti-GFP in injured *IL11EGFP* lungs (Supplementary Fig. 7g), and counterstained for KRT8. This revealed the emergence of numerous IL11+ KRT8+ *tdT*+ cells with spread out/elongated morphologies 14 days after BLM injury (Fig. 2j and Supplementary Fig. 7h). IL11 and KRT8 immunostaining were not observed in alveolar regions of uninjured *Sftpc-tdT* mice, as expected. These findings revealed that IL11-expressing Krt8+ transitional cells are derived from activated AT2 cells after lung injury.

## IL11 induces ECM production in alveolar epithelial cells

To investigate the functional importance of IL11 in alveolar epithelial cells, we performed two-dimensional (2D) cultures of primary human pulmonary alveolar epithelial cells (HPAEpiC). By immunostaining, we first confirmed that HPAEpiC expressed high levels of SFTPC and did not stain positive for AGER (Supplementary Fig. 9a). HPAEpiC expressed high levels of IL11RA and its co-receptor IL6ST (gp130) but lacked detectable IL6R expression (Supplementary Fig. 9a).

To test whether IL11 directly induces profibrotic EMT-like processes in alveolar epithelial cells, we stimulated HPAEpiC with IL11 (5 ng/ml, 24 h) and monitored for the expression of pathologic ECM components (Collagen I, fibronectin)[7] along with KRT8 using immunostaining and immunofluorescence quantification (Fig. 3a). In parallel, we treated HPAEpiC with TGFβ1 (5 ng/ml; 24 h), a potent inducer of both EMT and KRT8 expression in AT2 cells[28–31], and simultaneously added a neutralizing IL11 antibody (X203) or an IgG control antibody to investigate the effect of IL11 signaling downstream of TGFβ stimulation (Fig. 3a). This revealed that IL11 and TGFβ1 treatment led to upregulation of Collagen I, fibronectin, and KRT8 expression, as compared to untreated epithelial cells (Fig. 3b–d). By ELISA, we found that TGFβ1 stimulation significantly induced IL11 secretion by HPAEpiC (Supplementary Fig. 9b). The effects of TGFβ1 on the expression of ECM proteins and KRT8 were significantly blunted by X203 (Fig. 3b–d).

AT2 cell proliferation is crucial for alveolar repair after injury[1,32,33] and we tested the effects of IL11 or TGFβ1 on human alveolar epithelial cell proliferation. By EdU staining, we found that exposure of cells to either IL11 or TGFβ1 (24 h) impaired HPAEpiC proliferation (Supplementary Fig. 9c–e). Furthermore, the anti-proliferative effects of TGFβ1 on HPAEpiC could be reversed by X203. These data shows that IL11 directly induces KRT8 expression and EMT processes while impairing proliferation of human alveolar epithelial cells.

Next, we performed bulk RNA sequencing (RNA-seq) of IL11- or TGFβ1-stimulated HPAEpiC (5 ng/ml, 24 h) to evaluate the transcriptional effects of these cytokines on alveolar epithelial cells. RNA-seq analysis revealed that TGFβ1 induced transcriptomic features characteristic of KRT5−/KRT17+ cells from human fibrotic lungs (such as the

elevated expression of *CDKN2A, CDKN2B, CDH2, COL1A1, FN1, SOX9, SOX4, KRT8, KRT17, KRT18,* and reduced expression of *NKX2-1)* as compared to untreated cells (Supplementary Fig. 10a, b). Along with these changes, *IL11* was amongst the top upregulated genes in TGFβ1 treated HPAEpiC (4.65-fold, *Padj* = 1.28e-62) (Supplementary Data 4).

In contrast to TGFβ1 treatment and consistent with data from other cell types[22,34], IL11 (5 ng/ml, 24 h) did not result in significant changes in global transcription levels in HPAEpiC (Supplementary Fig. 10a,b), despite inducing the expression of several ECM-related and KRT8 proteins (Fig. 2b–d). In human cardiac and lung fibroblasts, IL11-stimulated ERK activation induces profibrotic protein expression and myofibroblast differentiation[22,34]. Correspondingly, the effects of IL11 on the expression of KRT8 and ECM proteins Collagen I and fibronectin expression in HPAEpiC were blocked by the ERK inhibitor U0126 (Supplementary Fig. 10c, d), supportive of an important role for IL11-ERK post-transcriptional gene regulation in human alveolar epithelial cells. In keeping with this, we observed numerous p-ERK+ IL11+ *tdT*+ cells within injured regions of lungs from BLM-injured *Sftpc-tdT* mice (Supplementary Fig. 10e) and a similar increase in p-ERK+ GFP+ KRT8+ cells in the injured lungs of IL11EGFP reporter mice (Supplementary Fig. 10f), indicating the activation of IL11-ERK signaling in transitional alveolar epithelial cells after lung injury that was not apparent in uninjured lungs.

To provide additional evidence to support the role of IL11 in driving pathologic ECM protein expression by lung epithelial cells, we performed similar in vitro experiments on human small airway epithelial cells (HSAEC) and on primary mouse AT2 cells that were isolated from tamoxifen-exposed *Sftpc-tdT* mice by FACS sorting for constitutive *tdT*+-expressing cells and cultured these primary cells under 2D conditions (Fig. 3f–j and Supplementary Fig. 11). By immunostaining, we observed that IL11 and TGFβ1 treatment significantly increased the expression of Collagen I and fibronectin and secreted collagen by HSAEC and Collagen I expression in mouse AT2 cells (Fig. 3g–j and Supplementary Fig. 11c, e, g). Furthermore, in these cells, the effects of TGFβ on the expression of these ECM proteins were largely dependent on downstream IL11-signaling and could be blocked by X203-treatment (Fig. 3g–j and Supplementary Fig. 11d, e, g). Similarly, the effects of IL11 on the expression of these ECM proteins could be significantly blunted by ERK inhibition (Fig. 3k, l and Supplementary Fig. 11d, e, h). Taken together, our data suggests that IL11-ERK signaling induces EMT-like features in lung epithelial cell dysfunction across species.

## IL11 stalls AT2-to-AT1 cell differentiation in vitro

AT2 cells largely increase their cell area and spontaneously undergo differentiation towards AT1-like cells when cultured under prolonged 2D culture conditions. Under these conditions, AT2 cells upregulate KRT8 during early differentiation, followed by a decline of KRT8 and the subsequent upregulation of mature AT1 markers (such as PDPN) during late differentiation[13,29]. To test if IL11 stalls the transition of AT2 cells into mature AT1 cells, we isolated mouse AT2 cells from tamoxifen-exposed *Sftpc-tdT* mice and cultured these primary AT2 cells under 2D culture conditions followed by treatment with IL11 (5 ng/ml) from day 1 to day 5 (Fig. 3m). By immunostaining and cell surface area analysis of *tdT*+ cells, we found that numerous cells expressed PDPN and greatly increased their surface area by 5 days of culture in untreated cells (Fig. 3n, o and Supplementary Fig. 12). In contrast, exposure to IL11 from days 1 to 5 stalled AT1 differentiation with cells expressing higher levels of KRT8, lower levels of PDPN and with reduced cell surface area, as compared to controls (Fig. 3n, o and Supplementary Fig. 12).

Since prolonged TGFβ signaling impairs terminal AT1 maturation[17,29,35], we further hypothesized that the maladaptive effects of prolonged TGFβ-exposure on AT1 maturation might be mediated, in part, by IL11. We tested for this by first priming AT2 cells with TGFβ1 for

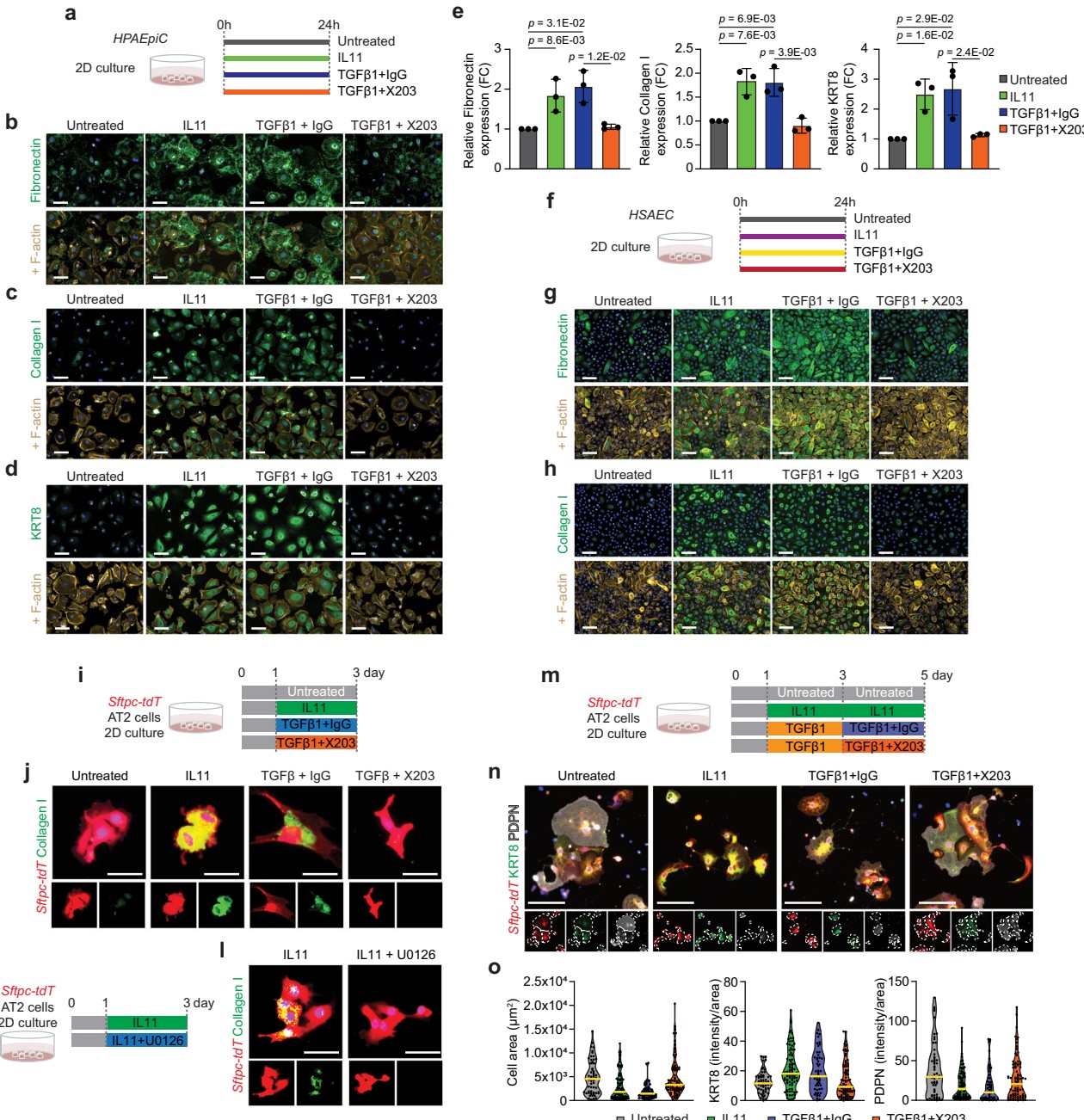

**Fig. 3 | IL11 induces KRT8 expression and profibrotic features in alveolar and distal airway epithelial cells and stalls AT2-to-AT1 cell differentiation in vitro.**
**a** Experimental design for the 2D culture of primary human pulmonary alveolar epithelial cells (HPAEpiC) treated with IL11 (5 ng/ml) or TGFβ1 (5 ng/ml) in the presence of anti-IL11 (X203) or IgG control antibodies (2 μg/ml).
**b**–**e** Immunostaining and quantification of Fibronectin, Collagen I or KRT8 expression in HPAEpiC. *n* = 3/group. FC fold change. **f** Experimental design for the 2D culture of primary human small airway epithelial cells (HSAEC) treated with IL11 (5 ng/ml) or TGFβ1 (5 ng/ml) with anti-IL11 (X203) or IgG control antibodies (2 μg/ml). **g**, **h** Immunostaining of Fibronectin or Collagen I expression in HSAEC. *n* = 2/group. Quantification data were shown in Supplementary Fig. 11c. **i** Experimental design and **j** immunostaining of Collagen I expression in *Sftpc-tdT*⁺ AT2 cells treated with IL11 (5 ng/ml), TGFβ1 (5 ng/ml), X203 or IgG control antibodies

(2 μg/ml). **k** Experimental design and **l** immunostaining images of Collagen I in *Sftpc-tdT*⁺ AT2 cells treated with IL11 (5 ng/ml) and MEK inhibitor U1026 (10 μM). Quantification of Collagen I expression are shown in Supplementary Fig. 11g, h. **m** Experimental design for the in vitro differentiation of *Sftpc-tdT*⁺ AT2 cells to AT1 cells in the presence of IL11 (5 ng/ml), TGFβ1 (5 ng/ml), X203 or IgG control antibodies (2 μg/ml). **n** Immunostaining of KRT8 and PDPN in *Sftpc-tdT*⁺ cells at day 5. Individual cells are highlighted within dotted lines in split channels. **o** Violin plots of cell size and immunostaining quantification of KRT8 or PDPN expression in *tdT*⁺ cells at day 5; One representative dataset of two independent experiments are shown (*n* > 50 cells/group). Data were representative of three independent experiments in (**b**–**d**) or two independent experiments in (**g**, **h**, **j**, **l**, **n**). Scale bars: 100 μm (**b**–**d**, **g**, **h**), 25 μm (**j**, **l**), 50 μm (**o**). Data in (**e**) are mean ± s.d. and *P* values determined by one-way ANOVA with Tukey's multiple comparison test.

2 consecutive days, followed by subsequent TGFβ1 treatment with X203 or IgG antibodies for an additional 2 days (Fig. 3m). Similar to the effects of sustained IL11 treatment, we found that cells treated with TGFβ1 followed by coincubation with IgG resulted in stalled AT1

differentiation, with cells that were less enlarged and expressed higher levels of KRT8 as compared to controls (Fig. 3n, o and Supplementary Fig. 12). On the other hand, coincubation with X203 partially-relieved the stalled AT1 differentiation phenotype and significantly increased

cell surface area and PDPN expression as compared to IgG-treated cells (Fig. 3n, o and Supplementary Fig. 12). These data show that IL11 directly promotes AT2 cell dysfunction by causing the accumulation of Krt8+ transitional cells and delaying the terminal differentiation of AT1 cells.

## IL11 signaling in AT2 cells promotes lung fibrosis in vivo

Having established that IL11 stimulation triggered EMT-related features in primary human alveolar and distal airway epithelial cells and mouse AT2 cells, we next surveyed mouse single-cell sequencing datasets and profiled the RNA expression of IL11 receptor (*Il11ra1*) in the adult mouse lung. Consistent with our human scRNA-seq analysis (Supplementary Fig. 3), *Il11ra1* was found to be highly expressed in mouse lung stromal populations (fibroblasts, smooth muscle cells), mesothelial cells and moderately expressed by macrophages and alveolar epithelial cells (Supplementary Fig. 13a–c). Notably, *Il11ra1* was consistently expressed in AT2 cells and injury emergent AT2-derived cells such as activated AT2 and Krt8 ADI, further illustrating the potential for auto/paracrine IL11-signaling across AT2-lineage cells in the injured mouse lung. We next employed a genetic loss-of-function approach to test the importance of IL11 signaling specifically in AT2 cells for lung fibrogenesis. We utilized *Sftpc-CreER; Il11ra1^{fl/fl}* mice in which *Il11ra1* could be temporally and conditionally deleted in AT2 and AT2-derived cells upon tamoxifen treatment. Mice were injected with tamoxifen 14 days prior to BLM-treatment and the lungs were assessed 12 and 21 days post-injury (Fig. 4a). Tamoxifen-exposed *Sftpc-CreER; Il11ra1^{+/+}* mice were used as controls. The deletion of *Il11ra1* in AT2 cells from tamoxifen-exposed *Sftpc-CreER; Il11ra1^{fl/fl}* mice was verified by qPCR of FACS-sorted CD31⁻ CD45⁻ EpCAM⁺ MHCII⁺ cells (Supplementary Fig. 13d, e). At baseline, the lungs of mice with AT2 cell-specific *Il11ra1* deletion appeared histologically normal (Supplementary Fig. 13f).

Further histology assessment of lungs from BLM-injured *Sftpc-CreER; Il11ra1^{+/+}* control mice at both 12 and 21 day time points indicated severe disruption to the lung architecture, increased collagen deposition and higher histopathological fibrosis scores, as compared to uninjured mice (Fig. 4b and Supplementary Fig. 14a–d). These pathologies were significantly reduced in mice where *Il11ra1* was deleted in AT2 cells. Lung hydroxyproline content was also significantly reduced in mice lacking *Il11ra1,* specifically in AT2 cells, as compared to controls (Fig. 4c and Supplementary Fig. 14c). There was a non-statistical trend of improved survival, body weights, and decreased lung weights in AT2-specific *Il11ra1*-deleted mice by the end of the 21 day study period (Supplementary Fig. 14e–g). Serum surfactant protein D (SFTPD) levels, a marker of lung inflammation and epithelial injury[36] was elevated in BLM-injured control mice but was significantly reduced in injured mice with AT2 cell-specific *Il11ra1* deletion (Supplementary Fig. 14h). Immunostaining for KRT8 revealed that KRT8-expressing cells were rarely observed in the alveolar compartment of AT2 cell-specific *Il11ra1* deleted mice post-BLM injury, as compared to controls (Fig. 4d). Furthermore, western blot analysis of lung lysates further confirmed the overall reduction in KRT8 expression and a corresponding increase in AGER protein in the lungs of *Sftpc-tdT; Il11ra1^{fl/fl}* mice as compared to controls (Supplementary Fig. 14i). Taken together, these data indicate that the loss of IL11-signaling in AT2 cells prevents the development of lung fibrosis.

## IL11 signaling potentiates fibrotic KRT8⁺ cell state in vivo

Recent studies have revealed that several pathological pathways (such as EMT, TGF-beta signaling, and p53 pathway) are highly enriched in aberrant transitional epithelial cells in human PF and in mouse Krt8+ transitional cells, and that these aberrant cells may secrete profibrotic factors and express pathologic ECM in the fibrotic lung[6,7,37]. We asked whether IL11-signaling in AT2 cells promotes fibrosis by regulating the profibrotic phenotype of aberrant transitional cells. To this end, we

performed scRNA-seq on sorted epithelial cells (CD31⁻ CD45⁻ EpCAM⁺) from the lungs of *Sftpc-CreER; Il11ra1^{fl/fl}* and *Il11ra1^{+/+}* mice 12 days post-BLM challenge (*n* = 1 mouse/uninjured groups and *n* = 2 mice/BLM-injured groups) (Fig. 4e). Our analysis recapitulated known homeostatic epithelial cell types in the lungs, such as AT2 (*Bex2, Lpl*), proliferating AT2 (*Mki67, Birc5,* and *Ube2c*), club (*Scgb1a1* and *Cyp2f2*), ciliated cells (*Tppp3* and *Dynlrb2*), and mature AT1 cells (*Igfbp2 and Cped1*). We also captured injury-emergent cell populations, including activated AT2 cells (*Lcn2, Il33*), Krt8+ transitional cells (*Cldn4, Krt8*) and immature AT1 cells (*Rtkn2, Krt8*) (Supplementary Fig. 15a, b)[13,38]. *Il11ra1* expression was reduced across AT2-lineage cells (AT2, activated AT2 and Krt8+ transitional cells) in *Sftpc-CreER; Il11ra1^{fl/fl}* mice, indicating the loss of IL11-signaling across various AT2-lineage cells in this model (Supplementary Fig. 15c).

Amongst the various AT2 cells and injury-emergent populations (Fig. 4f and Supplementary Fig. 15d), we identified a marked reduction in the proportion of Krt8+ transitional cells in *Sftpc-CreER; Il11ra1^{fl/fl}* mice post-BLM challenge (11.8 vs 1.95 %) (Fig. 4g and Supplementary Fig. 15e) which was consistent with our earlier histological analysis. We then focused on Krt8+ transitional cells. Gene set enrichment analysis (GSEA) of differentially expressed genes between *Sftpc-CreER; Il11ra1^{fl/fl}* and control Krt8+ transitional cells reveals significant downregulation of Hallmark pathways of Krt8+ ADI (FDR <0.05) such as "EMT", "TGF-beta signaling", and "p53 pathway"[13] in *Sftpc-CreER; Il11ra1^{fl/fl}* cells (Fig. 4h, Supplementary Fig. 16a, b, and Supplementary Data 5). From the leading-edge analysis of the GSEA result, we constructed a de-novo gene set of EMT, specific to Krt8+ ADI (see Methods). Notably, the transcriptomic signatures of this Krt8+ ADI EMT program, along with the expression of several ECM and profibrotic genes such as *Col1a1* ($P_{adj} = 1.9E\text{-}3$), *Fn1* ($P_{adj} = 3.3E\text{-}5$), and *Ccn2* ($P_{adj} = 3.6E\text{-}4$) were significantly downregulated in Krt8+ transitional cells from *Sftpc-CreER; Il11ra1^{fl/fl}* mice as compared to controls (Fig. 4i and Supplementary Fig. 16d and Supplementary Data 6). Taken together, abrogated IL11 signaling may attenuate the profibrotic phenotype of Krt8+ transitional cells.

We further investigated if IL11-signaling mediates the dysregulation of the differentiation from AT2 to AT1 cells. Therefore, we performed separate Slingshot[39] trajectory analysis of AT2 and injury emergent cells (activated AT2, AT1, and Krt8+ transitional cells) from BLM-injured *Sftpc-CreER; Il11ra1^{fl/fl}* and *Il11ra1^{+/+}* control mice, with origin of differentiation set at AT2 cells (Supplementary Fig. 16e). In injured controls, we found two distinct trajectories from (1) AT2 to Krt8+ transitional cells and (2) AT2 to AT1 cells. In the first trajectory predicted, the destination at Krt8+ cells implies that Krt8+ may be a terminally differentiated state, likely potentiated by disease-causing cues and reminiscent of the differentiation trajectories of AT2 to aberrant basaloid or KRT5⁻/KRT17⁺ cells in our earlier human scRNA-seq analysis (Fig. 1). In contrast, only a single trajectory from AT2 to AT1 cells was observed for cells from *Sftpc-CreER; Il11ra1^{fl/fl}* mice which suggests that AT2 cells lacking IL11 signaling may undergo effective terminal differentiation to AT1 cells.

To specifically test the hypothesis that IL11 signaling in AT2 cells promotes the accumulation of AT2-cell derived Krt8+ transitional cells and delays AT2-to-AT1 differentiation after lung injury in vivo, we crossed *Sftpc-CreER; Il11ra1^{fl/fl}* mice with R26-tdTomato (*tdT*) mice (*Sftpc-tdT; Il11ra1^{fl/fl}*) to allow the simultaneous deletion of *Il11ra1* and the constitutive expression of *tdT* specifically in AT2 and AT2-derived cells upon tamoxifen administration. *Sftpc-tdT; Il11ra1^{+/+}* mice were used as controls. We similarly injected tamoxifen 14 days prior to BLM injury and assessed the lungs of mice 12 days post-BLM treatment (Fig. 4j). Following BLM injury, immunostaining revealed numerous KRT8⁺ *tdT*⁺ cells and few newly differentiated AT1 cells (PDPN⁺ *tdT*⁺ cells) in *Sftpc-tdT; Il11ra1^{+/+}* control mice (Fig. 4k, l and Supplementary Fig. 17b). In contrast, parenchymal damage in BLM-treated *Sftpc-tdT; Il11ra1^{fl/fl}* mice was markedly reduced and KRT8⁺ *tdT*⁺ cells were rarely observed in alveolar regions from these mice, which mirrored the

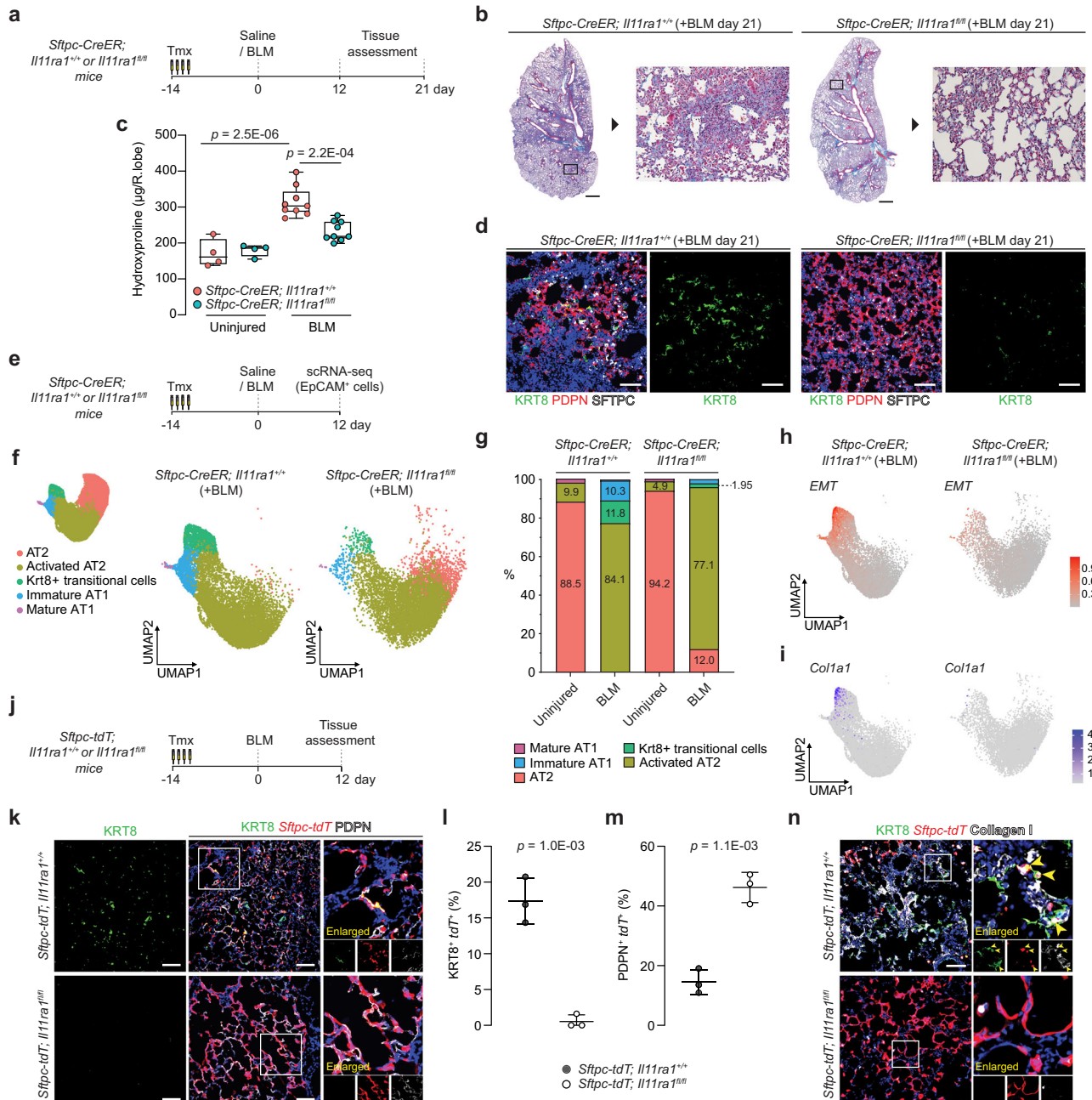

**Fig. 4 | IL11 signaling in AT2 cells disrupts AT2-to-AT1 differentiation after injury and is required for lung fibrosis. a** Schematic showing the induction of lung injury in *Sftpc-CreER; Il11ra1^{fl/fl} or Il11ra1^{+/+}* mice via oropharyngeal injection of BLM and the time points of tissue assessment. **b** Representative images of Masson's trichrome staining and **c** lung hydroxyproline content of right caudal lobes of *Sftpc-CreER; Il11ra1^{fl/fl} or Il11ra1^{+/+}* mice 21 days post-BLM injury; *n* = 4 control and 9 injured mice/genotype. **d** Immunostaining of KRT8 and PDPN in the lungs of BLM-treated *Sftpc-CreER; Il11ra1^{fl/fl} or Il11ra1^{+/+}* mice. **e** Schematic showing the induction of lung injury in *Sftpc-CreER; Il11ra1^{fl/fl} or Il11ra1^{+/+}* mice and at day 12, lung single cells were enriched for EpCAM+ epithelial cells and assessed using scRNA-seq. **f** UMAP embedding of alveolar epithelial cells from scRNA-seq analysis showing distinct cell states and **g** distribution of alveolar epithelial cell states from *Sftpc-CreER; Il11ra1^{fl/fl} or Il11ra1^{+/+}* mice 12 days post-BLM injury. **h** UMAP visualization of single cells colored by gene expression signature score for EMT pathway from the MSigDB Hallmark gene sets and **i** UMAP visualization of *Col1a1* expressing single cells (purple dots) from AT2-lineage cells from *Sftpc-CreER; Il11ra1^{fl/fl} or Il11ra1^{+/+}* mice post-BLM challenge. **j** Schematic showing the induction of lung injury in *Sftpc-tdT; Il11ra1^{fl/fl} or Il11ra1^{+/+}* mice. Lung tissues were assessed 12 days post-BLM challenge. **k** Immunostaining of KRT8 and PDPN in the lungs of BLM-treated *Sftpc-tdT; Il11ra1^{fl/fl} or Il11ra1^{+/+}* mice and the proportion of **l** KRT8^+ *tdT^+* or **m** PDPN^+ *tdT^+* relative to total *tdT^+* cells, *n* = 3 mice/group. **n** Images of immunostaining for KRT8 and Collagen I in lungs from BLM-treated *Sftpc-tdT; Il11ra1^{fl/fl} or Il11ra1^{+/+}* mice. Data were representative of three independent experiments (**d**, **n**). Data were median ± IQR and whiskers represent minimum to maximum values (**c**) and mean ± s.d. (**l**, **m**). *P* values were determined by one-way ANOVA with Tukey's multiple comparison test (**c**), two-tailed Student's *t*-test (**l**, **m**). Scale bars: 1000 μm (**b**), 100 μm (**d**, **k**, **n**).

phenotypes observed earlier with *Sftpc-CreER; Il11ra1^{fl/fl}* mice. Instead, we found numerous newly differentiated AT1 cells including regions of completely formed alveoli that were PDPN^+ *tdT^+* and AGER^+ *tdT^+* in BLM-treated *Sftpc-tdT; Il11ra1^{fl/fl}* mice (Fig. 4k, m and Supplementary Fig. 17b, c). These results show that the deletion of *Il11ra1* in AT2 cells

only reduces KRT8^+ cell accumulation and greatly enhances AT2-to-AT1 differentiation after BLM-injury.

We next validated the importance of IL11-signaling in AT2 cells for the acquisition of profibrotic transitional epithelial cell phenotypes identified from the earlier scRNA-seq findings. We performed

immunostaining of Collagen I or CTGF and KRT8 in the lungs of *Sftpc-tdT; Il11ra1^{fl/fl}* mice after BLM injury. In *Sftpc-tdT; Il11ra1^{+/+}* controls, we observed the presence of numerous Collagen I- and CTGF-expressing AT2-derived transitional cells (Collagen I^+ KRT8^+ *tdT*^+ cells and CTGF^+ KRT8^+ *tdT*^+ cells) in fibrotic regions of the lung (Fig. 4n and Supplementary Fig. 17d). In contrast, the expression of Collagen I and CTGF were markedly diminished in the lungs of BLM-challenged *Sftpc-tdT; Il11ra1^{fl/fl}* mice along with the lack of observable Collagen I- and CTGF-expressing lineage-traced cells (Fig. 4n and Supplementary Fig. 17d). Taken together, these findings further support the concept that IL11-signaling in AT2 cells impairs epithelial regeneration by promoting the differentiation of ECM-producing profibrotic KRT8^+ cells that may contribute directly to aberrant lung remodeling.

### *Il11* deletion in AT2 cells does not prevent lung fibrosis

Having determined earlier that IL11 expression is elevated in AT2 and AT2-derived Krt8+ transitional cells following lung injury (Fig. 2), we next sought to investigate whether the autocrine and/or paracrine activity of *Il11*, secreted by AT2 and AT2-derived cells is important for lung fibrogenesis. We utilized *Sftpc-CreER; Il11^{fl/fl}* mice in which *Il11* could be temporally and conditionally deleted in AT2 cells upon tamoxifen treatment. *Sftpc-CreER; Il11^{+/+}* mice were used as controls. Mice were injected with tamoxifen 14 days prior to BLM-treatment and the lungs were assessed for fibrosis 21 days post-injury (Supplementary Fig. 18). However, lung histopathological assessment and hydroxyproline content analysis revealed that BLM-injured *Sftpc-CreER; Il11^{fl/fl}* mice had comparable levels of lung collagen content and fibrosis to injured controls (Supplementary Fig. 18). There were also no apparent benefits of *Il11*-deletion in AT2 cells for survival at the end of the 21 day study period (Supplementary Fig. 18). These findings indicate that the deletion of *Il11* specifically in AT2 and AT2-lineage cells does not prevent aberrant remodeling after lung injury and further suggests that IL11 expression from non AT2-lineage cells, such as fibroblasts or airway progenitor-derived transitional cells may be of greater importance for lung fibrogenesis.

### IL11 inhibition promotes alveolar regeneration in vivo

In our previous therapeutic studies, we showed that X203-treatment significantly diminished lung inflammation and reversed established lung fibrosis in BLM-injured mice[22]. We next investigated whether anti-IL11 antibodies could promote AT2-to-AT1 differentiation and enhance alveolar regeneration when administered after lung injury. To test this, we performed BLM-induced injury to tamoxifen-exposed *Sftpc-tdT* mice followed by X203 or IgG control antibody administration starting from day 4 after injury, at a time point where alveolar KRT8^+ cells begin to accumulate, and assessed the lungs on day 12 (Fig. 5a and Supplementary Fig. 19a).

As compared to uninjured lungs, we observed widespread architectural disruption in IgG-treated mice, with a large increase in KRT8^+ cells that adopted elongated morphologies, along with a decline in the number of *tdT*^+ cells (Fig. 5b, c). Additionally, in IgG-treated mice, we found an increase in non-lineage-labeled KRT8^+ cells (KRT8^+ *tdT*^-) that stained weakly for the AT1 marker PDPN (Fig. 5b and Supplementary Fig. 19b), likely reflecting an influx of airway/progenitor cells that have committed to alveolar fates in regions of severe lung injury[13,16,40,41].

As compared to IgG-treated mice, BLM-induced parenchymal damage and fibrosis, as assessed by histopathological scoring of Masson's trichrome staining and lung hydroxyproline content was significantly attenuated by X203-treatment (Supplementary Fig. 19c–e). These changes coincided with reduced numbers of alveolar KRT8^+ cells and proportions of lineage-labeled transitional cells (KRT8^+ *tdT*^+ cells) and lineage-negative cells (KRT8^+ *tdT*^- cells) (Fig. 5b–d and Supplementary Fig. 19b). Flow cytometry analysis of lung Cldn4^{hi} *tdT*^+ epithelial cells further confirmed the reduction in the proportion of AT2-derived transitional cells following X203-treatment

(Fig. 5e and Supplementary Fig. 20a–d). Furthermore, X203-treatment partially restored *tdT*^+ cell numbers after injury to levels similar to those seen in uninjured lungs (Fig. 5b, c and Supplementary Fig 20b), which was associated with increased proliferation of surviving *tdT*^+ AT2 cells as determined by immunostaining for Ki67 (Supplementary Fig. 20e). Consistent with the role of IL11-ERK signaling in promoting a KRT8^+ cell state, as seen in vitro, we found numerous p-ERK^+ KRT8^+ cells in the lungs after BLM-injury in IgG-treated mice (Supplementary Fig. 21a). The occurrence of p-ERK^+ KRT8^+ cells were reduced in the lungs of X203-treated mice (Supplementary Fig. 21a).

Immunostaining for AT1 markers PDPN or AGER revealed that X203-treatment led to significantly enhanced differentiation of lineage-labeled cells into AT1 cells (PDPN^+ *tdT*^+ or AGER^+ *tdT*^+ cells) as compared to IgG (Fig. 5b, d and Supplementary Fig. 21b, c). Flow cytometry-based quantification of PDPN^+ *tdT*^+ epithelial lung cells and western blot analysis of KRT8 and AGER expression in lung lysates from X203 or IgG-treated mice further supported these observations (Fig. 5f and Supplementary Fig. 21d).

Lastly, to uncover additional mechanisms by which X203 prevents fibrosis, we performed similar scRNA-seq analysis on lung single cells suspensions of Cd45^- Cd31^- EpCAM^+ epithelial cells from uninjured and X203 or IgG-treated mice 12 days post-BLM (*n* = 1 mouse/group) and focused our analysis on AT2-derived injury-emergent cell populations (Fig. 5g and Supplementary Fig. 22a). Consistent with our histological and flow cytometry findings, the scRNA-seq analysis revealed that X203-treatment reduced the proportion of Krt8+ transitional cells as compared to IgG (32.5 vs. 15.0%) (Fig. 5h). Pathway analysis of differentially expressed genes in Krt8+ transitional cells revealed that the expression levels of genes related to unfolded protein response, TGF-beta signaling and EMT were modestly reduced following X203-treatment (Fig. 5i, Supplementary Fig. 22b, and Supplementary Data 7). Similar to scRNA-seq data on epithelial cells from AT2-specific *Il11ra1*-deleted mice (Fig. 4), the transcriptomic signatures of EMT-related genes in Krt8+ transitional cells were significantly reduced by X203-treatment (Supplementary Fig. 22c). Immunostaining of lungs sections for Collagen I, CTGF and for an ER-stress marker XBP1, further confirmed that the pharmacological inhibition of IL11 diminished the expression of pathologic ECM and profibrotic proteins by Krt8+ transitional cells (Collagen I^+ KRT8^+ *tdT*^+ cells, CTGF^+ KRT8^+ *tdT*^+ cells and XBP1^+ KRT8^+ *tdT*^+ cells) after BLM injury (Fig. 5j and Supplementary Fig. 22d, e).

## Discussion

Severe respiratory diseases such as IPF and SARS-COV-2 pneumonia are associated with defects in alveolar epithelial repair and irreversible loss of alveolar epithelial cells, which ultimately leads to fibrosis and lung function decline. We previously discovered an important role for IL11 in lung fibrosis, mediated via its profibrotic activity in lung fibroblasts and *IL11* expression was confirmed in diseased fibroblasts in the current study[22,25,42]. Here, we show that *IL11* is specifically upregulated in aberrant alveolar epithelial cells in human PF, and its expression is associated with pathological pro-EMT and inflammatory gene signatures in diseased epithelial cells. In complementary studies of mice with severe lung injury, we found that IL11 is expressed by activated AT2 cells, Cldn4^{hi}, and Krt8+ transitional cells.

Due to the complex signaling milieu that occurs following severe lung injury, multiple pathways likely contribute to the emergence and maintenance of Krt8+ transitional cells, among which TGFβ, which shows IL11 dependency for its effects, is of particular importance[17,35]. While inflammatory cytokines such as IL-1β and TNFα induce AT2 cell proliferation, IL-1β also primes a subset of *Il1r1*-expressing AT2 cells for differentiation into DATPS[14,43]. Intriguingly, while IL6 is a therapeutic target in some forms of PF[26,44], we show that the cell types expressing IL6 in the fibrotic lung differ from those expressing IL11 and IL11, but not IL6, expression is enriched in aberrant epithelial cells.

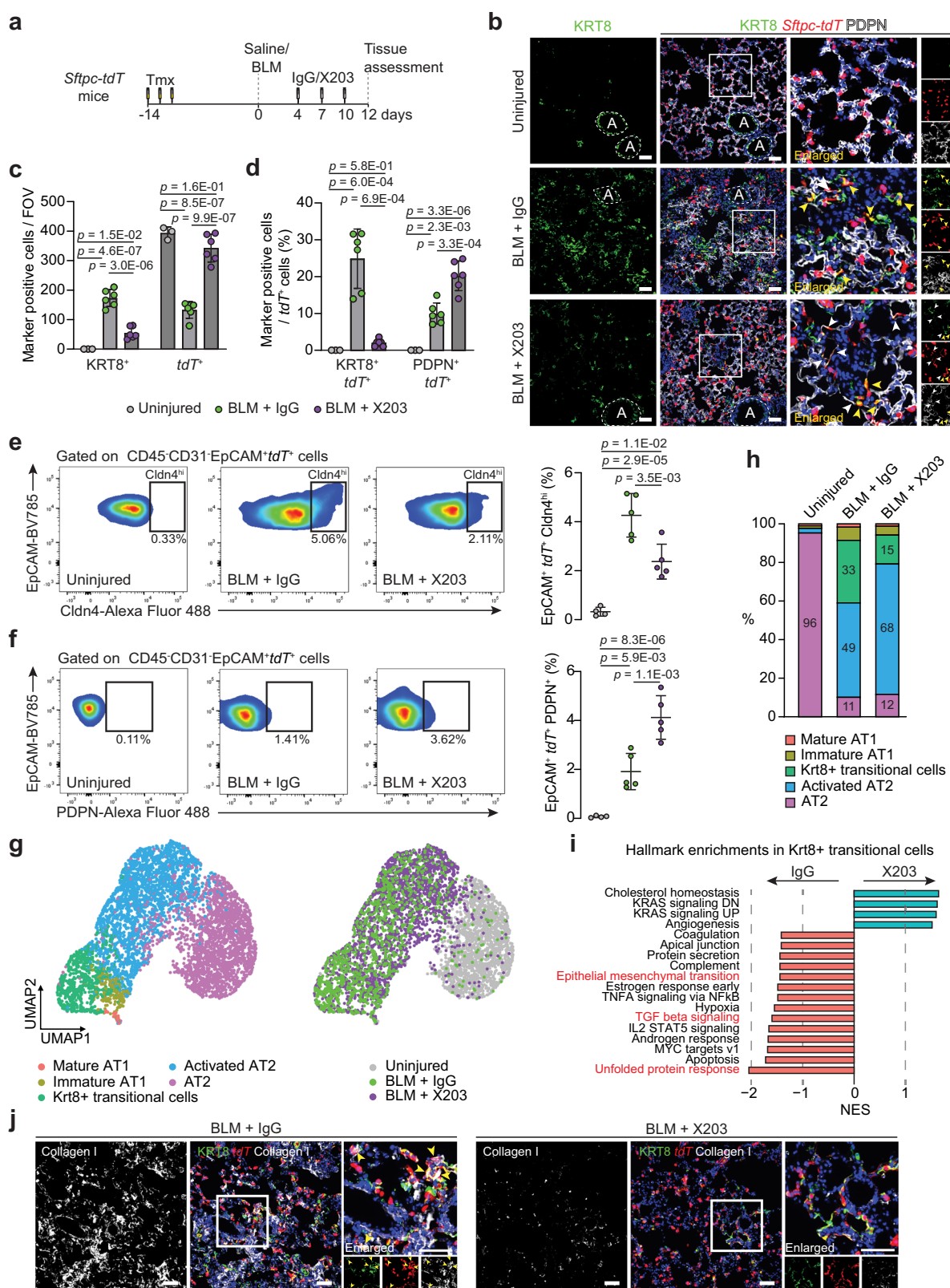

Our data identify an IL11-stimulated ERK-dependent signaling pathway that promotes and maintains AT2 cells in a profibrotic KRT8+ state and induces the protein expression of pathologic ECM by alveolar epithelial cells in vitro and in vivo. These findings further highlight the underappreciated potential of aberrant lung epithelial cells for the direct contribution of ECM components and profibrotic factors that drive pathological lung remodeling. Furthermore, we found that the

effects of TGFβ on the induction of ECM proteins and KRT8 expression in human alveolar and distal airway epithelial cells and mouse AT2 cells was, in part, mediated by IL11 signaling. Our data are consistent with a previous report that showed the importance of IL11-dependent ERK signaling in promoting EMT and senescence of AT2 cells in a *Bmi-1* deficient model of premature senescence[45]. Additionally, emerging evidence from a recent study utilizing lung epithelial cell organoids

**Fig. 5 | Pharmacological inhibition of IL11 prevents the accumulation of pro-fibrotic transitional epithelial cells and enhances AT2-to-AT1 cell differentiation during lung injury in vivo. a** Schematic showing the administration time points of BLM and X203 or IgG antibodies in *Sftpc-tdT* mice. Lung tissues were assessed 12 days post-BLM challenge. **b** Images of immunostaining for KRT8 and PDPN in lung sections from BLM-injured *Sftpc-tdT* mice treated with X203 or IgG antibodies. Yellow arrowheads indicate KRT8[+] *tdT*[+] cells. White arrowheads indicate flattened PDPN[+] *tdT*[+] cells. Recognizable airway regions are demarcated by white dotted lines. **c** Numbers of KRT8[+], *tdT*[+] per field of view (FOV) and **d** the proportions of KRT8[+] *tdT*[+] cells or PDPN[+] *tdT*[+] cells divided by the number of *tdT*[+] cells in injured lung regions (*n* = 3 control and 6 per BLM+antibody treatment group). **e** Flow cytometry analysis and quantification of lung. **h** EpCAM[+] *tdT*[+]

Cldn4[hi] cells and **f** EpCAM[+] *tdT*[+] PDPN[+] cells (*n* = 4 control and 5 per BLM+antibody treatment group). **g** UMAP embedding of alveolar epithelial cells from scRNA-seq analysis showing distinct cell states and treatment groups and **h** distribution of alveolar epithelial cell states from uninjured and BLM-injured mice (12 days post-BLM) treated with X203 or IgG antibodies. **i** Normalized enrichment scores (NES) of pathways significantly enriched in Krt8+ transitional cells from the MSigDB Hallmark gene sets. **j** Representative images of immunostaining for KRT8 and Collagen I in lung sections of BLM-injured mice treated with X203 or IgG antibodies. Yellow arrowheads indicate Collagen I-expressing KRT8[+] *tdT*[+] cells. Data were representative of at least three independent experiments (**b**, **j**). Scale bars: 100 μm (**b**, **j**). Data were mean ± s. d. and *P* values were determined by one-way ANOVA with Tukey's multiple comparison test (**c**–**f**).

showed that IL11 negatively impacted the formation of SFTPC-expressing organoids, suggesting additional roles for IL11 in causing alveolar epithelial progenitor dysfunction[46]. These findings may have implications for other airway/lung disorders such as Hermansky–Pudlak syndrome-associated pulmonary fibrosis, severe asthma, and severe viral pneumonitis, including SARS-COV-2 infection, where IL11 levels are elevated and implicated in disease pathogenesis[11,23,47–49].

Specialized lung mesenchymal cells form a supportive niche that maintains the progenitor properties of AT2 cells under homeostatic conditions[50,51]. In disease, impaired alveolar repair may arise due to disruption of this supportive niche and the development instead of a profibrotic niche composed of pathological fibroblasts and dysfunctional alveolar epithelial cells[52]. Given the elevated expression of IL11 in aberrant mesenchymal and epithelial cell types in PF and its roles in both fibroblast activation and AT2 cell dysfunction, we propose that IL11 may cause multiple aspects of pathobiology in different cell types in the diseased niche (Supplementary Fig. 23). In support of this concept, a recent study demonstrated that the expression of IL11 by pathological lung fibroblasts from ILD-patients can potentially initiate aberrant epithelial differentiation signatures in iPSC-derived alveolar organoid systems[53].

There are limitations to our study. Although several recent studies have shown that IL11 is upregulated in the lungs and *SFTPC*[+] cells from patients with IPF[22–24], in situ studies of IL11 expression in diseased human lung tissue are required to further validate these findings. Although our data on *Sftpc-CreER;Il11[fl/fl]* mice demonstrated that the specific deletion of *Il11* in AT2 cells only was not sufficient to protect mice against lung fibrosis, we did not dissect other cell type(s) expressing IL11 that can impact AT2-to-AT1 differentiation and fibrosis, despite our earlier studies suggesting a dominant role for IL11 secretion from fibroblasts for fibrosis phenotypes[25]. Moreover, conventional signaling studies on primary AT2 cells pose significant challenges due to the lack of proliferative capacity and rapid loss of primary phenotypes in vitro. Hence, the downstream molecular mechanisms of IL11 signaling in AT2 cells remain to be elucidated. In light of recent evidence highlighting the importance of distal airway secretory/basal cells in aberrant alveolar repair and fibrosis[52,54–56], the effects of IL11 on the recruitment and differentiation of airway/ progenitor cells towards KRT8[+] and AT1 cells require study.

In conclusion, we suggest that IL11 causes lung pathology in severe lung disease through at least two pathological processes. First, causing AT2 dysfunction and maintenance of a profibrotic KRT8[+] cell state, thus limiting terminal AT1 differentiation and impairing alveolar regeneration. And second, stimulating fibroblast-to-myofibroblast transformation and the expression of pathologic ECM proteins by profibrotic KRT8[+] cells that leads to lung fibrosis and inflammation[25]. Hence, anti-IL11 therapeutics, which are advancing towards clinical trials in patients with PF, may promote alveolar regeneration and mitigate lung fibrosis that would differentiate anti-IL11 therapy from anti-fibrotics currently used in the clinic.

## Methods

### Ethics

All experiments and animal procedures were approved and performed in accordance with guidelines set by the Institutional Animal Care and Use Committee at SingHealth (Singapore) and the SingHealth Institutional Biosafety Committee.

### Computational analysis of scRNA-seq datasets of human pulmonary fibrosis

Processed human PF scRNA-seq datasets by Habermann et al. and Adams et al., were downloaded from GEO with the accession number GSE135893 and GSE136831, respectively. Cell-type annotations and Uniform Manifold Approximation and Projection (UMAP) coordinates provided by the authors were used in subsequent analyses.

### Trajectory analysis

We re-classified alveolar epithelial cells in the Adams et al., dataset with cell-type annotations defined by Habermann et al., using Seurat's default label transfer pipeline. The quality of label transfer was evaluated by the Jaccard Index (See Assessment of transcriptomic similarities between epithelial cell-types below). Transitional AT2, KRT5[-]/KRT17[+], and AT1 cells were extracted from the Habermann and Adams et al. dataset for Slingshot trajectory analysis (Slingshot 1.8.0)[39], and the analysis was performed separately for each dataset. Briefly, Slingshot derives differentiation paths from a specified origin and calculates for each cell a pseudotime, which approximates the differentiation progression of a cell toward the destination of the trajectory. In this analysis, transitional AT2 cells were specified as the origin, and two differentiation trajectories were derived, one to KRT5[-]/KRT17[+] cells and the other to AT1 cells. Change in IL11 expression was evaluated along the two trajectories by fitting a generalized additive model (GAM) with the expression of IL11 against pseudotime.

### Assessment of transcriptomic similarities between epithelial cell-types

We examined transcriptional similarities of different epithelial clusters using the Jaccard index (a cluster here refers to cells of the same cell-type from a specific study, e.g., AT2 cells from Habermann et al. dataset). First, we performed differentially expressed gene (DEGs) analysis in epithelial cells from the same study, and for each cell-type retained upregulated DE genes with log2 fold change (log2FC) above the 85th percentile of the FC distribution and discarded genes with expression proportion in a cluster less than 40% compared to other cell-types. We refer to these genes as "markers" of a cluster, and a Jaccard index value was derived for all possible cluster pairing (of all epithelial clusters pooling together both datasets). A Jaccard index between cluster A and cluster B was calculated by dividing the size of the intersection of their markers over the size of the union of their markers.

### Network analysis

Cells assigned to the differentiation trajectory from transitional AT2 to KRT5[-]/KRT17[+] cells by Slingshot analysis were selected for IL11 co-

expression analysis, done individually in Habermann et al., and Adams et al., dataset. Briefly, spearman correlations were calculated between the expression of IL11 and genes expressed in the selected cells. Genes with Spearman correlation with FDR <0.2 were kept. In summary, 103 genes were found to be significantly correlated with IL11 in Adams et al dataset, 378 genes in Habermann et al. dataset, and 32 genes in both datasets. Using the R package EnrichR (enrichR 3.1.0)[39,57], functional pathway enrichment analysis was performed on genes significantly correlated with IL11 (in individual datasets and combined) querying several annotation databases including KEGG 2019 and MSigDB Hallmark 2020. Pathway terms with FDR <0.1 were retained. De-novo network construction was performed on the 32 genes significantly correlated with IL11 in both datasets. Each node in the network represents a gene and each edge (connecting a pair of genes) the Spearman correlation between the expression of the two genes in transitional AT2 and KRT5⁻/KRT17⁺ cells from Habermann et al., dataset. A graphical representation of the network was constructed in Cytoscape (Cytoscape 3.8.2)[58], and genes overlapping with the MSigDB Hallmark EMT process were colored.

### Computational analysis of scRNA-seq datasets of murine pulmonary fibrosis

Raw murine PF scRNA-seq datasets were downloaded by GEO with the following accession numbers: GSE141259, GSE184854, and GSE12703. Cell-type classifications from GSE141259 were used to annotate cell clusters in the other two datasets.

### Mouse studies

Animals were maintained in a specific pathogen-free environment and had ad libitum access to food and water, with a 12-h light/dark cycle, at an ambient temperature of 21–24 °C and humidity of 40–70%. The following mice strains were maintained on a C57BL/6 background and used for the study: Sftpc^tm1(cre/ERT2)Blh (*Sftpc-CreER*)[59], B6.Cg-*Gt(ROSA) 26Sor*^tm9(CAG-tdTomato)Hze/J (*R26-tdTomato* mice), C57BL/6-*Il11ra1*^em1Cook/J (*Il11ra1*^fl/fl mice)[25], C57BL/6-*Il11*^tm1.1Cook/J (*IL11*^EGFP reporter mice)[27], *Il11*^fl/fl mice[60]. *Sftpc-CreER* mice were crossed with *R26-tdTomato* mice to generate *Sftpc-CreER; R26-tdTomato* (*Sftpc-tdT*) mice for lineage tracing experiments. To model the deletion of *Il11ra1* in AT2 cells, *Sftpc-CreER* mice were crossed with *Il11ra1*^fl/fl mice to generate *Sftpc-CreER; Il11ra1*^fl/fl mice. Similarly, to model the deletion of *Il11* in AT2 cells, *Sftpc-CreER* mice were crossed with *Il11*^fl/fl mice to generate *Sftpc-CreER; Il11*^fl/fl mice. *Sftpc-CreER; Il11ra1*^fl/fl mice were further crossed with *R26-tdTomato* mice to generate *Sftpc-tdT; Il11ra1*^fl/fl mice. *Sftpc-tdT* mice were injected intraperitoneally with three consecutive doses of 100 mg/kg tamoxifen (Sigma-Aldrich) starting 14 days prior to bleomycin administration. *Sftpc-CreER; Il11ra1*^fl/fl mice, *Sftpc-CreER; Il11*^fl/fl mice, and *Sftpc-tdT; Il11ra1*^fl/fl mice were injected intraperitoneally with four doses of 75 mg/kg tamoxifen (Sigma-Aldrich) starting from 14 days prior to bleomycin administration. Therapeutic doses of monoclonal anti-IL11 (X203, Genovac) were established previously[22]. X203 or IgG control antibodies were injected intraperitoneally at 20 mg/kg starting from day 4 and subsequently on day 7 and day 10 post-bleomycin administration in the 12-day model of lung fibrosis.

### Bleomycin model of lung injury

The bleomycin model of lung fibrosis was performed as previously described[22]. Briefly, male mice at 10–14 weeks of age were anesthetized by isoflurane inhalation and subsequently administered a single dose of bleomycin (Sigma-Aldrich) oropharyngeally at 0.75 U/kg body weight (for *IL11*^EGFP reporter mice) or 1.5 U/kg body weight (for all other mouse strains) in a volume of saline not exceeding 50 μl per mouse. Uninjured control mice received equal volumes of saline oropharyngeally. Mice were sacrificed at indicated time points post-bleomycin administration and the lungs were collected for downstream analysis.

### Mouse lung dissociation, flow cytometry, and FACS analysis

Mice lung dissociation was performed as previously described with slight adjustments[61]. Briefly, the lungs were perfused with cold sterile saline through the right ventricle. The lungs were then intratracheally inflated with 1.5 ml of Dispase 50 U/ml (Corning) followed by installation of 0.5 ml of 1% low melting agarose (Bio-Rad) via the trachea. The lungs were excised and incubated on an orbital shaker for 45 min at room temperature. Each lobe was then minced into small pieces in DMEM (GIBCO) supplemented with 10% FBS (GIBCO) and 0.33 U/ml DNase I (Roche) and placed on the orbital shaker for an additional 10 min. The cells were then filtered through a 100 μm cell strainer and centrifuged at 400×*g* for 5 min at 4 °C. The cell pellet was resuspended in ACK-buffer (GIBCO), incubated for 2 min on ice, and then filtered through a 40 μm cell strainer. The cells were centrifuged at 400×*g* for 5 min at 4 °C and resuspended in DPBS (GIBCO) supplemented with 5% FBS, and then stained with the following antibodies: EpCAM-BV785 (BioLegend #118245), CD45-APC (BioLegend, 103112), CD31-APC/Cy7 (BioLegend, 102534), I-A/I-E - AlexaFluor488 (MHC-II) (BioLegend, 107616) and 4', 6-diamidino-2-phenylindole (DAPI) (Life Technologies, 62248) was used to eliminate dead cells. The cells were then sorted on the BD FACS Aria III system (BD Bioscience).

For flow cytometry analysis, lung single-cell suspensions were obtained as described above. The cells were then stained for the following markers (CD45-APC, CD31-APC/Cy7, EpCAM-BV785, PDPN-FITC; all antibodies at 1:200 dilution), fixed in 4% paraformaldehyde, permeabilized with 0.1% triton-X in DPBS, and stained for intracellular proteins in this order: Firstly, cells were stained with anti-Claudin 4 primary (Invitrogen, 36-4800, 1:100) and anti-rabbit Alexa Fluor 488 secondary (Invitrogen, A32731, 1:200) antibody, followed sequentially by PE-conjugated anti-GFP primary antibody (Abcam, ab303588) staining to prevent potential binding and overlap of Alexa Fluor 488 secondary and anti-GFP signals. The cells were then analyzed on the BD LSR Fortessa system (BD Biosciences) and data was analyzed using FlowJo software (Tree Star).

### Human and mouse cell cultures

Human pulmonary alveolar epithelial cells (HPAEpiC) (ScienCell Research Laboratories, 3200) were supplied at passage 1, cultured in complete AEpiCM (ScienCell Research Laboratories), and were directly used for experiments after 24 h of acclimatization. Human small airway epithelial cells (HSAEC) (Lonza Bioscience, CC-2547) were cultured in SAGM™ small airway epithelial cell growth medium kit (Lonza Bioscience, CC-3118) and used for experiments at passage 3. Briefly, HPAEpiC or HSAEC were seeded at a density of 1.5e4 cells per well in 96-well CellCarrier plates (PerkinElmer) or 3e5 cells per well in six-well tissue culture plates (Corning). HPAEpiC and HSAEC were synchronized in AEiCM basal medium or SABM™ basal medium, respectively, for 16 h prior to cytokine or antibody treatment. For the assessment of cell proliferation, cells were pulsed with EdU for 22 h prior to cell fixation and stained using the Click-iT EdU Labeling kit (Thermo Fisher Scientific, C10350) according to the manufacturer's protocol. For mouse AT2 cell cultures, *tdTomato* positive (*tdT*⁺) cells were isolated from *Sftpc-tdT* mice lungs by FACS sorting for live CD45⁻ CD31⁻ EpCAM⁺ *tdT*⁺ cells. The FACS-sorted cells were then seeded at a density of 2e4 cells per well in rat tail collagen (Invitrogen, A1048301) coated 96-well CellCarrier plates (PerkinElmer) and cultured in DMEM supplemented with 10% FBS. Mouse AT2 cells were allowed to adhere for 24 h prior to cytokine or antibody treatment. The various cytokines and antibodies used for in vitro experiments are as follows: Recombinant human IL11 (UniProtKB:P20809, GenScript), recombinant human TGFβ1 (PHP143B, Bio-Rad), anti-IL11 antibody (X203, Genovac), IgG antibody (IIE10, Genovac), U0126 (Cell Signaling Technology, 9903), recombinant mouse IL11 (UniProtKB: P47873, GenScript), recombinant mouse TGFβ1 (R&D Systems, 7666-MB).

## In vitro immunofluorescence imaging and analysis

Immunofluorescence imaging and quantification of HPAEpiC, HSAEC, and mouse AT2 cells were performed on the Operetta High-Content Imaging System (PerkinElmer) as previously described in ref. 22. The cells were first fixed in 4% paraformaldehyde (Thermo Fisher Scientific) and permeabilized with 0.1% Triton X-100 in phosphate-buffered saline (PBS). The cells were then incubated with the following primary antibodies against: KRT8 (Merck Millipore, MABT329, 1:100), Collagen I (Abcam, ab34710, 1:100), fibronectin (Abcam, ab2413, 1:100), IL11RA (Abcam, ab125015, 1:100), gp130 (Thermo Fisher Scientific, PA5-28932, 1:100), IL6RA (Thermo Fisher Scientific, MA1-80456, 1:100), SFTPC (Santa Cruz, sc-518029; 1:100), or AGER (R&D Systems, MAB1179, 1:100) and visualized using Alexa Flour 488-conjugated secondary antibodies. Cellular morphology was assessed by counterstaining with Phalloidin-iFluor 555 reagent (Abcam, ab176756). The permeabilization step was omitted for membrane staining of gp130, IL6RA, and IL11RA. Plates were scanned and images were collected with the Operetta high-content imaging system (PerkinElmer). The percentages of proliferating cells (EdU+ve cells) were quantified using the Harmony software version 3.5.2 (PerkinElmer). Each treatment condition was run in duplicate wells, and 7 to 14 fixed non-overlapping fields were imaged and analyzed per treatment group. Quantification of immunofluorescence of protein markers in HPAEpiC and HSAEC experiments was performed using the built-in cell analysis tool on the Columbus software (version 2.7.2, PerkinElmer). To determine the fluorescence intensities for each cell, individual cells were denoted based on the DAPI nuclei staining, and cell areas were established based on the total cytoplasmic Alexa Fluor 488 signal. Fluorescence intensities of cytoplasmic Alexa Fluor 488 signals within each demarcated cell area were concurrently measured, and fluorescence intensities for each cell were then further normalized to their respective area. Mean intensity/area per analyzed field are presented as one datapoint. Quantification of immunostaining intensity of KRT8, PDPN, and Collagen I in mouse AT2 cells were analyzed by Fiji software and fluorescence intensities were normalized to cell area.

## RNA-seq

Total RNA was isolated from HPAEpiC with or without TGFβ1 or IL11 stimulation using RNeasy columns (Qiagen). RNA was quantified using Qubit™ RNA Broad Range Assay Kit (Life Technologies) and assessed for degradation based on RNA Quality Score (RQS) using the RNA Assay and DNA 5 K/RNA/CZE HT Chip on a LabChip GX Touch HT Nucleic Acid Analyzer (PerkinElmer). TruSeq Stranded mRNA Library Prep kit (Illumina) was used to assess transcript abundance following standard instructions from the manufacturer. Briefly, poly(A) + RNA was purified from 1 μg of total RNA with RQS >9, fragmented, and used for cDNA conversion, followed by 3′ adenylation, adapter ligation, and PCR amplification. The final libraries were quantified using Qubit™ DNA Broad Range Assay Kit (Life Technologies) according to the manufacturer's guide. The quality and average fragment size of the final libraries were determined using DNA 1 K/12 K/Hi Sensitivity Assay LabChip and DNA High Sensitivity Reagent Kit (PerkinElmer). Libraries with 16 unique dual indexes were pooled and sequenced on a NextSeq 500 benchtop sequencer (Illumina) using the NextSeq 500 High Output v2 kit and 75-bp paired-end sequencing chemistry.

## RNA-seq analysis

Libraries were demultiplexed using bcl2fastq v2.19.0.316 with the --no-lane-splitting option. Adapter sequences were then trimmed using trimmomatic v0.36[62] in paired end mode with the options MAXINFO:35:0.5 MINLEN:35. Trimmed reads were aligned to the Homo sapiens GRCh38 using STAR v.2.2.1[63] with the options --outFilterType BySJout --outFilterMultimapNmax 20 --alignSJoverhangMin 8 --alignSJDBoverhangMin 1 --outFilterMismatchNmax 999 --alignIntronMin 20 --alignIntronMax 1000000 retained for counting. Counts were calculated at the gene level using the FeatureCounts module from subread v.1.5.1[64], with the options -O -s 2 -J -T 8 -p -R -G. The combined transcript model annotation file was constructed using Ensembl hg38 and FANTOM5 hg38 as previously described[65] and used as an annotation to prepare STAR indexes and for FeatureCounts. Differential expression analyses were performed in R v4.2.0 using the Bioconductor package DESeq2 v1.36.0[66], using the Wald test for comparisons. For sample groups, the design for the model was specified as ~ stimulus (IL11/TGFβ1/baseline) + source (commercial tube 1–4), to account for the confounding effect of different batches of cells.

## Sample preparation for mouse lung scRNA-seq

Mouse lung single-cell suspensions were generated and stained with antibodies as described above for FACS analysis. Lung epithelial cells were enriched by sorting for live CD45− CD31− EpCAM+ cells and the cells from different mice were then labeled with unique sample oligo barcodes (sample-tag) using BD™ Ms Single Cell Sample Multiplexing Kit (BD Biosciences) and evaluated for cell concentration and viability using C-Chip disposable hemocytometer (NanoEnTek) on a BD Rhapsody Scanner (BD Biosciences).

## Single-cell capture, cDNA library construction, and sequencing

Single-cell capturing was performed using the BD Rhapsody™ Express Single-Cell Analysis System (BD Biosciences). Libraries were generated using BD Rhapsody Whole-Transcriptome Analysis (WTA) Amplification kit according to the manufacturer's protocols. Briefly, 15,000 cells from each uniquely tagged sample were loaded together with beads with oligonucleotide barcodes onto the cartridge containing microwells. Individual cells were lysed allowing the hybridization of mRNA-sample-tag molecules with beads before pooling and cDNA conversion. Sample-tags with barcode information were denatured off of the beads, PCR-amplified, and indexed to generate sample-tag libraries. Then, random primers were hybridized to the cDNA on the remaining beads without sample-tags, followed by an extension with an enzyme. Second-strand complementary DNA were then synthesized and ligated with adapters for PCR amplification to generate the whole-transcriptome libraries. Next, the sample-tag and whole-transcriptome libraries were combined and spiked with 5% PhiX genome to increase the library complexity. The final libraries were subsequently sequenced on a NovaSeq sequencer (Illumina) using a 150-bp paired-end run.

## Single-cell data pre-processing and analysis

Raw sequencing data were demultiplexed using bcl2fastq v2.19.0.316 with the --no-lane-splitting option. FASTQ files were demultiplexed using the unique sample-tags, trimmed, mapped, and annotated using the BD Rhapsody™ Sequence Analysis Pipeline (Revision 2.0) on the Seven Bridges Genomics platform (accessed on June 2023). Low-quality read pairs were removed based on read length, average base quality score, and highest single-base frequency. Filtered reads were aligned to the Mus musculus genome (GRCm39 assembly) using STAR v2.7.4a embedded in the pipeline and annotated using the mouse GENCODE release M31 GTF. Reads with identical cell labels, identical unique molecular identifier (UMI) sequences, and identical genes were collapsed into a single raw molecule, followed by removing artifacts using recursive substitution error correction (RSEC) developed by BD Biosciences. Cells that had been identified as doublets or labeled as "undetermined" and genes that were expressed in <3 cells were removed. Next, cells with expression of ≥200 genes, ≥500 unique molecular identifier (UMI) counts, >0.8 log10GenesPerUMI, and mitochondrial gene fraction <30% were further processed. Seurat v5.0.1 was used to perform anchor-based CCA integration on the datasets. The top eight integrated components were used for uniform manifold approximation and projection (UMAP) dimensional reduction before rejoining layers and data visualization in two dimensions.

Unsupervised clustering was performed by the FindClusters function using a shared nearest neighbor (SNN) modularity optimization-based clustering algorithm. The resulting clusters were manually labeled based on the top differentially expressed genes in each cluster against all other clusters and conserved among the groups.

### Pseudobulk differential expression, GSEA, and AddModuleScore analyses

Krt8+ transitional epithelial cells labeled as Krt8ADI (*Krt8*+, *Cldn4*+) were subset from the entire epithelial cell dataset. Raw counts in all Krt8ADI cells were extracted from the Seurat object and subsequently aggregated to form count matrices for each sample. Differential expression analyses were performed in R v4.2.0 using the Bioconductor package DESeq2 v1.38.3, using the Wald test for comparisons. Gene set enrichment analyses (GSEA) were run using the fgsea v.1.22.0, pre-ranking the gene list by the "stat" column of the DESeq2 results output, and $10^5$ permutations against mouse MSigDB (msigdbr v.7.5.1) "Hallmark"[67] mouse gene sets. Enriched gene set activities were calculated using the AddModuleScore Seurat function to compare the expression of the genes of interest as defined by the GSEA leading-edge analysis. Wilcoxon test was then performed to study the difference in gene set activities between genotypes or treatment groups. Visualizations were generated with Seurat, Nebulosa, scCustomize, and ggplot2 R packages[68,69]. For trajectory analysis, AT2, Activated AT2, Krt8ADI, and immature and mature AT1 were subset from the alveolar epithelial cell dataset and split into two different Seurat objects based on the genotypes for Slingshot trajectory analyses (v2.6.0). Pseudotime estimate, lineage(s) assignment, and simultaneous principal curve(s) were constructed by Slingshot *getLineages()* and *getCurves()* functions for different genotypes to infer cell transitional stages.

### Lung histology and immunohistochemistry

Mouse lung tissue (left lobes) were fixed in 10% formalin for 16–20 h, dehydrated and embedded in paraffin, and sectioned (7 μm) for Masson's trichrome staining as described previously[22]. Histological analysis for fibrosis was performed blinded to genotype and treatment exposure as previously described[22]. For immunostaining, the lungs were fixed in 4% paraformaldehyde at 4 °C for 16 h, followed by serial 15 to 30% sucrose in PBS dehydration for 48 h. The tissues were then embedded in an OCT compound prior to sectioning (10 μm). The sections were incubated overnight at 4 °C with the following primary antibodies: KRT8 (Merck Millipore, MABT329, 1:100), p-ERK (Cell Signaling Technology, 4370 or 5726, 1:100), Ki67 (Abcam, ab16667, 1:100), SFTPC (Abcam, ab211326, 1:100), GFP (Abcam, ab290/ab6673, 1:100), AGER (R&D Systems, MAB1179, 1:200), Podoplanin (R&D Systems, AF3244, 1:200), IL11 (Invitrogen, PA5-95982, 1:100), CD45 (Proteintech, 20103-1-AP, 1:50), PDGFRA (R&D Systems, AF1062, 1:100), Collagen I (Abcam, ab21286, 1:100), CTGF (Abcam, ab6992, 1:100), XBP1 (Abcam, ab37152, 1:100). Alexa Fluor-conjugated secondary antibodies (Invitrogen, 1:500) were incubated at room temperature for 60 min. Nuclei were stained with DAPI (Invitrogen, 1:1000). Images were captured using the Leica DMi8 microscope (Leica Microsystems) with a 20X or 40X objective. The cells were counted based on positive staining for immunohistochemistry markers and DAPI using Fiji software. Five to ten non-overlapping images for each unique marker were analyzed per mouse lung, and the mean values per mouse were presented.

### Colorimetric assays

Detection of secreted IL11 into the supernatant of HPAEpiC and HSAEC cultures were performed using the human IL-11 ELISA kit (R&D systems, D1100) according to manufacturers' instructions. Detection of SFTPD in mouse serum was performed using the mouse SP-D ELISA kit (ab240683) according to the manufacturer's instructions. Total lung hydroxyproline content of the right lung of *Sftpc-CreER; Il11$^{fl/fl}$* mice or the right caudal lung lobes of *Sftpc-tdT and Sftpc-CreER; Il11ra1$^{fl/fl}$* mice were measured using the Quickzyme Total Collagen assay kit (Quickzyme Biosciences) as previously described[22]. Soluble collagen in HSAEC culture supernatants were quantified using the Sirius Red collagen detection kit (9062, Chondrex) following the manufacturer's protocol.

### Western blot

Total proteins were extracted from mouse right lung tissues using RIPA lysis buffer (Thermo Fisher Scientific) containing protease and phosphatase inhibitors (Thermo Fisher Scientific). Protein concentrations were determined by a BCA protein assay kit (Thermo Fisher Scientific). Protein lysates were separated by SDS-PAGE before being transferred onto PVDF membranes and stained with the following primary antibodies against: RAGE/AGER (Proteintech, 16346-1-AP), KRT8 (Merck Millipore, MABT329) or GAPDH (Cell Signaling, 2118). Blots were then incubated with the appropriate secondary antibodies before being visualized using ECL Western Blotting substrate (Thermo Fisher Scientific).

### Statistical analysis

Statistical analyses for in vitro and in vivo data were performed using GraphPad Prism (v9). Analyses of experimental data were performed using two-tailed Student's *t*-test or one-way ANOVA, as indicated in the figure legends. For comparisons between multiple treatment groups, *P* values were corrected for multiple testing using Tukey's test. *P* values <0.05 were considered statistically significant.

### Reporting summary

Further information on research design is available in the Nature Portfolio Reporting Summary linked to this article.

## Data availability

All data associated with this study are presented in the paper or in the Supplementary Materials. Raw RNA sequencing data generated for this study have been uploaded onto Gene Expression Omnibus under the accession GSE261794. Source data are provided in the Supplementary Information/Source Data file. Source data are provided with this paper.

## Code availability

All codes generated for this study are available on Zenodo repository: [https://doi.org/10.5281/zenodo.13315637] and the GitHub link: [https://github.com/henryhuang12345/IL11_AEC_2024].

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

## Acknowledgements

This work was supported by the National Medical Research Council (NMRC) Singapore (NMRC/STaR/0029/2017 to S.A.C., MOH-OFIRG21nov-0006 to A.A.W., and NMRC-OFYIRG21jun-0022 to B.N. and S.V.), NMRC Central Grant to NHCS (MOH-CIRG18nov-0002 to S.A.C.), Goh Cardiovascular Research Award (Duke-NUS-GCR/2015/0014 to S.A.C.), Tanoto Foundation (to S.A.C.), Advanced Manufacturing and Engineering Young Individual Research Grant (AME YIRG) of Agency for Science, Technology and Research (A*STAR) award (A2084c0157 to W.-W.L. and B.N.); and a research grant from Boehringer Ingelheim. We thank Dr. Jinrui Dong for her support with in vivo experiments and Mr. Daryl Yeong for his support for histology. We thank Laura and Charles Lou at AMC research core services for their technical support for FACS analysis.

## Author contributions

B.N. and S.A.C. conceptualized the study. K.Y.H. and C.J.P. performed computational analyses. B.N., F.F.K., B.L.G., and S.V. performed in vitro and ex vivo experiments. B.N., W.-W.L., F.F.K., and Y.-N.L. performed in vivo studies and histological evaluations. C.J.P. and A.A.H. performed scRNA-seq and bulk RNA-seq experiments. B.L.G. performed animal genotyping and provided administrative support. B.N., E.P., and S.A.C. supervised the study. B.N., A.A.W., and S.A.C acquired funding for the study. Data were curated, processed, and visualized by B.N., K.Y.H., C.J.P., and W.-W.L. B.N. and S.A.C. drafted, reviewed, and edited the manuscript with input from all other authors.

## Competing interests

S.A.C. is a co-inventor of the patent applications (WO/2017/103108) and (WO/2018/109170). S.A.C., W.-W.L., and B.N. are co-inventors of the patent application (WO/2019/073057). S.A.C. is a co-founder and shareholder of Enleofen Bio PTE LTD, a company that develops anti-IL11 therapeutics, which were acquired for further development by Boehringer Ingelheim. The remaining authors declare no competing interests.
