## [Peer Review File · Nature Communications]

Interleukin-11 causes alveolar type 2 cell dysfunction and prevents alveolar regenerationEditorial note: Parts of this Peer Review File have been redacted as indicated to remove third-party material where no permission to publish could be obtained.

REVIEWER COMMENTS

Reviewer #1 (Remarks to the Author):

Ng et al ask whether AT2 cell IL11 receptor signaling promotes pulmonary fibrosis. Although it is known that IL11 promotes pulmonary fibrosis it has remained unclear if these effects were due to inhibiting AT2 to AT1 transition. The authors here nicely leverage existing human PF scRNA data sets and several robust experimental models to show that IL11R signaling stalls AT2 differentiation and promotes the expansion of KRT8+ transitional cells with pronounced EMT expression profiles. This is a very well written manuscript with clearly described experimental designs and findings that sheds light into poorly understood mechanisms of IL11-mediated fibrosis. I have very few concerns:

1. Define IRE1 alpha as inositol-requiring transmembrane kinase/endoribonuclease 1 α
2. The rarity of the IL11+EGFP+ EpCAM+ cells in BLM suggested by the FACS data in Fig. 2A seems to contradict the prolific KRT8+ GFP immunostaining shown in Fig. 2C. It is well known that measuring EGFP expression fluorescence is challenging with highly autofluorescent lung epithelial cells. Given that KRT8 (Mab clone 3G9) and EGFP (clone EPR14104) antibodies also work in FACS assays it would inspire confidence in the data as well as provide critical quantitative assessments of IL-11+KRT8+ transitional cells during disease progression if FACS analysis were conducted with anti-EGFP staining on a EpCAM+ KRT8+ and KRT8- gate.
3. The differences in KRT8 expression are not very convincing in Fig 3f weakening the argument that AT1 differentiation is stalled by IL11 in the 2D culture system. A quantitative assessment of KRT8 expression would provide more evidence for the effects of IL11 on inhibiting AT1 terminal differentiation.
4. A schematic providing the relationship between IL11 TGF β AT2 to KRT8+ transitional or

AT1 differentiation would help better highlight the author's findings.

Reviewer #2 (Remarks to the Author):

In the alveoli, alveolar type 2 (AT2) cells play a crucial role in maintaining lung homeostasis and facilitating regeneration following injury by proliferating and differentiating into new alveolar type 1 (AT1) cells specialized for gas exchange. Given the vital role of AT2 cells, their self-renewal and differentiation must be carefully orchestrated to uphold tissue integrity and promote efficient repair. Recent studies suggest that Krt8+ AT2+ lineage populations emerge during tissue repair following lung injury. However, chronic inflammation impedes AT1 differentiation and results in the accumulation of Krt8+ cells and impaired alveolar regeneration. IL-11, a member of the IL-6 family of cytokines, binds to IL-11Ra1 and transmits the signal through a protein called glycoprotein 130 (gp130). The authors previously reported that Il11ra1-deficient mice are protected from fibrosis in the bleomycin (BLM) mouse model of pulmonary fibrosis (PF). Treatment with an anti-IL-11 antibody they made also inhibited inflammation and fibrosis in the same bleomycin-induced PF mouse model (Ng et al., *Sci. Transl. Med.*, 2019). Furthermore, the authors reported that conditional deletion of Il11ra1 using Col1a2-CreERT mice led to a significant inhibition of fibrosis (Ng et al., *FASEB J*, 2020).

In this study, Ng et al. aimed to demonstrate the role of IL-11 signaling in the KRT8+ cells (the murine counterpart of human KRT5- KRT17+ cells) during BLM-induced PF models. The authors first demonstrated that IL-11 was expressed in AT2 and KRT5-KRT17+ cells using human single-cell-RNA-seq data and in KRT8+ cells in BLM-treated IL-11 reporter mice. The authors stimulated 2D-cultured human and mouse alveolar epithelial cells with IL-11 and demonstrated that IL-11 increased KRT8 expression and EMT-related genes. In addition, the authors generated conditional deletion of Il11ra1 using SfpccreER mice. They showed that the deletion of Il11ra1 in the lineage of AT2 cells reduced numbers of KRT8+ cells, thereby attenuating lung fibrosis and promoting AT1 cell differentiation. Although the presented results provide further insight into the mechanisms underlying the IL-11-dependent promotion of lung fibrosis, several concerns must be addressed. The following are specific comments.

Major comments:

1. The authors' previous study found that deletion of *Il11ra1* in *Col1a2*⁺ cells (fibroblasts) reduced BLM-induced lung fibrosis. However, this study proposes a crucial role for *IL-11ra1* in AT2 cell-derived lineage cells using *Spfc-CreERT2* mice. The authors need to discuss and reconcile these seemingly inconsistent findings, specifically addressing whether fibroblasts and AT2 cell-derived cells contribute to exacerbating lung fibrosis and if AT2 or *KRT8*⁺ cells express *Col1a2*. Additionally, scRNA-seq analysis of human PF patients suggests that the primary source of IL-11 may be fibroblasts or myofibroblasts, with lower expression in *KRT17*⁺ cells and AT2 cells. The authors should address the cellular origin of IL-11 and its relative contribution to promoting fibrosis in PF patients or BLM-induced lung fibrosis model.
2. In Figure 1 and Supplementary Figure 1, to gain a comprehensive understanding of PF patients' lungs, the authors should include all cell types, including fibroblasts and immune cells, in their UMAP analysis in Supplementary Figure 1. Additionally, the authors should include the expression of the *Il11ra1* gene in Figures 1a and b.
3. In Figure 1g-h, to improve transparency, the authors should provide a full list of genes in the *Il11* co-expression module as Supplementary Data.
4. In Figure 1i, the authors should provide a clearer explanation for the density plot and specify the purpose of the y-axis.
5. In Figure 2h, the authors should identify the cells that are *Sftpc*-negative *KRT8*-positive.
6. In Figure 3b, the authors analyzed the expression of EMT-related molecules in human pulmonary alveolar epithelial cells following stimulation with IL-11, *TGFb1*+IgG, or *TGFb1*+X203. Of note, the morphology of cells appeared to be different in cells stained with anti-Fibronectin, anti-Collagen I, or anti-SNAIL antibodies despite the same treatment. The authors need to clarify these points. Alternatively, the authors should change to representative images. Moreover, SNAIL is a transcriptional factor that should be localized in the nucleus. However, the data suggests that SNAIL is predominantly localized in the

cytoplasm. The authors need to explain this discrepancy.

7. In Figure 3d, the authors should provide the image data to create this graph.

8. To evaluate the changes in cell morphology more accurately, the authors should provide images of the cells on day 1 and day 3 together in Figure 3f.

9. In Figure 4, Western blotting analysis of KRT8, PDPN, and tdT using lung tissue homogenates before and after BLM would be informative.

10. In Figure 5, to calculate cell numbers more quantitatively, the authors should measure them by flow cytometry but not count them by immunohistochemistry, as the cell margin among cells seems obscure.

11. In Supplementary Figure 5c, according to the Materials and Methods section, anti-GFP and anti-IL-11 antibodies were raised by rabbit, and co-immunostaining is technically impossible using these antibodies. The same applies to Supplementary 6g, since anti-ERK and anti-IL-11 antibodies were raised in rabbit.

12. In Supplementary Figure 6a, gp130 and IL11RA are membrane proteins. However, the positive signals are not observed in the cell membrane but are also in the nucleus, suggesting that the specificity of these antibodies is questionable. The authors should use other antibodies that can specifically detect these molecules or verify their specificity using knockdown cells of these genes. Negative signals using IgG control antibody does not ensure the specificity of these antibodies.

13. In Supplementary Figure 6c, to understand the whole picture of the genes induced by TGFb-induced IL-11 and IL-11-induced ERK pathway, the authors should perform RNA-seq analysis of cells stimulated with TGFb + IgG, TGFb+X203, and IL-11+U0126. Then the authors compared gene expression profiles between TGFb1+IgG vs. TGFb1+X203 and IL-11 vs. IL-11+U0126.

Moreover, the authors should explain Log2 fold change, and Log2 mean expression in detail for readers unfamiliar with RNA-seq analyses.

14. In Supplementary Figure 6e, although the authors discussed the post-translational regulation of EMT-related genes by IL-11, the quality of the imaging data is not sufficient. The authors should demonstrate whether IL-11 regulates the EMT-related proteins through post-translational regulation by Western blotting.

15. In Supplementary Figure 6f, given that U0126 attenuated the expression of EMT-related genes, it would be interesting to test whether a MEK inhibitor, such as Trametinib, attenuates BLM-induced fibrosis. This experiment would further strengthen the importance of the present study.

Minor comments:

1. In Supplementary Figure 7b, gating strategy of AT cells was not described.

Reviewer #3 (Remarks to the Author):

The study by Benjamin Ng et Al titled “Interleukin-11 causes alveolar type 2 cell dysfunction and prevents alveolar regeneration” follows up a previous study from the same group on the role of IL11 in the pathogenesis of Idiopathic Pulmonary Fibrosis (Benjamin Ng et Al, Sci Transl Med. 2019). In the Sci Transl Med. 2019 paper, Benjamin Ng and colleagues proposed a model in which IPF lung fibroblasts secretes IL-11 and IL-11 promotes fibroblast to myofibroblast activation, thus generating a profibrotic positive feedback that is fundamental for fibrosis development in the bleomycin-mouse.

In the current study, Benjamin Ng and colleagues, shifts the focus on ATII cells. This study aims at defining the role of IL11 in alveolar regeneration with a specific focus on the role of IL11 in regulating ATII to ATI trans-differentiation a known and physiological process of alveolar re-epithelialization. It is known that KRT5-/KRT17+ epithelial cells are disease-associated transitional AT2 cells, and these cells accumulate in the lungs of patients with IPF as previously shown, ref 6 and 7 in original manuscript. These KRT5-/KRT17+ cells were shown to have a high similarity with the mouse KRT8+ cells enriched in the mouse lung after injury, as shown in ref 13-14-15 in original manuscript.

The topic of ATII to ATI trans-differentiation is, in this moment, very relevant in the field of lung regeneration and any findings in this context could represent a consistent advance in better understanding the pathophysiology of IPF disease and eventually pave the way to the development of new and more effective therapies. However, I am frankly concerned by the overall structure and consistency of this paper. Here below it will follow a detailed list of major comments that authors should address to adequately support their findings.

Major comments

1) The study is based on the re-analysis of large-scale scRNA-seq data of lung cells from patients with pulmonary fibrosis from two independent studies by Habermann et. al., and Adams et. al. (GSE135893 and GSE136831 respectively). The authors claims that the IL11-expressing aberrant ATII cells are specifically enriched in IPF patients vs Control and this is well supported by the data shown in Figure1A-C and Supplementary Figure1. Afterwards, the authors tried to parallel the human findings in a mouse model of acute lung damage (bleomycin mouse model). Obviously, this is a critical point of the paper, since all the other experiments are conducted in mouse models of bleomycin-induced lung damage. The authors claims that “after bleomycin injury the proportions of IL11EGFP+ cells in hematopoietic (CD45+ CD31-; P=.0136), epithelial (CD45- CD31- EpCAM+; P=.0002) and stromal cell populations (CD45- CD31- EpCAM-; P=.0200) were found increased. These data are reported in Figure2A, B and supplementary Figure4”. The shifts of IL11EGFP+ cells in the bleomycin treated condition, shown in the representative flow cytometry analysis chart, is really minimal, with very few IL11EGFP+ in the epithelial population (0.150%). On the contrary in figure 2c they are showing IFs with a remarkable number of IL11+ cells, in particular at 7 days post bleomycin treatment.

Could authors please comment on this?

Could authors please confirm their findings by showing an IF experiment using an antibody targeting IL-11 instead of the genetically encoded reporter gene?

2) In Figure2, and in all other panels including IF images, authors are always showing single field of view at medium magnification. Authors should include the stitches of multiple images reconstructing the entire lung organ. This will be much more convincing and therefor better supporting the conclusions.

3) Page 7, line 197, Authors claim “IL11EGFP was localized to numerous SFTPC+ cells adjacent to regions of tissue disruption with an elongated morphology suggestive of AT2-to-AT1 differentiation (Fig. 2c). The elongated shape is not a commonly recognized marker for ATII to ATI transitioning phase, include citations of the relevant literature supporting this statement or provide experimental evidence.

4) In Figure 2e, authors should select a different region of the organ, possibly with no bronchi that are by definition KRT8+ and therefore are confounding in evaluating the increase in the KRT8+ cell number. If possible include images showing the highest possible number of alveoli. In addition, as shown by ref.13 in original text, the KRT8+ cells are: 1) SPC+, 2) in close proximity with the alveoli, upon bleomycin treatment; 3) some are lining the alveolar border. On the contrary the images shown in this manuscript are very different, no alveoli are shown and KRT8+ cells are spread all over lung parenchyma. Please select better representative images.

5) In figure 2h the number of SPC+ cells is too high, and doesn't seem to be physiological. In a normal alveolus the area covered by ATII cells is 5% of the total alveolar surface, with an average ratio ATII to ATI of 2:1, furthermore the number of ATII cells in the intraparenchymal region is extremely high. Please be sure that the signal intensity levels are appropriate in order to avoid signal saturation. If this is not the problem, probe different slides and animals to find a more representative image and confirm your finding by staining using an antibody against SPC instead of the genetically encoded reporter gene.

6) To investigate the functional importance of IL11 in alveolar epithelial cells, authors decided to use the Human Pulmonary Alveolar Epithelial Cells (HPAEpiC, please correct the name in the manuscript). First, I don't understand the idea of moving into human cells to test KRT8 positivity that is a peculiar feature of ATII mouse transitioning cells. Second, these cells have very high level of IL11RA that is very different what is reported in Fig1 for human cells and figure 2 for mouse cells. In addition, the staining of SPC, reported in supplementary figure 6A shows a very strange pattern instead of the characteristic perinuclear pattern normally shown by SPC immunostainings in primary mouse and human ATII cells. In

addition, authors claims that recombinant IL11 directly drives cytopathic effect in HPAEpiC cells. Authors should define what they mean for cytopathic effect and select the appropriate markers to measure the phenotype of interest. However, Pathological Extracellular Matrix (ECM) components (Collagen I, fibronectin), EMT related protein (SNAIL) and KRT8 are not standard markers of cellular toxicity and therefore, if authors meant to measure cellular toxicity, they must be implement their data by including at least one of the following markers: 1) apoptosis by (Caspase3/7 assay or TUNEL); 2) cellular viability by ATP intracellular content or equivalent (CelltiterGlow assay).

7) Figure3C please include a figure legend is too difficult to understand that the colors refer to Figure3A.

8) Figure3d, all cellular proliferation studies have not been conducted in appropriate settings. Authors didn't mention for how long cells have been pulsed with Edu, this is fundamental to assess the proliferation rate. In addition, study cellular proliferation in HPAEpiC cells with 10-15% basal proliferation is not supporting any conclusion on ATII cells that, on the contrary, doesn't show a very limited proliferative potential in vitro and are very poorly proliferative in vivo. This set of experiment is misleading and improper. Either remove or reproduce in primary mouse ATII cells. Last, EdU is not a marker of proliferation but a marker for DNA synthesis, please show in the implement with the total cell numbers for each experimental condition, correlate it with EdU positivity and exclude DNA damage by γ -H2AX staining. Staining for other proliferation markers such as Ki-67, phosphor histone H3 or AuroraB kinase. The authors must coinfirm all the experiments reported in figure3, A,B,C, D by using mouse primary alveolar ATII cells.

9) Figure 3F: I suppose that these cells are primary murine ATII cells, isolated from the transgenic mice. It is not well described in the text, please correct. Page 19 line 574, "The FACS sorted cells were then seeded at a density of 2×10^4 cells per well in rat tail collagen (Invitrogen, A1048301) coated 96-well CellCarrier plates (PerkinElmer) and cultured in DMEM supplemented with 10% FBS. The cells were allowed to adhere for 24 hours prior to treatment with mouse IL11 (UniProtKB: P47873, GenScript) or mouse TGF β 1 (R&D Systems,

7666-MB). It is known in the field that ATII cells doesn't survive isolation if cultured in DMEM, specific and properly supplemented medium are needed, in example the commercially available Pneumacult (StemCellTechnology). Secondly, at 5 days post isolation, viable ATII cells, if cultured in collagen coated plastic plates, should be almost completely transdifferentiated in ATI cells, with no SPC positive cells and the majority of RAGE+ with almost no transitioning (double positive) cells. IN figure 3F we can appreciate some PDPLN positive cells, as expected, that are also 100% SPC positive. This is not resembling the normal biology of cultured ATII cells and we invite the author to probe the cells with anti SPC antibody to confirm SPC positivity by IF staining. In addition please include larger field of view for all conditions.

10) Figure 4, authors should include Masson's trichrome staining for the experiment reported in panel B and IF stainings as in B for panel F.

11) In the images reported in Figure4B for the Il11ra1 floxed condition, the SP-C staining is very different from the control condition. In the Il11ra1+/+ condition, SP-C staining appears as expected by staining ATII cells (small cuboidal structure). In the Il11ra1 floxed condition, the SP-C staining is completely lining the alveolus resembling the shape of ATI cells. The authors in the text claims "Instead, we found numerous newly differentiated AT1 cells including regions of completely formed alveoli that were tdT+ in BLM-treated Sftpc-tdT; Il11ra1fl/fl mice (Fig. 4b-d)." (Page 11 line 325). The overlap between Sftpc-tdT and PDPN, in the selected image, is almost 100%, thus suggesting that after Tamoxifen treatment all ATI derives from ATII to ATI differentiation. This appears honestly too effective, thus we suggest to 1) show additional images from other lung regions; 2) Further confirm by SPC/RAGE immunostaining.

12) Experimental timepoints for the in vivo experiments are always different, without and obvious explanation, in example:

- Figure 2a Bleo day 0 and tissue collection at 7-10 and 21 days
- Figure 2g Bleo day 0 and tissue collection at 14 days
- Figure 4a Bleo day 0 and tissue collection at 12 days
- Figure 4e Bleo day 0 and tissue collection at 21 days
- Figure 5a Bleo day 0 and tissue collection at 12 days

It is known that fibrotic remodelling upon bleomycin treatment follows a specific kinetic: Day 0-14 (Inflammation), Day 21 (starts fibrotic scarring), Day 30 (peak of fibrotic remodelling), Day 45 (start of spontaneous resolution), Day 60 (evident resolution). The author must specifically explain why they selected these timepoints, and why these are different in almost every experiment.

13) Figure 4F, the lung from IL11RA1+/+ in uninjured condition shows a clear immune infiltration and this must be commented in the text. In addition authors should provide complete lung images with enlargement of selected portions. It's better if the enlarged portion is always selected in the same region in different experimental conditions.

14) Figure 5b, please provide Masson's trichrome staining and lung hydroxyproline content for all experimental conditions. In addition, authors should provide complete lung images with enlargement of selected portions. It's better if the enlarged portion is always selected in the same region in different experimental conditions.

15) In figure 5b, in the BLM+X203 treated condition, the Sftpc-tdT staining has a completely different pattern compared to what is shown in the bleomycin-treated IL11RA1-/- Figure 4b. Here in figure 5b the pattern is what is normally expected, showing single SPC positive cells with cuboidal shape usually seen in ATII cells. On the contrary in figure 4b the Sftpc-tdT is completely overlapping with PDPN pattern. Here there are two options: 1) the X203 is protective but not so effective in promoting ATII to ATI trans-differentiation and most likely working on lung fibroblast; 2) the ATII to ATI trans-differentiation shown in the figure 4b is highly overestimated. Please explain, select better and more representative images or remove.

16) Page 14 line 382, "Additionally, in IgG treated mice, we found an increase in non-lineage labelled KRT8+ cells (KRT8+ tdT-) that stained weakly for the AT1 marker PDPN (Fig. 5b-d)" We can appreciate an increase in the KRT8+ cells but the PDPN staining in IgG treated condition is even stronger than ctrl condition. This sentence is not supported by the images. . Please explain, select better and more representative images or remove.

17) Figure 5f, the only reasonable conclusion we can take from this image is a reduction in Collagen I deposition upon Bleo+X203 treatment. Other conclusions regarding possible mechanistic links between IL11 and p-ERK pathway should be supported with dedicated biochemical studies. In addition, how could you exclude that X203 is acting on lung fibroblast? Please provide relevant data on this last question.

18) Line 441 page 16 “Furthermore, we found that the anti-proliferative effects of TGF β and its induction of ECM proteins and KRT8 expression in AT2 cells was, in part, mediated by IL11 signaling.” Rephrase according to point 6,7,8

19) Line 457, page 16. “Given the elevated expression of IL11 in aberrant mesenchymal and epithelial cell types in PF and its roles in both fibroblasts activation and AT2 cell dysfunction, we propose that IL11 may cause multiple aspects of pathobiology in different cell types in the diseased niche.” In human or in mouse?

Point by point rebuttal.

Reviewer #1 (Remarks to the Author):

Ng et ask whether AT2 cell IL11 receptor signaling promotes pulmonary fibrosis. Although it is known that IL11 promotes pulmonary fibrosis it has remained unclear if these effects were due to inhibiting AT2 to AT1 transition. The authors here nicely leverage existing human PF scRNA data sets and several robust experimental models to show that IL11R signaling stalls AT2 differentiation and promotes the expansion of KRT8+ transitional cells with pronounced EMT expression profiles. This is a very well written manuscript with clearly described experimental designs and findings that sheds light into poorly understood mechanisms of IL11-mediated fibrosis. I have very few concerns:

1. Define IRE1 alpha as inositol-requiring transmembrane kinase/endoribonuclease 1a

Author response:

We thank the reviewer for this comment and have now amended this definition accordingly.

2. The rarity of the IL11+EGFP+ EpCAM+ cells in BLM suggested by the FACS data in Fig. 2A seems to contradict the prolific KRT8+ GFP immunostaining shown in Fig. 2C. It is well known that measuring EGFP expression fluorescence is challenging with highly autofluorescent lung epithelial cells. Given that KRT8 (Mab clone 3G9) and EGFP (clone EPR14104) antibodies also work in FACS assays it would inspire confidence in the data as well as provide critical quantitative assessments of IL-11+KRT8+ transitional cells during disease progression if FACS analysis were conducted with anti-EGFP staining on a EpCAM+ KRT8+ and KRT8- gate.

Author response:

We acknowledge the reviewers' concerns on the quantification of endogenous GFP expression in lung cells from our *IL11^{EGFP}* reporter mice. To further complement our existing IF-based histological analyses, we have now performed additional flow cytometry experiments on lung cells with intracellular staining for anti-GFP (clone EPR14104; Abcam ab303588).

Firstly, we tested both anti-KRT8 (clone 3G9) and anti-KRT8 (clone TROMA-1) antibodies for flow cytometry compatibility of lung epithelial cells from BLM-injured mice. However, we were not able to distinguish between populations of KRT8-hi or KRT8-low cells in both injured and uninjured lungs using these 2 anti-KRT8 antibodies.

We next stained cells for Cldn4, an alternative established marker for transitional epithelial cells¹, and by using flow cytometry gating methods previously described by Choi et al.¹ managed to discern distinct populations of Cldn4^{hi} from Cldn4^{low} EpCAM+ epithelial cells, with the former population found to be robustly increased following BLM-injury (~12.9% in BLM group vs. ~0.25% in the uninjured group) (**Fig. R1.1**). As expected, GFP staining was barely detected in uninjured lung EpCAM+ epithelial cells and there was a small increase in GFP+ EpCAM+ cells in BLM-injured mice (**Fig. R1.2**). In contrast, there was a significant increase in GFP+ EpCAM+ Cldn4^{hi} cells (~6.73% in BLM group vs. 0% in the uninjured group) following BLM-injury (**Fig. R1.1**). Furthermore, there was a small but significant increase (~0.73% in BLM group vs. ~0% in the uninjured group) in the proportion of GFP+ EpCAM+ Cldn4^{low} cells following BLM-injury, indicating that the majority of IL11-expressing epithelial cells in the injured lung are Cldn4^{hi} transitional epithelial cells. These

new data which are incorporated into **Fig. 2g** and **Supplementary Fig 8** provides further evidence for the emergence of IL11+ transitional epithelial cells in the injured lung and we thank the reviewer for the helpful suggestions.

Fig. R1.1 Flow cytometry analysis of GFP expressing Cldn4^{hi} and Cldn4^{low} epithelial cells by gating for Cldn4 and GFP in EpCAM⁺ lung epithelial cells from IL11^{EGFP} reporter mice 10 days post-BLM injury.

Fig. R1.2 Flow cytometry analysis of GFP expressing epithelial cells by gating for GFP in EpCAM⁺ lung epithelial cells from IL11^{EGFP} reporter mice 10 days post-BLM injury.

3. The differences in KRT8 expression are not very convincing in Fig 3f weakening the argument that AT1 differentiation is stalled by IL11 in the 2D culture system. A quantitative assessment of KRT8 expression would provide more evidence for the effects of IL11 on inhibiting AT1 terminal differentiation.

Author response:

We thank the reviewer for this comment and have now assessed the expression of both KRT8 and PDPN expression in our AT2 cultures. In **Fig. R1.3** we plot the normalized expression of KRT8 and PDPN over cell area. Overall, our analysis confirms that IL11 and TGF-beta treatments induced high levels of KRT8 which delayed AT1 maturation by suppressing PDPN expression. We now hope that this now clarifies the reviewer's concerns.

Fig. R1.3. Violin plots showing the quantification of cell area, KRT8 or PDPN immunostaining in 2D cultures of AT2 cells treated with IL11, TGFβ1+IgG or TGFβ1+X203. Red line indicates median values.

4. A schematic providing the relationship between IL11 TGFβ AT2 to KRT8+ transitional or AT1 differentiation would help better highlight the author's findings.

Author response:

We thank the reviewer for this helpful suggestion. We now illustrate the effects of IL11 on cells in the fibrotic alveolar niche in a new summary diagram (**Fig R1.4** and in revised **Supplementary Fig. 23**).

[editorial note: figure redacted]

Fig. R1.4. *Schematic illustrating the proposed effects of IL11 in the healthy or fibrotic alveolar niche. During alveolar regeneration, AT2 cells differentiate into AT1 cells via a Krt8+ transitional cell state. However, in the context of persistent pathology, aberrant ECM-producing Krt8+ transitional cells with defective AT1 differentiation capacity accumulate in the lung and the factors that regulate the differentiation and maintenance of these aberrant cells are poorly understood. In the diseased lung, IL11 is upregulated by activated AT2 cells, Krt8+ transitional cells and pathological fibroblasts. In AT2 cells, IL11 stimulates ERK signaling pathway dependent pro-EMT programs that maintain AT2 cells in a dysfunctional KRT8+ state which impairs alveolar epithelial regeneration. IL11 induces the expression of pathologic ECM proteins (such as Collagen I and CTGF) by KRT8+ cells which may directly promote aberrant lung remodeling. Additionally, the elevated expression of IL11 by pathological fibroblasts in the fibrotic niche may contribute to aberrant epithelial cell differentiation in lung fibrosis.*

Reviewer #2 (Remarks to the Author):

In the alveoli, alveolar type 2 (AT2) cells play a crucial role in maintaining lung homeostasis and facilitating regeneration following injury by proliferating and differentiating into new alveolar type 1 (AT1) cells specialized for gas exchange. Given the vital role of AT2 cells, their self-renewal and differentiation must be carefully orchestrated to uphold tissue integrity and promote efficient repair. Recent studies suggest that Krt8+ AT2+ lineage populations emerge during tissue repair following lung injury. However, chronic inflammation impedes AT1 differentiation and results in the accumulation of Krt8+ cells and impaired alveolar regeneration. IL-11, a member of the IL-6 family of cytokines, binds to IL-11Ra1 and transmits the signal through a protein called glycoprotein 130 (gp130). The authors previously reported that Il11ra1-deficient mice are protected from fibrosis in the bleomycin (BLM) mouse model of pulmonary fibrosis (PF). Treatment with an anti-IL-11 antibody they made also inhibited inflammation and fibrosis in the same bleomycin-induced PF mouse model (Ng et al., Sci. Transl. Med, 2019). Furthermore, the authors reported that conditional deletion of Il11ra1 using Col1a2-CreERT mice led to a significant inhibition of fibrosis (Ng et al., FASEB J, 2020).

In this study, Ng et al. aimed to demonstrate the role of IL-11 signaling in the KRT8+ cells (the murine counterpart of human KRT5- KRT17+ cells) during BLM-induced PF models. The authors first demonstrated that IL-11 was expressed in AT2 and KRT5-KRT17+ cells using human single-cell-RNA-seq data and in KRT8+ cells in BLM-treated IL-11 reporter mice. The authors stimulated 2D-cultured human and mouse alveolar epithelial cells with IL-11 and demonstrated that IL-11 increased KRT8 expression and EMT-related genes. In addition, the authors generated conditional deletion of Il11ra1 using SfpC-CreER mice. They showed that the deletion of Il11ra1 in the lineage of AT2 cells reduced numbers of KRT8+ cells, thereby attenuating lung fibrosis and promoting AT1 cell differentiation. Although the presented results provide further insight into the mechanisms underlying the IL-11-dependent promotion of lung fibrosis, several concerns must be addressed. The following are specific comments.

Major comments:

1. The authors' previous study found that deletion of Il11ra1 in Col1a2+ cells (fibroblasts) reduced BLM-induced lung fibrosis. However, this study proposes a crucial role for IL-11ra1 in AT2 cell-derived lineage cells using SfpC-CreERT2 mice. The authors need to discuss and reconcile these seemingly inconsistent findings, specifically addressing whether fibroblasts and AT2 cell-derived cells contribute to exacerbating lung fibrosis and if AT2 or KRT8+ cells express Col1a2. Additionally, scRNA-seq analysis of human PF patients suggests that the primary source of IL-11 may be fibroblasts or myofibroblasts, with lower expression in KRT17+ cells and AT2 cells. The authors should address the cellular origin of IL-11 and its relative contribution to promoting fibrosis in PF patients or BLM-induced lung fibrosis model.

Author response:

We firstly like to thank the reviewer for his/her helpful suggestions and insightful comments, which allowed us the opportunity to further fine tune our manuscript. We acknowledge the reviewers' interests regarding the relative contributions of various IL11-expressing cell types to fibrosis. However, since IL11 is secreted and can potentially act on various IL11RA-expressing cell types via auto/paracrine, we have instead chosen to focus solely on IL11-responsive cells such as lung and alveolar epithelial cells for this study. We have specifically mentioned this as limitations to the study:

“We did not dissect the specific cell type expressing IL11 that impacts AT2-to-AT1 differentiation and fibrosis, although our earlier studies suggest a dominant role for IL11 secretion from fibroblasts for fibrosis phenotypes.”

Nonetheless, to build on these, we have now added the expression profile of *IL11RA* in the human lung for our response to the reviewers' comments in **point 2** below. Furthermore, we have also included new data from a new mouse model, which only became ready to us much later during the study's period, in which we deleted IL11 specifically in AT2 and AT2-lineage cells (*Sftpc-CreER; Il11-flox* mice) to test whether IL11-expression by AT2-lineage cells alone is sufficient for lung fibrosis progression (**Fig R2.1** and in revised **Supplementary Fig 18**). Our new data on *Sftpc-CreER; Il11-flox* mice showed that mice with IL11-deletion in AT2-lineage cells were not protected from lung fibrosis as indicated by histopathological scoring of Masson's trichrome staining and lung hydroxyproline assay. These new findings indicate that IL11-expression by AT2 lineage cells may not be the most dominant/prominent source of IL11 that drives lung fibrosis development after injury. Future in vivo studies on mice with fibroblast-specific IL11-deletion may reveal the importance of IL11-expression in fibroblasts for lung fibrogenesis.

Fig R2.1. (a) Schematic showing the period of tamoxifen (Tmx) administration and the induction of lung fibrosis in $Sftpc-CreER; Il11^{fl/fl}$ or $Il11^{+/+}$ mice via oropharyngeal injection of bleomycin (BLM). (b) Percentage body weight change post-BLM injury (day 21 versus day 0), (c) lung weight to body weight indices and (d) survival analysis of tamoxifen-treated $Sftpc-CreER; Il11^{fl/fl}$ or $Il11^{+/+}$ mice 21 days post-BLM injury. (e) Representative Images of Masson's trichrome staining, (f) lung histopathological fibrosis scoring and (g) lung hydroxyproline content in right lungs of tamoxifen-treated $Sftpc-CreER; Il11^{fl/fl}$ or $Il11^{+/+}$ mice 21 days post-BLM injury. Scale bars in e: 1000 μ m (top panel) 100 μ m (bottom panel). Data are represented as mean \pm s.d. P values were determined by one-way ANOVA (Tukey's test) in panel b, c, f and by Log-rank (Mantel-Cox) test in panel d.

The reviewer also correctly highlights the similarities in the phenotypes presented by the present model ($Sftpc-creER; Il11ral-flox$ mice) with our previous model ($Colla2-CreER; Il11ral-flox$ mice) in that both strains share similar protection against lung fibrosis. In view of very recent evidence from scRNA-seq studies of fibrotic human and mouse lungs that describe aberrant epithelial cells as having a profibrotic phenotype with high expression levels of mesenchymal genes (such as *COL1A1*, *FNI*, *CTGF*). It remains unclear whether aberrant epithelial cells can express other mesenchymal-related genes such as *COL1A2* in the context of lung fibrosis.

To address this, we firstly performed a survey of recent human PF and mouse lung scRNA-seq datasets but did not find a clear signature of differential *COL1A2* expression by epithelial cell types. We then performed protein staining for *COL1A2* (using polyclonal anti-*COL1A2*

antibody; Thermo Fisher, PA5-106555) and KRT8 in the lungs of *Sftpc-tdT* mice 12 days post-BLM injury during the early fibrotic phase. Interestingly, and in contrast to scRNA-seq data, we found that BLM injury caused an accumulation of COL1A2-expressing AT2-lineage traced cells (COL1A2⁺ KRT8⁺ *tdT*⁺ and COL1A2⁺ *tdT*⁺ cells) in regions of injury (**Fig. R2.2**), which indicates that activated AT2 cells / transitional AT2 cells can upregulate COL1A2 protein after BLM injury. Hence, in light of these findings, it is possible that the prolonged tamoxifen regime (where tamoxifen was administered before and after BLM treatment) and the resultant phenotypes observed in *Colla2-CreER; Il11ral-flox* mice in our previous study may in part be due to the additional and unintended deletion of *Il11ral* in transient COL1A2-expressing transitional AT2 cells after BLM-injury.

Nonetheless, given the data from the present study which suggests that IL11-signaling in AT2-lineage cells is required for lung fibrosis progression, our present findings further suggests that IL11-driven AT2 cell dysfunction likely precedes and promotes the activation of mesenchymal cells during bleomycin-induced lung injury.

Fig. R2.2. Images of COL1A2 immunostaining in the lungs of *Sftpc-tdT* mice after BLM-injury. Yellow arrows indicate COL1A2⁺KRT8⁺*tdT*⁺ cells; blue arrows indicate COL1A2⁺*tdT*⁺ cells.

2. In Figure 1 and Supplementary Figure 1, to gain a comprehensive understanding of PF patients' lungs, the authors should include all cell types, including fibroblasts and immune cells, in their UMAP analysis in Supplementary Figure 1. Additionally, the authors should include the expression of the *Il11ral* gene in Figures 1a and b.

Author response:

We thank the reviewer for this comment and have now supplemented the original violin plots in **supplementary Fig 1** with UMAP and dot plots showing *IL11* and *IL11RA* expression across all cell types in human lung datasets (**Fig R2.3** and **R2.4** below and revised **Supplementary Fig 2** and **3**). Our new plots highlight that *IL11* expression is specific to pathological fibroblast populations, KRT5⁻/KRT17⁺ and aberrant basaloid cells, whereas *IL11RA* is more highly expressed by stromal, endothelial and epithelial cell types and less so by immune populations in the human lung. We believe that these new plots provide a more comprehensive overview of *IL11* and the *IL11RA*-expression cells that can potentially respond to IL11 in the human lung.

Fig R2.3. UMAP visualization of *IL11* or *L11RA* expressing single cells in scRNA-seq data from control and PF samples in the (a-c) Habermann et. al. (GSE135893) and (d-f) Adams et. al. (GSE136831) datasets.

Fig R2.4. *IL11* and *IL11RA* expression in various cell types in the human lung. Dot-plot showing the expression of *IL11RA* in individual cell types in scRNA-seq data from control and PF samples in the Habermann et. al. (GSE135893) and Adams et. al. (GSE136831) datasets.

3. In Figure 1g-h, to improve transparency, the authors should provide a full list of genes in the IL11 co-expression module as Supplementary Data.

Author response:

We have now provided a new table in **Supplementary Data 3** which lists the genes that are co-expressed with IL11 in aberrant epithelial cells in Adams et al and Habermann et al datasets. We hope this now improves the transparency of our analysis.

4. In Figure 1i, the authors should provide a clearer explanation for the density plot and specify the purpose of the y-axis.

Author response:

We acknowledge the ambiguity in the description of **Fig 1i**. In this analysis, we calculated the spearman correlation between the gene expression of IL11 and IL11-coexpressing genes in Transitional AT2 and KRT5-/KR517+ cells, separately for control and ILD sample using the Habermann dataset. We then plotted the distribution of these spearman correlations (salmon color for the distribution for the correlations calculated in control cells, and turquoise for ILD).

We have now changed the y-axis and caption of the plot to below and hope they are more intuitive and reflective of the computational analysis performed.

“ y-axis: Density of spearman correlation”

5. In Figure 2h, the authors should identify the cells that are Sftpc-negative KRT8-positive.

Author response:

We appreciate the reviewers' interest in the possible identities of Sftpc- KRT8+ cells in the injured lung. Firstly, KRT8 is expressed by airway cells, and is also a marker for transitional alveolar epithelial cells across a diverse range of lung injury models in mice. The reviewer might also be aware that recent studies have highlighted the capacity of non-AT2-lineage cells such as airway secretory cells, Sox2+ airway stem cells and MHC-II+ club can regenerate the epithelium by differentiating into AT1 cells via a Krt8+ transitional state ^{2,3}.

Although it may not be apparent from our images in **Fig 2h**, we wish to highlight that IL11+ SFTPC- KRT8+ cells were very rarely observed after injury. This suggests that non-AT2 derived KRT8+ cells may not be a prominent source of IL11 in the injured alveolar epithelium. Hence, we have not made attempts to investigate non-SFTPC expressing epithelial cells in this study.

6. In Figure 3b, the authors analyzed the expression of EMT-related molecules in human pulmonary alveolar epithelial cells following stimulation with IL-11, TGFb1+IgG, or TGFb1+X203. Of note, the morphology of cells appeared to be different in cells stained with anti-Fibronectin, anti-Collagen I, or anti-SNAIL antibodies despite the same treatment. The authors need to clarify these points. Alternatively, the authors should change to representative images. Moreover, SNAIL is a transcriptional factor that should be localized in the nucleus. However, the data suggests that SNAIL is predominantly localized in the cytoplasm. The authors need to explain this discrepancy.

Author response:

We acknowledge that the staining patterns observed for Collagen I, fibronectin and KRT8 may have given the impression that cell morphology was altered following cytokine treatments. To better highlight cellular morphology in our in vitro phenotypic assays, we now include new images of Phalloidin counterstaining (**Fig. R2.5** and in revised **Fig. 3**), which shows that the cell morphologies of HPAEpiCs were not dramatically altered across all treatment groups (IL11/TGF-beta + IgG/X203). We hope that this clarifies the reviewer's concerns.

Fig R2.5. Image of immunostaining for KRT8, Fibronectin or Collagen I in HPAEpiC. Cells were costained with phalloidin to visualize cellular morphology.

We also acknowledge that the pattern of cytoplasmic SNAIL expression shown in original **Fig 3b** was somewhat unexpected. To rule out potential anomalies, we performed additional verification of the originally described anti-SNAIL (PA5-85493) alongside two separate anti-SNAIL (MA5-14801) and anti-SNAIL/SLUG antibodies (ab180714). However, both of the alternative antibodies failed to pick up any form of SNAIL expression in HPAEpiC following IL11 treatment (**Fig R2.6**). In light of these discrepancies, and the possibility of unspecific staining with PA5-85493, we have decided to exclude all existing SNAIL data from the revised manuscript. This change however, does not impact the overall message of **Fig 3**, as we have now also provided additional evidence for the role of IL11 in causing EMT-like features in additional primary epithelial cell types, which can be found in our response to **point 12** below. We thank the reviewer for highlighting this issue to us.

Fig R2.6. Images of immunostaining for SNAIL or SNAIL/SLUG (green) in HPAEpiC treated with IL11 (5 ng/ml; 24 hours). Nuclei stained with DAPI (blue).

7. In Figure 3d, the authors should provide the image data to create this graph.

Author response:

We apologize for not incorporating EdU staining images in the initial submission, which we have now done so below (revised **Supplementary Fig 9**).

Fig R2.7. Images of EdU staining in HPAEpiC treated with IL11, TGFβ1, X203 or IgG control antibodies. Scale bars: 100 μm.

8. To evaluate the changes in cell morphology more accurately, the authors should provide images of the cells on day 1 and day 3 together in Figure 3f.

Author response:

We acknowledge the reviewer's comment and have performed additional cultures and now provided new images of cells throughout the 5 days differentiation period (for day 1, 3 and 5) (**Fig R2.8**). These images show that untreated AT2 cells begin to enlarge/flatten out and differentiate into PDPN expressing AT1-like cells by day 3. Both IL11 and TGF-beta induce high and sustained levels of KRT8 by day 3, which resultantly stalls the differentiation of AT1 cells. We have incorporated these images into a new **Supplementary Fig 12**.

Fig R2.8. Time series (day 1, 3 and 5) of cultures of *Sftpc-tdT*⁺ AT2 cells treated with IL11 (5 ng/ml), TGFβ1 (5 ng/ml), X203 or IgG control antibodies (2 μg/ml). *tdT*⁺ cells (red) were immunostained for KRT8 (green), PDPN (white) and counterstained with DAPI (blue). Scale bars: 50 μm.

9. In Figure 4, Western blotting analysis of KRT8, PDPN, and *tdT* using lung tissue homogenates before and after BLM would be informative.

Author response:

Due to limited availability of lung tissues from our *Sftpc-tdT*; *Il11ra1*^{fl/fl} model, we have instead performed western blots for KRT8 and an AT1 marker (AGER/RAGE) on lung homogenates from our complementary *Sftpc-CreER*; *Il11ra1*^{fl/fl} mouse model. This showed an expected reduction in AGER (~43 kDa) and a corresponding increase in KRT8 (~55 kDa) expression in control mice lungs after BLM injury as compared to uninjured lungs. In contrast, AGER expression was partially restored, whereas there was a non-significant trend of reduced KRT8 expression in the lungs of mice with AT2-specific *Il11ra1* deletion after BLM injury (**Fig R2.9**). In general, these results are in line with the epithelial phenotypes presented for this model and we have now incorporated these new data in revised **Supplementary Fig 14**.

Fig R2.9. Western blot and densitometry analysis of AGER and KRT8 expression in lung homogenates from *Sftpc-CreER; Il11ra1^{fl/fl}* or *Il11ra1^{+/+}* mice 12 days post-BLM injury. $n = 3$ mice / group. KRT8 (~55 kDa band) and AGER protein (~43 kDa band) were quantified by densitometry. Dotted lines indicate mean values of uninjured controls.

10. In Figure 5, to calculate cell numbers more quantitatively, the authors should measure them by flow cytometry but not count them by immunohistochemistry, as the cell margin among cells seems obscure.

Author response:

We acknowledge the reviewer's concerns about the quantification of transitional cells by histology in Fig 5. We have included additional flow cytometry data to provide unbiased quantification of lung transitional cells in our anti-IL11 treatment model. We subjected single cell suspensions from the lungs of uninjured or BLM+IgG/X203 treated *Sftpc-tdT* mice (at a similar time point 12 days post-BLM) to flow analysis and monitored for the proportions of lineage traced AT2 cells (EpCAM+ *tdT*+), transitional cells (EpCAM+ *tdT*+ Cldn4^{hi}) along with lineage traced AT2 cells that have acquired the AT1 marker PDPN expression (EpCAM+ *tdT*+ PDPN+). Overall, these new flow cytometry data were consistent with our histological findings which confirmed the decrease in lineage-traced *tdT*+ cells and a corresponding increase in the proportion of Cldn4^{hi} transitional AT2 cells in the injured lungs of IgG treated mice (**Fig R2.10**). X203 treatment partially restored/preserved AT2 cells after injury and that this was linked to a reduction in lineage traced Cldn4^{hi} transitional cells and a significantly increased proportion of newly derived AT1 cells (EpCAM+ *tdT*+ PDPN+) as compared to IgG treatment. We have now incorporated these new data into **Fig. 5** and **Supplementary Fig. 20**.

Fig R2.10. Flow cytometry analysis of CD45⁻ CD31⁻ EpCAM⁺ tdT⁺ ; Cldn4⁺ or PDPN⁺ lung cells from single cell suspensions from uninjured or BLM-challenged *Sftpc-tdT* mice treated with either IgG or X203. *n* = 3 mice / group.

11. In Supplementary Figure 5c, according to the Materials and Methods section, anti-GFP and anti-IL-11 antibodies were raised by rabbit, and co-immunostaining is technically impossible using these antibodies. The same applies to Supplementary 6g, since anti-ERK and anti-IL-11 antibodies were raised in rabbit.

Author response:

We apologize for not including the catalog information for the specific anti-p-ERK antibodies used with rabbit polyclonal anti-IL11 in **Supplementary Fig 5c**. For this particular stain, we used species compatible mouse monoclonal anti-p-ERK (D1H6G clone, #5726). We thank the reviewer for spotting this error and we have now corrected the methods accordingly.

12. In Supplementary Figure 6a, gp130 and IL11RA are membrane proteins. However, the positive signals are not observed in the cell membrane but are also in the nucleus, suggesting that the specificity of these antibodies is questionable. The authors should use other antibodies that can specifically detect these molecules or verify their specificity using

knockdown cells of these genes. Negative signals using IgG control antibody does not ensure the specificity of these antibodies.

Author response:

We acknowledge the reviewer's concerns on the expression of gp130 and IL11RA in HPAEpiCs, which may have appeared to be partially localized in the nucleus at the low magnification (10X) shown in the original supplemental figure.

For the benefit of the reviewer, we have now included higher magnification images (20X) of (non-permeabilized) HPAEpiCs stained for membrane gp130 / IL11RA using two different antibodies each. Consistent results were obtained with these antibodies showing the abundance of membrane-only staining for these proteins (**Fig R2.11**). We have included the images in the top row into revised **Supplementary Fig 9**.

Fig R2.11. Images of immunostaining for gp130 or IL11RA in HPAEpiC using the indicated antibodies.

13. In Supplementary Figure 6c, to understand the whole picture of the genes induced by TGFb-induced IL-11 and IL-11-induced ERK pathway, the authors should perform RNA-seq analysis of cells stimulated with TGFb + IgG, TGFb+X203, and IL-11+U0126. Then the authors compared gene expression profiles between TGFb1+IgG vs. TGFb1+X203 and IL-11 vs. IL-11+U0126. Moreover, the authors should explain Log2 fold change, and Log2 mean expression in detail for readers unfamiliar with RNA-seq analyses.

Author response:

The reviewer makes an interesting request here, one that we do not think is needed or indeed achievable due to several inherent limitations which we describe here.

Firstly, to clarify, the primary focus of our bulk RNA-seq experiment was to investigate direct transcriptomic effects of IL11 on lung alveolar epithelial cells. Our results highlight that IL11 treatment alone did not induce any significant genome-wide RNA changes despite strong and robust induction of EMT-related proteins (Fibronectin/Collagen I) and also KRT8 in HPAEpiC, which also appear to be dependent on ERK-signaling. We were unsurprised by these findings as we have now repeatedly observed similar effects, translationally-driven and protein-level specific of IL11 across several different primary human cells (including lung fibroblasts, cardiac fibroblasts, hepatic stellate cells and vascular smooth muscle cells).

Secondly, acquiring large numbers of hAECs for further extensive RNA studies is extremely challenging as the proliferation rates of primary hAECs in cultures are extremely low and that these cells cannot be expanded/passaged without them losing their AT2-like characteristics. To circumvent these limitations we only performed short term (24hrs) stimulation studies on small numbers of non-passaged cells (~1e5 to 1e6 cells/donor) in our 96 well plate imaging platform. As a consequence, we regretfully inform the reviewer that for these reasons, we are not able to perform the suggested RNA-seq experiments for the revision.

Instead of the experiment suggested above, we wish to highlight to the reviewer that we have now completed and incorporated two additional scRNA-seq experiments (for this revision) on enriched epithelial cell populations from: 1) *Sftpc-CreER; Il11ra1^{fl/fl}* mice and 2) from X203/IgG treated mice after BLM injury, to further uncover mechanisms by which IL11-signaling in AT2-lineage cells promotes fibrosis (**Fig. 4e-i, Fig. 5g-i and Supplementary Fig. 15, 16 and 22**). Overall, our new computational analyses clearly supports this concept and shows that IL11-signaling in AT2-derived Krt8⁺ transitional cells is strongly associated with aberrant transcriptomic signatures of multiple disease-related pathways (such as EMT, TGF-beta signaling, p53, inflammation and unfolded protein response pathways), with genes related to EMT (such as *Colla1*, *Fnl1* and *Ccn2*) amongst the most downregulated from abrogated IL11-signaling. Additionally, IF validation of ECM and profibrotic related proteins Collagen I and CTGF, showed that abrogated IL11-signaling strongly attenuates pathologic ECM and stress related responses of Krt8⁺ transitional cells. These key findings demonstrate that IL11 signaling likely promotes the profibrotic phenotypes of Krt8⁺ transitional cells and that these cells may also directly contribute to the aberrant accumulation of ECM during lung fibrosis.

To address the reviewer's comment on log₂ fold change: the log₂ fold change of each gene refers to the ratio of expression between treatment and control, scaled by log₂-transformation. For instance, a log₂ FC of 1 for a particular gene would equate to a 2 fold increase in gene expression of that particular gene.

14. In Supplementary Figure 6e, although the authors discussed the post-translational regulation of EMT-related genes by IL-11, the quality of the imaging data is not sufficient. The authors should demonstrate whether IL-11 regulates the EMT-related proteins through post-translational regulation by Western blotting.

Author response:

We acknowledge the reviewer's concerns. However, due to limitations in acquiring sufficient quantities of HPAEpiCs as explained above, we were not able to acquire sufficient proteins for signaling studies. We hope the reviewer understands these limitations.

We have instead performed additional experiments on primary human small airway epithelial cells (HSAEC) and on primary AT2 cells isolated from *Sftpc-tdT* mice to complement our existing in vitro data.

Firstly, and similar to our previous HPAEpiC data, we show that HSAEC express high levels of IL11RA protein by immunostaining and that TGF-beta stimulation strongly induces IL11 secretion by ELISA analysis. IL11-treatment similarly induced EMT-related fibronectin and Collagen I expression by HSAEC and Collagen I expression by mouse AT2 cells as assessed by immunostaining and sirius red quantification of soluble collagen secretion. Likewise, the

effects of IL11 on the expression of these profibrotic proteins and collagen secretion were dependent on ERK-signaling and can be blocked by the MEK-inhibitor U0126 (**Fig R2.12** and **R2.13**).

We believe that these new data now provide a deeper understanding of IL11-driven pathological EMT-related processes across distal airway and alveolar epithelial cells and incorporate these new data into revised **Fig 3** and **Supplementary Fig 11**.

Fig R2.12. (a-d) Representative images of immunostaining of Fibronectin and Collagen I (green), in primary human small airway epithelial cells (HSAEC) treated as indicated for 24 hours. Scale bars: 100 μm . (e) Secreted collagen concentration was determined by sirius red assay.

Fig R2.13. Representative images of immunostaining of Collagen I (green), in *Sftpc-tdT*⁺ AT2 cells from *Sftpc-tdT* mice treated with IL11 (5 ng/ml) in the presence/absence of MEK inhibitor U0126 (10 μ M) for 48 hours. Scale bars: 50 μ m.

15. In Supplementary Figure 6f, given that U0126 attenuated the expression of EMT-related genes, it would be interesting to test whether a MEK inhibitor, such as Trametinib, attenuates BLM-induced fibrosis. This experiment would further strengthen the importance of the present study.

Author response:

The reviewer makes an interesting request here regarding the therapeutic potential of MEK inhibitors for the prevention of BLM-induced fibrosis. However, such studies have already been performed by others using MEK inhibitor PD98059⁴. The authors showed that MEK-inhibition was sufficient to reduce both inflammation and fibrosis following BLM injury. Importantly, and in line with these findings, we have shown that the pharmacological/genetic inhibition of IL11-signaling greatly diminishes pathological ERK and SMAD signaling in the lungs and protects against BLM-induced fibrosis^{5,6}. In further support of this, another recent study further demonstrated that inhaled siRNA nanoparticles that target IL11 had similar effects by attenuating lung fibrosis, diminished lung ERK and SMAD signaling and also improved lung function in the BLM model^{5,7}.

We have previously shown the fundamental importance of IL11-ERK signaling for lung fibroblast activation. Equally important, we have now demonstrated that p-ERK⁺ IL11⁺ Krt8⁺ cells appear to accumulate in regions of lung injury (**Fig. S10e,f**), and that the occurrence of these p-ERK⁺ transitional cells are reduced following X203 treatment (**Fig. S21a**). These highlight the importance of IL11-ERK signaling in promoting the acquisition/maintenance of Krt8⁺ cells in the injured lung.

Given the existing evidence described here, we hope that the reviewer understands that additional testing of MEK-inhibitors would detract from our current focus and be better off performed as separate studies.

Minor comments:

1. In Supplementary Figure 7b, gating strategy of AT cells was not described.

Author response:

We now include our FACS isolation strategy of AT2 cells based on gating for CD45⁺ CD31⁻ or CD45⁻ CD31⁻ EpCAM⁺ MHCII⁺ lung cells from *Sftpc-CreER*; *Il11ra1^{fl/fl}* mice (**Fig R2.14** and **Supplementary Fig 13a**). We thank the reviewer for highlighting this to us.

Fig. R2.14. Gating for FACS sorting of CD45⁺ CD31⁻ or CD45⁻ CD31⁻ EpCAM⁺ MHCII⁺ lung cells from *Sftpc-CreER*; *Il11ra1^{fl/fl}* mice.

Reviewer #3 (Remarks to the Author):

The study by Benjamin Ng et Al titled “Interleukin-11 causes alveolar type 2 cell dysfunction and prevents alveolar regeneration” follows up a previous study from the same group on the role of IL11 in the pathogenesis of Idiopathic Pulmonary Fibrosis (Benjamin Ng et Al, Sci Transl Med. 2019). In the Sci Transl Med. 2019 paper, Benjamin Ng and colleagues proposed a model in which IPF lung fibroblasts secretes IL-11 and IL-11 promotes fibroblast to myofibroblast activation, thus generating a profibrotic positive feedback that is fundamental for fibrosis development in the bleomycin-mouse.

In the current study, Benjamin Ng and colleagues, shifts the focus on ATII cells. This study aims at defining the role of IL11 in alveolar regeneration with a specific focus on the role of IL11 in regulating ATII to ATI trans-differentiation a known and physiological process of alveolar re-epithelialization. It is known that KRT5-/KRT17+ epithelial cells are disease-associated transitional AT2 cells, and these cells accumulate in the lungs of patients with IPF as previously shown, ref 6 and 7 in original manuscript. These KRT5-/KRT17+ cells were shown to have a high similarity with the mouse KRT8+ cells enriched in the mouse lung after injury, as shown in ref 13-14-15 in original manuscript.

The topic of ATII to ATI trans-differentiation is, in this moment, very relevant in the field of lung regeneration and any findings in this context could represent a consistent advance in better understanding the pathophysiology of IPF disease and eventually pave the way to the development of new and more effective therapies. However, I am frankly concerned by the overall structure and consistency of this paper. Here below it will follow a detailed list of major comments that authors should address to adequately support their findings.

Major comments

1) The study is based on the re-analysis of large-scale scRNA-seq data of lung cells from patients with pulmonary fibrosis from two independent studies by Habermann et. al., and Adams et. al. (GSE135893 and GSE136831 respectively). The authors claims that the IL11-expressing aberrant ATII cells are specifically enriched in IPF patients vs Control and this is well supported by the data shown in Figure1A-C and Supplementary Figure1. Afterwards, the authors tried to parallel the human findings in a mouse model of acute lung damage (bleomycin mouse model). Obviously, this is a critical point of the paper, since all the other experiments are conducted in mouse models of bleomycin-induced lung damage. The authors claims that “after bleomycin injury the proportions of IL11EGFP+ cells in hematopoietic (CD45+ CD31-; P=.0136), epithelial (CD45- CD31- EpCAM+; P=.0002) and stromal cell populations (CD45- CD31- EpCAM-; P=.0200) were found increased. These data are reported in Figure2A, B and supplementary Figure4”. The shifts of IL11EGFP+ cells in the bleomycin treated condition, shown in the representative flow cytometry analysis chart, is really minimal, with very few IL11EGFP+ in the epithelial population (0.150%). On the contrary in figure 2c they are showing IFs with a remarkable number of IL11+ cells, in particular at 7 days post bleomycin treatment.

Could authors please comment on this?

Could authors please confirm their findings by showing an IF experiment using an antibody targeting IL-11 instead of the genetically encoded reporter gene?

Author response:

Firstly, we thank the reviewer for his/her extensive review of our work and highly constructive comments, which we have tried to address in this revision.

As the reviewer has rightfully picked up here, the detection of lung epithelial cells with endogenous IL11-EGFP expression in injured IL11-EGFP lungs by flow cytometry were very rare events. In keeping with this, we found that the extremely low proportions of IL11+ cells in lungs of injured IL11-EGFP mice are surprisingly consistent with those shown in human scRNAseq data. In the Habermann dataset, *IL11*+ cells were detected in 0.17% of epithelial cells and 3.3% of fibroblasts. Comparatively, *IL11*+ cells were also observed at equally low proportions in 0.65% of epithelial cells and 1.8% of fibroblasts from the Adams dataset. These studies not only show that IL11 expressing cells are indeed very rare but may also reflect on the challenges involved in isolating these cells from the both mice/human lungs by conventional tissue dissociation methods. Hence, our cytometry findings in the fibrotic mice lungs are somewhat consistent and perhaps expected.

To further supplement our current data, we have performed additional flow cytometry experiments on a separate cohort of IL11-EGFP mice (similar 10 days post-BLM time point) using anti-GFP antibodies (as requested by another Reviewer) and we found a similarly small but significantly increased proportion of GFP+ EpCAM+ cells following BLM-injury (BLM: 0.73% vs uninjured: 0%) (**Fig R3.1** and revised **Supplementary Fig. 8**). Importantly, GFP+ EpCAM+ cells were absent in uninjured lungs, again indicating that IL11+ expression is only specifically upregulated in response to alveolar epithelial injury. These new data provide additional confirmation for the rare but significantly increased proportion of IL11+ epithelial cells in the fibrotic lung.

Fig R3.1. Flow cytometry analysis of intracellular GFP expression in EpCAM+ epithelial cells in the lungs from IL11-EGFP reporter mice after bleomycin injury.

With regards to the reviewers' comment on performing IL11-targeted antibody IF experiments. We have already provided immunostaining of IL11 in AT2 lineage traced cells in *Sftpc-tdT* mice in original **Fig 2h** and **Supplementary Fig 5c** which the reviewer might have missed. We showed that anti-IL11 staining is co-localized to subsets of lineage-labeled (*Sftpc-tdT*) AT2 cells after BLM-injury, which indicates that activated / differentiating AT2 cells may be potential sources of IL11 in the injured lung. Unfortunately, none of the anti-IL11 antibodies that we have tested were specific for flow cytometry analysis and hence we are unable to provide further quantification of IL11+ cells in the injured lung.

2) In Figure 2, and in all other panels including IF images, authors are always showing single field of view at medium magnification. Authors should include the stitches of multiple images reconstructing the entire lung organ. This will be much more convincing and therefore better supporting the conclusions.

Author response:

We apologize that we are unable to provide stitched images of the entire lung organ as we routinely only assess the lower left lungs for IF studies. As such, in the initial submission, we chose to present images of regions with observable alveolar damage (20-40X magnification) and carefully avoided displaying regions with obvious artifacts. Nonetheless, we have now included additional images (taken at 10X magnification) for GFP + SFTPC / PDPN counterstaining (Fig R3.2 and Supplementary Fig 7) which consistently shows that IL11 is induced specifically by the injured epithelium after lung injury.

Fig R3.2. Images of immunostaining for GFP and Sftpc or Pdpn in the lungs of IL11-EGFP reporter mice 7 days post-BLM injury. White arrows indicate GFP+ in marker positive cells.

3) Page 7, line 197, Authors claim “IL11EGFP was localized to numerous SFTPC+ cells adjacent to regions of tissue disruption with an elongated morphology suggestive of AT2-to-AT1 differentiation (Fig. 2c). The elongated shape is not a commonly recognized marker for ATII to ATI transitioning phase, include citations of the relevant literature supporting this statement or provide experimental evidence.

Author response:

We are perplexed by the reviewers’ comment here as the acquisition of an elongated/enlarged morphology by AT2 cells is a phenomenon central to the prevailing model of AT2 - AT1 cell differentiation in response to injury. Several recent in vivo lineage-tracing studies have now shown that AT2 cells, and to certain extent airway-derived cells, lose their cuboidal morphology and adopt squamous-like changes as these cells differentiate towards AT1 cells^{1,2,8}. It is also evident that differentiating AT2 cells gradually lose SFTPC expression during differentiation to Krt8+ / AT1 cells. It is for these reasons that we have also incorporated complementary AT2-lineage-tracing experiments with *Sftpc-tdT* mice at the end of Fig 2 to bolster our initial findings from our IL11-EGFP reporter mice.

To provide further experimental evidence for elongated/enlarged SFTPC+ cells during lung injury, we stained lungs of *Sftpc-tdT* mice for SFTPC and AGER or KRT8 to monitor for differentiating AT2 cells post-BLM injury (Fig R3.3). This revealed numerous elongated/enlarged SFTPC+ *tdT*+ and KRT8+ SFTPC+ *tdT*+ cells in the injured lung (marked by yellow cells/arrowheads) and provides evidence for the expression of SFTPC by differentiating elongated KRT8+ AT2-lineage cells. Furthermore, and as expected, there was considerable overlap of SFTPC staining and *tdT* signal.

Fig R3.3. Images of immunostaining for SFTPC and AGER or KRT8 in the lungs of *Sftpc-tdT* reporter mice post-BLM injury. Yellow arrows indicate elongated SFTPC+ *tdT*+ cells. Scale bar: 50 μ m.

4) In Figure 2e, authors should select a different region of the organ, possibly with no bronchi that are by definition KRT8+ and therefore are confounding in evaluating the increase in the KRT8+ cell number. If possible include images showing the highest possible number of alveoli. In addition, as shown by ref.13 in original text, the KRT8+ cells are: 1) SPC+, 2) in close proximity with the alveoli, upon bleomycin treatment; 3) some are lining the alveolar border. On the contrary the images shown in this manuscript are very different, no alveoli are shown and KRT8+ cells are spread all over lung parenchyma. Please select better representative images.

Author response:

We apologize for the unclear images of KRT8+ cells shown in *Fig 2e*. To address the reviewers' concerns, these images have been replaced with better representative images of KRT8+ cells in regions of alveolar damage (*Fig R3.4* and *R3.5* and incorporated into *Fig. 3* and revised *Supplementary Fig. 7e*).

Fig R3.4. Images of immunostaining for GFP and KRT8 in the lungs of IL11-EGFP reporter mice 7 days post-BLM injury. A indicates airways.

Fig R3.5. Additional images of immunostaining for GFP and Krt8 in the lungs of IL11-EGFP reporter mice 7 days post-BLM injury. White arrows indicate GFP+ KRT8+ cells.

5) In figure 2h the number of SPC+ cells is too high, and doesn't seem to be physiological. In a normal alveolus the area covered by ATII cells is 5% of the total alveolar surface, with an average ratio ATII to ATI of 2:1, furthermore the number of ATII cells in the intraparenchymal region is extremely high. Please be sure that the signal intensity levels are appropriate in order to avoid signal saturation. If this is not the problem, probe different slides and animals to find a more representative image and confirm your finding by staining using an antibody against SPC instead of the genetically encoded reporter gene.

Author response:

We acknowledge the reviewer's concerns over the numbers of SFTPC+ cells shown in uninjured mice in original **Fig 2h**, and assure the reviewer that these were not due to inappropriate *tdT* signal intensity. To reassure the reviewer of this, we have already provided IF staining showing considerable overlap of SFTPC and *tdT* signal from the lungs of *Sftpc-tdT* reporter mice in our response above (**Fig R3.3**). We have also now replaced the original images with better representative images showing the expression of IL11 in KRT8 expressing AT2-lineage traced cells (**Fig R3.6**). The respective enlarged images in **Fig R3.6** have now been incorporated into revised **Fig. 2k**.

Fig R3.6. (a) Images of immunostaining for SFTPC in lungs of tamoxifen induced Sftpc-tdT mice. (b) images of immunostaining for KRT8 and IL11 in lungs from Sftpc-tdT mice after BLM-injury.

6) To investigate the functional importance of IL11 in alveolar epithelial cells, authors decided to use the Human Pulmonary Alveolar Epithelial Cells (HPAEpiC, please correct the name in the manuscript). First, I don't understand the idea of moving into human cells to test KRT8 positivity that is a peculiar feature of ATII mouse transitioning cells. Second, these cells have very high level of IL11RA that is very different what is reported in Fig1 for human cells and figure 2 for mouse cells. In addition, the staining of SPC, reported in supplementary figure 6A shows a very strange pattern instead of the characteristic perinuclear pattern normally shown by SPC immunostainings in primary mouse and human ATII cells. In addition, authors claims that recombinant IL11 directly drives cytopathic effect in HPAEpiC cells. Authors should define what they mean for cytopathic effect and select the appropriate markers to measure the phenotype of interest. However, Pathological Extracellular Matrix (ECM) components (Collagen I, fibronectin), EMT related protein (SNAIL) and KRT8 are not standard markers of cellular toxicity and therefore, if authors meant to measure cellular toxicity, they must be implement their data by including at least one of the following markers: 1) apoptosis by (Caspase3/7 assay or Tunel); 2) cellular viability by ATP intracellular content or equivalent (CelltiterGlow assay).

Author response:

We thank the reviewer for highlighting these concerns. We have now corrected the nomenclature for HPAEpiC throughout the manuscript.

Our main decision to assess human cells for KRT8 expression was based on findings from several studies that classified KRT8-hi expression as a characteristic feature of aberrant epithelial cells in human IPF^{2,9}. Overall, our data clearly shows that both IL11 and TGF-beta stimulation triggers the upregulation of KRT8 protein and the acquisition of an EMT-like fate, which are also another notable pathological feature of aberrant epithelial cells in IPF.

As to the reviewer's second point about high levels of IL11RA in HPAEpiC. We point out that only data for IL11, but not for IL11RA, were presented in Figs 1 and 2. To provide better

clarity on the expression of *IL11RA*, we now include UMAP and dot plots of scRNA-seq expression profiles of *IL11RA* across all cell types in the human lung (**Fig. R3.7** and **R3.8** and incorporated into revised **Supplementary Fig. 2 - 3**). These show that lung fibroblasts express the highest levels of *IL11RA*, followed by endothelial and epithelial cells (AT1, AT2, KRT5⁺/KRT17⁺ and aberrant basaloid). In contrast, *IL11RA* is very lowly expressed by immune cells.

Fig R3.7. UMAP visualization of *IL11* or *IL11RA* expressing single cells in scRNA-seq data from control and PF samples in the (a-c) Habermann et. al. (GSE135893) and (d-f) Adams et. al. (GSE136831) dataset. Colored dots indicate different cell clusters in a and c.; *IL11* or *IL11RA* expressing cells are colored in dark blue in b, c, e, f.

Fig R3.8. *IL11RA* expression in various cell types in the human lung. Dot-plot showing the expression of *IL11RA* in individual cell types in scRNA-seq data from control and PF samples in the Habermann et. al. (GSE135893) and Adams et. al. (GSE136831) datasets.

With regards to the reviewer’s third point on the SFTPC immunostaining profile that was shown in **supplementary fig 6a**. We completely agree with the reviewer that the SFTPC staining profile did not appear to be expected of AT2 cells and would like to apologize for this oversight. It has also been brought to our attention that the anti-SFTPC antibody (Abcam, ab40879) that was used may have been unsuitable and that this antibody has also recently been discontinued. We have now re-assessed the expression of SFTPC in HPAEpiC using a different human specific anti-SFTPC antibody (H-8, from Santa Cruz Biotech), which now indicates the expected SFTPC expression in perinuclear regions (**Fig R3.9**).

Fig R3.9. Images of Immunostaining for SFTPC in HPAEpiC using (a) anti-SFTPC (Abcam ab40879) or (b) anti-SFTPC antibody (H-8, from Santa Cruz Biotech).

As to the reviewer’s last point on the use of the term “Cytopathic”, we apologize for the misleading use of this term to describe the pathologic ECM expression and EMT-related features of aberrant epithelial cells. These have been reworded to “EMT-related” or “profibrotic” to better reflect our focus and findings more accurately.

7) Figure3C please include a figure legend is too difficult to understand that the colors refer to Figure3A.

Author response:

We acknowledge this concern and have now included legends for all graphs in **Fig 3**.

8) Figure3d, all cellular proliferation studies have not been conducted in appropriate settings. Authors didn't mention for how long cells have been pulsed with Edu, this is fundamental to assess the proliferation rate. In addition, study cellular proliferation in HPAEpiC cells with 10-15% basal proliferation is not supporting any conclusion on ATII cells that, on the contrary, doesn't show a very limited proliferative potential in vitro and are very poorly proliferative in vivo. This set of experiment is misleading and improper. Either remove or reproduce in primary mouse ATII cells. Last, EdU is not a marker of proliferation but a marker for DNA synthesis, please show in the implement with the total cell numbers for each experimental condition, correlate it with EdU positivity and exclude DNA damage by γ -H2AX staining. Staining for other proliferation markers such as Ki-67, phosphor histone H3 or AuroraB kinase. The authors must coinfirm all the experiments reported in figure3, A,B,C, D by using mouse primary alveolar ATII cells.

Author response:

We acknowledge the reviewers' concerns about the EdU data shown in original **Fig 3d**, and would like to highlight that the reported proliferation rates observed were based on cells cultured in the commercially recommended AEpiCM media, which may contain certain growth stimulants, that are not disclosed by the retailer, that can promote proliferation. We also now disclose that the cells were pulsed with EdU for 22 hours prior to cell fixation and staining and also provide new representative images for EdU staining and graphed the mean cell numbers analyzed in each field for the reviewer's consideration (**Fig R3.10**). Overall, we observed a non-statistical trend of lower cell numbers in IL11 and TGF-beta stimulated cells over the 24 hours stimulation period. Nonetheless, in view of these and a similar concern by another reviewer, we have now refocused main **Fig 3** on profibrotic EMT-related effects and have moved the EdU proliferation data into the supplement.

Fig R3.10. Images of EdU staining in HPAEpiC treated with IL11, TGF β 1, X203 or IgG control antibodies. Scale bars: 100 μ m.

The reviewer has also made a request to replicate our findings in HPAEpiC on mouse AT2 cells. However, due to technical challenges in performing short term stimulation studies on isolated primary mouse AT2 cells, which include low cell viability and poor cell attachment capacity of primary AT2 cells, we were only able to performed short term stimulation studies on mouse AT2 cells and qualitatively assess the expression of the EMT-related marker (Collagen I). To do this, we cultured freshly isolated AT2 cells from tamoxifen-exposed *Sftpc-tdT* mice and treated the cells with cytokines and antibodies for 48 hours. Consistent

with our human data, we show that recombinant mouse IL11 directly induces Collagen I expression and that IL11 acts downstream of TGF-beta in mouse AT2 cells (**Fig R3.11**). Further, the effects of IL11 on the expression of Collagen I in mouse AT2 cells can also be blocked by the MEK inhibitor U0126.

Additionally, and in view of the reviewers' comments here, we have also replicated our findings on primary human small airway epithelial cells (HSAEC) (**Fig R3.12**). Overall, our data consistently shows that IL11 induces profibrotic EMT-related changes in human alveolar and distal airway epithelial cells and mouse AT2 cells. We now incorporate these new in vitro data in **Fig. 3f-i** and in **Supplementary Fig 11**.

Fig R3.11. (a) Gating for FACS sorting of AT2 cells (CD45⁻ CD31⁻ EpCAM⁺ tdT⁺) from *Sftpc-tdT* reporter mice. Representative images of immunostaining of Collagen I (green), in *Sftpc-tdT*⁺ cells (red) from *Sftpc-tdT* mice treated with (b) IL11 (5 ng/ml), TGFβ1 (5 ng/ml) in the presence of anti-IL11 (X203) or IgG control antibodies (2 μg/ml) or (c) IL11 (5 ng/ml) in the presence/absence of MEK inhibitor U0126 (10 μM) for 48 hours. Scale bars: 50 μm.

Fig R3.12. (a-d) Representative images of immunostaining of Fibronectin and Collagen I (green), in primary human small airway epithelial cells (HSAEC) treated as indicated for 24 hours. Scale bars: 100 μ m. (e) Secreted collagen concentration was determined by sirius red assay.

9) Figure 3F: I suppose that these cells are primary murine ATII cells, isolated from the transgenic mice. It is not well described in the text, please correct. Page 19 line 574, "The FACS sorted cells were then seeded at a density of $2e4$ cells per well in rat tail collagen (Invitrogen, A1048301) coated 96-well CellCarrier plates (PerkinElmer) and cultured in

DMEM supplemented with 10% FBS. The cells were allowed to adhere for 24 hours prior to treatment with mouse IL11 (UniProtKB: P47873, GenScript) or mouse TGFβ1 (R&D Systems, 7666-MB). It is known in the field that ATII cells doesn't survive isolation if cultured in DMEM, specific and properly supplemented medium are needed, in example the commercially available Pneumacult (StemCellTechnology). Secondly, at 5 days post isolation, viable ATII cells, if cultured in collagen coated plastic plates, should be almost completely transdifferentiated in ATI cells, with no SPC positive cells and the majority of RAGE+ with almost no transitioning (double positive) cells. IN figure 3F we can appreciate some PDPLN positive cells, as expected, that are also 100% SPC positive. This is not resembling the normal biology of cultured ATII cells and we invite the author to probe the cells with anti SPC antibody to confirm SPC positivity by IF staining. In addition please include larger field of view for all conditions.

Author response:

We apologize for any lack of clarity in our description for the primary mouse AT2 cells used in **Fig 3** and have reworded those lines accordingly.

“To test if IL11 stalls the transition of AT2 cells into mature AT1 cells, we firstly isolated mouse AT2 cells from tamoxifen-exposed Sftpc-tdT mice by FACS sorting for constitutive tdT⁺-expressing cells and cultured these primary AT2 cells under 2D culture conditions followed by treatment with IL11 (5 ng/ml) from day 1 to day 5”

We also acknowledge the reviewer's concerns about the media (DMEM media containing up to 10% FBS) used in mouse AT2 cultures. However, this media has been described in the literature to be suitable for the differentiation of mouse AT2 cells to AT1-like cells in 2D cultures. We refer the reviewer to these publications, which we have adapted and cited in our manuscript^{10,11}. Additionally, the reviewer has mentioned that specialized media such as the Pneumocult media can be tested here. However, this specialized media is only recommended for human airway cells and contains species-incompatible factors for mouse cell cultures.

The reviewer also appears to be confused about the expression of *Sftpc-tdT* in our AT2 cultures in original **Fig 3e/f**. To clarify, we used FACS sorted EpCAM⁺ *tdT*⁺ cells isolated from tamoxifen-exposed *Sftpc-Cre;Rosa26-tdTomato (Sftpc-tdT)* mice for this assay. In this model, *tdT* is constitutively expressed by *Sftpc*-expressing AT2 cells upon tamoxifen treatment. Hence, the red *tdT* fluorescence signal shown in **Fig 3** indicates cells that are constitutively expressing *tdT* and is not a direct reflection of active SFTPC levels.

To address the reviewer's last point, we have now included new images with larger fields of view across all conditions along the differentiation time points (day 1, 3, 5) (**Fig R3.13** and revised **Supplementary Fig. 12**).

Fig R3.13. Time series (day 1, 3 and 5) of cultures of *Sftpc-tdT*⁺ AT2 cells treated with IL11 (5 ng/ml), TGFβ1 (5 ng/ml), X203 or IgG control antibodies (2 μg/ml). *tdT*⁺ cells (red) were immunostained for KRT8 (green), PDPN (white) and counterstained with DAPI (blue). Scale bars: 50 μm.

10) Figure 4, authors should include Masson's trichrome staining for the experiment reported in panel B and IF stainings as in B for panel F.

Author response:

We thank the reviewer for this suggestion and have now included Masson's trichrome staining, ashcroft scoring and lung hydroxyproline content of AT2-specific *Il11ral* deleted mice at the early day 12 time point. We observed that mice with AT2-specific *Il11ral* deletion were similarly protected from fibrosis at this earlier time point (Fig R3.14) which further reinforces the importance of IL11 signaling in AT2 cells for the initiation of lung fibrosis. These new data are incorporated into revised Supplementary Fig 14.

Fig R3.14. Images of Masson's trichrome staining, lung histopathological fibrosis scores, lung hydroxyproline content of right caudal lobes of *Sftpc-CreER; Il11ra1^{fl/fl}* or *Il11ra1^{+/+}* mice 12 days post-BLM injury. Dotted lines indicate mean values of uninjured controls.

11) In the images reported in Figure4B for the *Il11ra1* floxed condition, the SP-C staining is very different from the control condition. In the *Il11ra1^{+/+}* condition, SP-C staining appears as expected by staining ATII cells (small cuboidal structure). In the *Il11ra1* floxed condition, the SP-C staining is completely lining the alveolus resembling the shape of ATI cells. The authors in the text claims "Instead, we found numerous newly differentiated AT1 cells including regions of completely formed alveoli that were *tdT⁺* in BLM-treated *Sftpc-tdT; Il11ra1^{fl/fl}* mice (Fig. 4b-d)." (Page 11 line 325). The overlap between *Sftpc-tdT* and PDPN, in the selected image, is almost 100%, thus suggesting that after Tamoxifen treatment all ATI derives from ATII to ATI differentiation. This appears honestly too effective, thus we suggest to 1) show additional images from other lung regions; 2) Further confirm by SPC/RAGE immunostaining.

Author response:

We acknowledge the reviewers' concerns here that regenerative phenotype observed in *Sftpc-tdT; Il11ra1^{fl/fl}* mice may appear rather too effective. To further support this, we have now included additional images in the revised supplement, which consistently shows equally large regions of newly regenerated alveoli marked by flattened *tdT⁺*+PDPN+ cells and devoid of KRT8hi expressing cells across different replicates of *Sftpc-tdT; Il11ra1^{fl/fl}* mice (Fig R3.15). We have also performed SFTPC and AGER immunostaining as per the reviewers' request (Fig R3.16), which consistently highlights regions of newly regenerated alveoli marked by flattened SFTPC+ *tdT⁺*+ AGER+ cells in the lungs of *Sftpc-tdT; Il11ra1^{fl/fl}* mice as compared to controls. We hope that the additional data here provides further proof of principle that *Il11ra1*-deletion in AT2 cells greatly enhances alveolar epithelial repair after injury.

Fig R3.15. Immunostaining for KRT8 and PDPN in lungs of additional replicates of *Sftpc-tdT*; *Il11ra1^{fl/fl}* or *Il11ra1^{+/+}* mice post-BLM challenge. Scale bars: 100 μ m.

Fig R3.16. Immunostaining for AGER and SFTPC in lungs of *Sftpc-tdT*; *Il11ra1^{fl/fl}* or *Il11ra1^{+/+}* mice post-BLM challenge. Scale bars: 100 μ m.

In addition to these profound histological findings, we wish to highlight that we have now completed and incorporated two additional scRNA-seq experiments (for this revision) on enriched epithelial cell populations from: 1) *Sftpc-CreER*; *Il11ra1^{fl/fl}* mice and 2) from X203/IgG treated mice after BLM injury, to further uncover mechanisms by which IL11-signaling in AT2-lineage cells promotes fibrosis (**Fig 4e-i**, **Fig 5g-i** and **Supplementary Fig 15, 16** and **22**). Overall, our new computational analyses uncovered strong associations between IL11-signaling and aberrant transcriptomic signatures of multiple disease-related pathways (such as EMT, TGF-beta signaling, p53, inflammation and unfolded protein response pathways) in Krt8+ transitional cells. Furthermore, genes related to EMT in Krt8+ transitional cells such as (*Colla1*, *Fn1* and *Ccn2*) were amongst the most downregulated from abrogated IL11-signaling. Additionally, IF validation of ECM and profibrotic related proteins Collagen I and CTGF, showed that abrogated IL11-signaling strongly attenuates pathologic ECM and stress related responses of Krt8+ transitional cells. These key findings demonstrate that IL11 signaling in AT2 cells likely promotes the profibrotic phenotypes of Krt8+ transitional cells and that these cells may also directly contribute to the aberrant accumulation of ECM during lung fibrosis.

Moreover, our trajectory analysis also predicts that *Il11ra1*-deleted AT2 cells may potentially differentiate more effectively towards AT1-cells after injury by bypassing the profibrotic

Krt8⁺ state (**Supplementary Fig 16e**), which are in line with the pro regenerative phenotype observed in histological findings from *Sftpc-tdT*; *Il11ra1^{fl/fl}* mice. Interestingly, a similar model of effective alveolar progenitor cell differentiation to AT1 cells via a separate intermediate transitional AT1 state that is distinct from stressed Krt8⁺ state has recently been reported from detailed observations of mouse alveolar organoid differentiation¹². However, direct *in vivo* evidence for this alternate differentiation program is still currently lacking and that our observations in *Sftpc-tdT*; *Il11ra1^{fl/fl}* mice are highly reminiscent of this alternative differentiation route. We hypothesize that abrogated IL11 signaling may promote AT2-to-AT1 differentiation via this less stressed transitional state. Further detailed studies that interrogate multiple disease time points in our genetic model may provide additional *in vivo* evidence to support this.

12) Experimental timepoints for the *in vivo* experiments are always different, without and obvious explanation, in example:

- **Figure 2a Bleo day 0 and tissue collection at 7-10 and 21 days**
- **Figure 2g Bleo day 0 and tissue collection at 14 days**
- **Figure 4a Bleo day 0 and tissue collection at 12 days**
- **Figure 4e Bleo day 0 and tissue collection at 21 days**
- **Figure 5a Bleo day 0 and tissue collection at 12 days**

It is known that fibrotic remodelling upon bleomycin treatment follows a specific kinetic: Day 0-14 (Inflammation), Day 21 (starts fibrotic scarring), Day 30 (peak of fibrotic remodelling), Day 45 (start of spontaneous resolution), Day 60 (evident resolution). The author must specifically explain why they selected these timepoints, and why these are different in almost every experiment.

Author response:

For the large part, our lineage-tracing studies (using *Sftpc-tdT* and *IL11-EGFP* reporter mice) were designed to capture the transient occurrence of injury-emergent Krt8⁺ transitional cells. These time points were carefully selected based on those described in recent studies from Schiller's, Lee's and Tata's groups^{1,2,8} - that clearly demonstrated that the accumulation of Krt8⁺ ADI / DATPS / PATS cells usually begin as early as 3 days post-BLM injury and peaks at around day 10-14. The numbers of alveolar Krt8⁺ cells then gradually declines after 14 days as fibrosis resolution usually occurs after this point.

As such, 12-14 day time points were carefully selected in our AT2-lineage tracing studies to monitor for the peak of alveolar Krt8⁺ cells. Additionally, these time points also allowed us to monitor for the elongated morphology of *tdT* lineage-traced cells as these AT2 cells differentiate towards AT1 cells, again as described in the studies by Tata and Lee^{1,8}.

Likewise, for our experiments using *IL11-EGFP* reporter mice, early time points of day 7-10 were selected to best capture the accumulation of transitional epithelial cells, and at day 21 to monitor for the decline in transitional cells during the early resolution phase.

For fibrosis end points, the reviewer is correct to point out that fibrosis remodeling peaks after the second week post-injury. We have in fact observed extensive lung fibrosis by day 12 post-BLM (from new data described in **point 10** above; **Fig R3.14**). Survival studies were also performed until day 21 as mortality is not commonly observed in the first 2 weeks post-BLM injury in our models.

13) Figure4F, the lung from *IL11RA1*^{+/+} in uninjured condition shows a clear immune infiltration and this must be commented in the text. In addition authors should provide complete lung images with enlargement of selected portions. It's better if the enlarged portion is always selected in the same region in different experimental conditions.

Author response:

The reviewer makes a valid point regarding the choice of images shown in original **Fig 4f**, that in hindsight, were not representative and showed signs of cell infiltrates / compressed alveolar structures. We have now replaced these with more representative images from distal regions showing the expected uninjured lung architectures (**Fig R3.18** and revised **Supplementary fig 13**).

Fig R3.18. Images of Masson's trichrome staining in uninjured lungs of *Sftpc-CreER; Il11ra1^{fl/fl}* or *Il11ra1^{+/+}* mice.

14) Figure5b, please provide Masson's trichrome staining and lung hydroxyproline content for all experimental conditions. In addition, authors should provide complete lung images with enlargement of selected portions. It's better if the enlarged portion is always selected in the same region in different experimental conditions.

Author response:

As requested by the reviewer, we have now included Masson's trichrome staining, ashcroft scoring and lung hydroxyproline content for in vivo therapeutic studies (**Fig R3.19** and revised **Supplementary fig 19**). Our data shows that mice treated with BLM+X203 displayed marked reduction in collagen deposition, lung damage and fibrosis as compared to BLM+IgG controls.

Fig R3.19. (a) Representative images of Masson's trichrome staining, (b) lung histopathological fibrosis scoring and (c) lung hydroxyproline content of right caudal lung lobes of *Sftpc-tdT* mice treated with X203 or IgG antibodies 12 days post-BLM injury.

15) In figure 5b, in the BLM+X203 treated condition, the *Sftpc-tdT* staining has a completely different pattern compared to what is shown in the bleomycin-treated *IL11RA1-/-* Figure4b. Here in figure 5b the pattern is what is normally expected, showing single SPC positive cells with cuboidal shape usually seen in ATII cells. On the contrary in figure4b the *Sftpc-tdT* is completely overlapping with PDPN pattern. Here there are two options: 1) the X203 is protective but not so effective in promoting ATII to ATI trans-differentiation and most likely working on lung fibroblast; 2) the ATII to ATI trans-differentiation shown in the figure4b is highly overestimated. Please explain, select better and more representative images or remove.

Author response:

The reviewer makes a repeat comment here regarding the effectiveness of AT2 - AT1 transdifferentiation in injured *Sftpc-tdT*; *Il11ral-floxed* mice in **Fig 4b**, which we have elaborated extensively on in **point 11** above.

Although we are uncertain as to the precise extent in which X203 affects the function and/or differentiation of various cell types in the injured lung, our data clearly shows that X203 can have profound effects on alveolar epithelial regeneration after lung injury. Importantly, our findings indicate that IL11-signaling in AT2 cells is crucial for the development of lung fibrosis and that IL11-inhibition likely promotes alveolar repair by limiting the differentiation and aberrant profibrotic phenotypes of Krt8+ transitional cells.

Given that IL11 can be secreted by and may have prominent role(s) in the differentiation of pathological fibroblasts, we hypothesize that IL11 may have additional detrimental effects on alveolar regeneration by modulating the alveolar epithelial support functions of mesenchymal cells. Coincidentally, this theory has recently been tested by another group¹³. In their study, the authors utilized unbiased lung fibroblast secretome analysis and showed that the expression of IL11 by pathological lung fibroblasts from ILD-patients can potentially initiate aberrant epithelial differentiation signatures in iPSC-derived alveolar organoids cultures.

Recent studies have also revealed that disease-specific CTHRC1-expressing lung fibroblasts emerge in the fibrotic lungs of BLM injured mice and in human IPF, and that these pathological fibroblasts express high levels of TGF-beta and are closely associated with aberrant epithelial cells in the fibrotic niche^{14,15}. To build on this and to test the effects of X203 treatment on pathological fibroblasts in the injured lung, we performed preliminary protein staining for CTHRC1 in the lungs of BLM + IgG/X203 treated mice. This showed that X203-treatment significantly reduces the numbers of disease-specific CTHRC1+ cells in injured regions after BLM-injury as compared to IgG (**Fig R3.20**), which suggests that IL11-inhibition may help prevent the differentiation/accumulation of pathological lung fibroblasts that may also have consequential effects on alveolar regeneration in this context. These interesting findings raise important questions about the additional role(s) of IL11 in the activated lung stromal compartment in the fibrotic niche, which clearly warrants future investigation.

Fig R3.20. Images of immunostaining for CTHRC1 in injured lung regions from Sftpc-tdT mice treated with X203 or IgG antibodies 12 days post-BLM injury. Scale bars: 100 μ m.

16) Page 14 line 382, “Additionally, in IgG treated mice, we found an increase in non-lineage labelled KRT8+ cells (KRT8+ tdT-) that stained weakly for the AT1 marker PDPN (Fig. 5b-d)” We can appreciate an increase in the KRT8+ cells but the PDPN staining in IgG treated condition is even stronger than ctrl condition. This sentence is not supported by the images. . Please explain, select better and more representative images or remove.

Author response:

We acknowledge that our image selection of uninjured lungs did not show the appropriate levels of PDPN staining and thank the reviewer for highlighting this to us. We have now selected better representative images of uninjured controls in revised Fig 5.

Fig R3.21. Images of immunostaining for KRT8 and PDPN in lung sections from uninjured tamoxifen-exposed Sftpc-tdT mice. Scale bars: 100 μ m.

17) Figure5f, the only reasonable conclusion we can take from this image is a reduction in CollagenI deposition upon Bleo+X203 treatment. Other conclusions regarding possible mechanistic links between IL11 and p-ERK pathway should be supported with dedicate biochemical studies. In addition, how could you exclude that X203 is acting on lung fibroblast? Please provide relevant data on this last question.

Author response:

The reviewer suggests additional biochemical studies of IL11/ERK and Collagen expression from our in vivo BLM+X203 experiment. As such, we have now included hydroxyproline content measurements of BLM+IgG/X203 mice in our responses above (Fig R3.19). We have also provided substantial evidence to support the link between IL11 and p-ERK in the acquisition of KRT8-hi state in vivo and in vitro (Supplementary Fig. 10-11). Furthermore, we now provide additional IF staining showing that X203-treatment not only reduces the expression of Collagen I (Fig 5), but also reduces the expression of profibrotic CTGF protein

and the ER-stress marker XBP1 in Krt8+ transitional cells after BLM-injury (**Fig. R3.22** and **Supplementary Fig. 22d-e**). Notably, these changes are associated with the reductions in the numbers of p-ERK+ KRT8+ cells in X203-treated mice (**Supplementary Fig. 21a**) and demonstrates a link between IL11-ERK signaling and the potentiation of profibrotic Krt8+ cell states *in vivo*.

As for the reviewers' comment on the effects of X203 on lung fibroblasts, we have elaborated on this in our response to **point 15** above.

Fig. R3.22. Representative images of immunostaining of KRT8 and (d) XBP1 or (e) CTGF in the lungs of X203 or IgG treated *Sftpc-tdT* mice post-BLM-injury. Scale bars: 100 μ m.

18) Line 441 page 16 “Furthermore, we found that the anti-proliferative effects of TGF β and its induction of ECM proteins and KRT8 expression in AT2 cells was, in part, mediated by IL11 signaling.” Rephrase according to point 6,7,8

Author response:

We have now rephrased the sentence to exclude the emphasis of the proliferation phenotypes accordingly.

“Furthermore, we found that the effects of TGF β and its induction of pathologic ECM proteins and KRT8 expression in AT2 cells was, in part, mediated by IL11 signaling”

19) Line 457, page 16. “Given the elevated expression of IL11 in aberrant mesenchymal and epithelial cell types in PF and its roles in both fibroblasts activation and AT2 cell dysfunction, we propose that IL11 may cause multiple aspects of pathobiology in different cell types in the diseased niche.” In human or in mouse?

Author response:

We believe that the computational and experimental evidence presented in this study strongly supports the role of IL11 in promoting AT2 cell dysfunction across both species.

References

1. Choi, J. *et al.* Inflammatory Signals Induce AT2 Cell-Derived Damage-Associated Transient Progenitors that Mediate Alveolar Regeneration. *Cell Stem Cell* **27**, 366–382.e7 (2020).
2. Strunz, M. *et al.* Alveolar regeneration through a Krt8⁺ transitional stem cell state that persists in human lung fibrosis. *Nat. Commun.* **11**, 3559 (2020).
3. Choi, J. *et al.* Release of Notch activity coordinated by IL-1 β signalling confers differentiation plasticity of airway progenitors via Fosl2 during alveolar regeneration. *Nat. Cell Biol.* **23**, 953–966 (2021).
4. Galuppo, M. *et al.* MEK inhibition suppresses the development of lung fibrosis in the bleomycin model. *Naunyn. Schmiedebergs. Arch. Pharmacol.* **384**, 21–37 (2011).
5. Ng, B. *et al.* Interleukin-11 is a therapeutic target in idiopathic pulmonary fibrosis. *Sci. Transl. Med.* **11**, (2019).
6. Ng, B. *et al.* Similarities and differences between IL11 and IL11RA1 knockout mice for lung fibro-inflammation, fertility and craniosynostosis. *Sci. Rep.* **11**, 14088 (2021).
7. Bai, X. *et al.* Inhaled siRNA nanoparticles targeting inhibit lung fibrosis and improve pulmonary function post-bleomycin challenge. *Sci Adv* **8**, eabn7162 (2022).
8. Kobayashi, Y. *et al.* Persistence of a regeneration-associated, transitional alveolar epithelial cell state in pulmonary fibrosis. *Nat. Cell Biol.* **22**, 934–946 (2020).
9. Jiang, P. *et al.* Ineffectual Type 2-to-Type 1 Alveolar Epithelial Cell Differentiation in Idiopathic Pulmonary Fibrosis: Persistence of the KRT8hi Transitional State. *Am. J. Respir. Crit. Care Med.* **201**, 1443–1447 (2020).
10. Chen, Q. & Liu, Y. Isolation and culture of mouse alveolar type II cells to study type II to type I cell differentiation. *STAR Protoc* **2**, 100241 (2021).

11. Riemondy, K. A. *et al.* Single cell RNA sequencing identifies TGF β as a key regenerative cue following LPS-induced lung injury. *JCI Insight* **5**, (2019).
12. Toth, A. *et al.* Alveolar epithelial progenitor cells require Nkx2-1 to maintain progenitor-specific epigenomic state during lung homeostasis and regeneration. *Nat. Commun.* **14**, 8452 (2023).
13. Kastlmeier, M. T. *et al.* Cytokine signaling converging on inq inq fibroblasts provokes aberrant epithelial differentiation signatures. *Front. Immunol.* **14**, 1128239 (2023).
14. Tsukui, T. *et al.* Collagen-producing lung cell atlas identifies multiple subsets with distinct localization and relevance to fibrosis. *Nat. Commun.* **11**, 1920 (2020).
15. Kathiriya, J. J. *et al.* Human alveolar type 2 epithelium transdifferentiates into metaplastic KRT5 basal cells. *Nat. Cell Biol.* **24**, 10–23 (2022).

REVIEWER COMMENTS

Reviewer #1 (Remarks to the Author):

The authors have very satisfactorily addressed all my concerns and are to be congratulated for an excellent study.

Reviewer #2 (Remarks to the Author):

The authors have made significant progress in addressing previous concerns; however, additional refinement is necessary in the immunofluorescent and flow cytometric analyses to clarify the results.

The specific comments are as follows:

1. The fluorescent signals for the specified proteins are overly intense across several figures, which obscures the margins of cells and intracellular distribution of the indicated molecules in positive cells. It is advisable for the authors to modulate the fluorescent intensities to mitigate this issue.
2. The discrimination of Cldn4^{high} and Cldn4^{low} cells in Figures 2g-i and Supplementary Figure 8 is unclear. The average intensities of Cldn4 in lung epithelial cells from BLM-treated mice appeared to shift right side compared to those from untreated mice. The authors should compare the histograms of the expression of Cldn4 in lung epithelial cells from both uninjured and BLM-treated mice. The authors should explain this issue in detail. Incorporating isotype control antibodies in Supplementary Figure 8c would be required. Moreover, It would be beneficial to increase the count of EPCAM⁺ cells shown on BLM day 10 in flow cytometry to distinguish GFP-positive cells more clearly.
3. There is a discrepancy between FACS and IHC data. The authors should carefully calculate and interpret the results in Figure 2g and FigR1.1 regarding IL11-expressing epithelial cells. According to our calculation, Cldn4^{low} GFP⁺ cells and Cldn4^{high} GFP⁺ cells comprise 0.64% (0.87 x 0.73%) and 0.87% (0.129 x 6.73%) among all epithelial cells, respectively. Thus, the authors should reanalyze other data and create a new Figure to include and compare

accurate percentages of Cldn4^{high} GFP⁺ and Cldn4^{low} GFP⁺ cells among all epithelial cells. This reanalysis should address the inconsistencies and ensure the results are accurately presented in the context of the study.

Similarly, data from Figure 2e show that IL-11-GFP⁺ cells do not express KRT8, which further implies that a subset of EpCAM⁺Cldn4^{low} cells also exhibit GFP expression.

4. The authors should provide a detailed method for how the protein intensity calculations were performed in Figures 3e and i, specifying whether they were conducted across different areas within a single well or multiple wells.

5. It would be instructive for the authors to calculate the percentages of collagen 1⁺ cells among Sftpc⁺ cells under various conditions, and show in Supplementary Figures 11g and h.

6. In Figures 5e and f, the Cldn4⁺ and PDPN⁺ cells do not form distinct populations. To resolve this, an increase in the number of analyzed EpCAM⁺ cells via flow cytometry is recommended to delineate these cell populations more effectively.

7. The expression patterns of IL11ra1 in different types of epithelial cells in mice remain unclear. In addition, Figure 5g and Supplementary Figures 15 and 22 should include the characteristic genes of each identified cell cluster for better clarity.

8. Moreover, a recent study reported that IL-11 induced by TGF β does not play a crucial role in TGF β -induced fibrosis (Tan et al., Front Immunol 2024). Thus, the authors should discuss this point in detail in the Discussion.

Reviewer #3 (Remarks to the Author):

Dear Authros,

please refer to the attached Pdf document that includes a point by point reply to the rebuttal letter.

[**editorial note:** please see the next page(s) for the PDF.]

General

I would like to express my gratitude to the authors for their thorough response to the comments and for the extensive addition of new data in the revised version of the paper leading to a significant overhaul of the manuscript, which includes two main figures that are largely new (Figures 3 and 4) as well as the new set of data on the *Sftpc-CreER; Il11fl/fl* mice reported in supplementary 12. However, there are still some aspects that, in my opinion, need further clarification.

The EPCAM⁺ IL-11⁺ cell population is exceedingly rare, and its biological relevance has been extensively evaluated in this study, yielding convincing results regarding the pathological role of IL11 in alveolar epithelial cells. The authors have provided extensive results from lineage-traced ATII cell experiments, demonstrating that ATII-restricted *Il1ra1* knockout promotes ATII to ATI transdifferentiation, thereby supporting alveolar repair following bleomycin injury in mice. This effect is appreciable both histologically and biochemically (HP) in experiments utilizing the *Il1ra1* knockout mouse model.

It is a common consensus in the field that IL11 represents a valuable target for the development of anti-IL11 therapies for IPF either using antibodies targeting IL11 or its receptor. Hence, the authors of this manuscript, who have extensive background in anti-IL11 treatment for cardiac fibrosis are understandably motivated to demonstrate the efficacy of the anti-IL11 treatments in other fibrotic diseases, such as IPF.

In my assessment, upon reviewing histological images and HP data, the efficacy of X203 appears less pronounced compared to the ATII-restricted *Il1ra1* KO in the bleomycin-induced lung fibrosis model. This well fit the idea that genetic ablation of the receptor is more effective than pharmacological inhibition of the ligand.

What still remains counterintuitive, are the disparities in the impact of X203 on restoring ATII to ATI transdifferentiation, which, overall, seems convincing, yet demonstrates limited effectiveness in reducing lung fibrosis. Although there seems to be a reduction in lung fibrosis based on histological examination (only one whole slide shown), this reduction lacks significance in terms of HP contents, given the small sample size of only 4 animals per condition, which is a very limited number for this type of study. Moreover, lungs treated with X203 exhibit evident signs of tissue remodeling, including alveolar loss, when compared to the IL11 knockout condition (once again, only one slide shown) (Supplementary Figure 19).

Considering the striking impact of X203 demonstrated in Figure 5b in promoting ATII to ATI transdifferentiation, I am apprehensive that the limited efficacy observed in Masson's trichrome staining and HP content may be attributed to the shorter time window (12 days) utilized for this series of experiments. It is plausible that evaluating lung tissue at 21 days post-bleomycin administration would have provided a more fully resolved phenotype as a long waved consequence of the improve ATII to ATI trans-differentiation capacity.

Last, all the *in vivo* experiments in mice, despite the use of different genetic strains presents uninjured control lungs, that looks all but not uninjured, thus eliciting the possibility of concomitant lung infections or generally not very well controlled experimental conditions in the bleomycin-mouse model. It is recognized in the field that the bleomycin model of lung fibrosis needs a very careful optimization to be informative and reproducible as well as

adequate numbers of animal in each experimental group to balance the intrinsically high variability of the model.

Here below it follows a point-by-point discussion of authors' rebuttal letter, including additional request to the authors.

1) I appreciate their inclusion of flow cytometry analysis using an anti-GFP antibody instead of relying solely on the endogenously encoded EGFP (Figure R3.1). However, we have some concerns about the following sentence:

“With regards to the reviewers’ comment on performing IL11-targeted antibody IF experiments. We have already provided immunostaining of IL11 in AT2 lineage traced cells in Sftpc-tdT mice in original Fig 2h and Supplementary Fig 5c which the reviewer might have missed. We showed that anti-IL11 staining is co-localized to subsets of lineage-labeled (Sftpc-tdT) AT2 cells after BLM-injury, which indicates that activated / differentiating AT2 cells may be potential sources of IL11 in the injured lung. Unfortunately, none of the anti- IL11 antibodies that we have tested were specific for flow cytometry analysis and hence we are unable to provide further quantification of IL11+ cells in the injured lung.”

“Could authors please confirm their findings by showing an IF experiment using an antibody targeting IL-11 instead of the genetically encoded reporter gene?”

Here my request was very simple, and I’m sorry if I wasn’t clear on this point. I was actually asking for an IF image of Uninjured vs Bleo-injured lungs of IL11-EGFP mice **stained for IL11/SFTPC/IL11-EGFP and DAPI** (which is different from what authors included in Supplementary Figure 7g (EGFP-IL11) and h (SFTPCtdT-IL11-KRT8)). Could the author please provide the reviewer with this set of images?

I was and I am still concerned that the rarity of events shown in Fig 2B doesn’t match with the abundance of cells reported in figure 2C and highlighted by **“White arrowheads indicate marker positive IL11EGFP+ cells.”** There are 10 highlighted objects (IL11/SFTPC) in a field of view with ≈ 200 total cells, out of which epithelial cells are a subpopulation inclusive of the SFTPC cells. Therefore, it results in a % of IL-11-positive lung epithelial cells that has to be **higher** than 4% (10/250), which is however 25 folds (4/0.150) more than the one showed in figure 2A by cytofluorimetry. For this specific reason we asked to stain tissue slides with anti IL-11 and anti SPC antibody to assess if the events labelled by the genetically encoded IL-11-EGFP were truly IL-11 expressing cells. It’s a simple experiment and we would have really appreciated if the authors have included It in the supplementary figures.

Could author please either comment on this point, or include a better representative image for figure 2b?

2) I am ok with figure R3.2. and supplementary figure 7 regarding the stitched images.

3) I thank the authors for the reply and for providing additional evidence, and I’m ok with Fig R3.3.

4) I thank the authors for providing additional images as requested. I’m ok with the images reported in R3.4 and 3.5, and for incorporating R3.4 image in Figure 2.

5) I thank the authors for the reply and for providing additional evidence as requested. I agree on including R3.6 B in main figure 2 k.

6-8) I thank the authors for the reply. I align with the authors on the KRT8-hi presence in human lungs as reported by Kobayashi and Strunz (2 and 9 of rebuttal letter), and also Jiang-Zemans 2020 in AJRCCM “*Ineffectual Type 2-to-Type 1 Alveolar Epithelial Cell Differentiation in Idiopathic Pulmonary Fibrosis: Persistence of the KRT8hi Transitional State*”. I’m ok with the new SFTPC staining, that now resembles much better SFTPC staining usually seen in primary ATII cells. I also agree on the rewording of “Cytopathic”.

In my humble opinion the use of HPAEpiC and HSAEC cells, simply doesn’t add any value to the story. Both the Strunz and Kobayashi’s papers, as well as this manuscript, largely model the KRT8+ cells in IPF pathology using mouse models, and therefore the obvious follow-up set of experiments, after figure 2, **should have included only data on primary mouse ATII cells or human primary ATII cells, if the aim was to show efficacy in human cells.**

In alignment with our opinion, the data shown in Figure 3E on KRT8 (intensity/area) are quite different if compared with Figure 3I KRT8 (intensity/area). In particular, the difference between TGFb1+IgG and TGFb1+X203 are partially, if not completely, blunted in primary mouse ATII cells.

Consequently, I appreciated the decision of the authors to include some data on mouse primary ATII cells in Figure 3 J I K and supplementary Fig 11 F-H. However, I kindly ask to the authors to include in figure 3 the IF images present in Fig 11 F-H and a violin plot corresponding to the systematic quantification of Collagen1(intensity/area) in primary ATII cells, thus generating a similar set of data as the one shown in figure3E-I for HPAEpiC and HSAEC respectively

Page 9 Line 262: “AT2 cell proliferation is crucial for alveolar repair after injury and we tested the effects of IL11 or TGFβ1 ~~on AT2 cell proliferation~~” please either provide data on primary mouse or human AT2 cells or rephrase in “AT2 cell proliferation is crucial for alveolar repair after injury and we tested the effects of IL11 or TGFβ1 on proliferation of **human alveolar epithelial cell**”

Page 11 line 299: “gene regulation in AT2 cells” please either provide data on primary mouse or human AT2 cells or rephrase in “gene regulation in **human alveolar epithelial cell**”.

Page 11 line 311. Supplementary figure 11e is on HSAEC and not ATII, accordingly to the figure legend.

Last, we agree on the decision of the authors of removing all proliferation data from the main figure. Following up on our concerns in using HPAEpiC (point 6), we remain firmly convinced that study proliferation in HPAEpiC, as a surrogate assay for studying the effect of IL11 on proliferation of primary mouse ATII cells, **remains very stretched and poorly informative, if not misleading.**

7) I am ok with the new figure legend.

9) I thank the reviewer for providing references on the used protocol. According to our direct experience, primary mouse ATII cells culture in DMEM supplemented with 10% FBS, are poorly viable if compared to primary mouse ATII cells cultured in other specialized cell culture media, and therefore we still find these results surprising. However, these discrepancies could be partially explained by different isolation methods and therefore we trust the authors on their experimental setup.

We apologize for the confusion on the SFTPC tdT, we agree with the authors on this point. We are ok with new supplementary figure 12.

10) I thank the authors for the reply for providing additional evidence as reported in Fig R3.14

11) I thank the authors for the reply for providing convincing additional evidence as reported in Fig R3.15-16. However, the image included in Figure 4k (4b in the original submission) it has not been changed. Again in this figure, ***“The overlap between Sftpc-tdT and PDPN, in the selected image, is almost 100%, thus suggesting that after Tamoxifen treatment all ATI derives from ATII to ATI differentiation”*** Please include a more representative image such as the one reported in R3.15 animal #2 (Supplementary figure 17b), where there are a lot of PDPN+/ tdT- cells and many PDPN-/ tdT+ cells that still has the small cuboidal shape of ATII cells.

12) Regarding the use of different time points, we concur with the authors' explanation, but we remain puzzled about the rationale behind using day 21 for tissue assessment in the Sftpc-CreER; Il11ra1 fl/fl experiment (Figure 4a) and day 12 in the Sftpc-tdT + IgG/X203 experiment. Considering the clear impact of X203 illustrated in Figure 5b on ATII to ATI transdifferentiation, I am concerned that the limited efficacy observed in Masson's trichrome staining and HP content may stem from the shorter time window (12 days) employed in these experiments. It is conceivable that evaluating lung tissue at 21 days post-bleomycin administration would have yielded a more comprehensive understanding of the phenotype as well as provide data consistency if compared with Figure 4.

I kindly request the authors to provide data (including whole lung Masson's trichrome staining, Ashcroft score, and HP content) on X203 efficacy at 21 days post-bleomycin treatment. If this additional dataset confirms improved tissue regeneration at day 21, it will also assist in addressing points 14 and 15.

13) I appreciate the author for incorporating whole slide histological images in supplementary 13C, which now provide a clearer depiction of the overall histology of these lungs. The Il11ra1 fl/fl lungs appear significantly damaged under basal uninjured conditions, with regions distinctly positive for collagen (blue). Could the authors kindly provide comments on this observation?

A similar concern arises regarding the *Sftpc-CreER;IL11+/+* uninjured control mice in Supplementary Figure 18. The uninjured controls exhibit an HP content that is twice as high as the uninjured control in Figure 4c. Could the authors please comment on this discrepancy?

14) I appreciate the authors' response and their provision of additional data, as depicted in Figure R3.19. However, the uninjured lung appears to exhibit signs of pathological tissue remodeling, rather than appearing truly uninjured (as per image below). Furthermore, the HP levels are higher than those observed in uninjured mice in R3.14 or Supplementary 13c, which already exhibited pathological features (as mentioned previously).

Regarding the effect of X203, the Masson's trichrome staining indicates a comparable loss of alveoli in the BLM+iGG group compared to the BLM+X203 group. Additionally, the authors probed a specific region (highlighted in red) in the BLM+IgG group, which is evidently not representative of the entire lung (as evidenced by regions in green).

Overall, it is possible to appreciate a mild reduction in fibrosis at the histological level (one slides only), reflected by a non-significant trend in HP reduction (n=4), as also confirmed by the authors at line 511 page 18.

Last, authors included only 4 animals per condition, which is by far a very limited number in this type of study. Is sample size supported by power analysis?

15) Please refer to point 11 regarding the AT2 - AT1 trans-differentiation in injured Sftpc-tdT. Regarding the potential impact of X203 on specific subclasses of pathological fibroblasts, the provided images in R3.20 are quite intriguing, and I appreciate the author for including them. However, this secondary effect of X203 on fibroblasts should ideally translate into an even stronger anti-fibrotic effect of X203 on bleomycin-induced lung fibrosis. On the contrary X203 appears less effective, in terms of Masson's trichrome and HP levels, if compared with ATII-restricted IL1ra1 KO model. It's worth noting that the ATII-restricted IL1ra1 KO model should be considered pure model of ATII-restricted IL11 signaling and excluding by definition any effect of IL11 on fibroblasts. Could the authors kindly provide insight into this matter?

12-15) In summary the set of histological and biochemical data produced for the X203 experiment are not convincing and needs the following major revisions:

- 12) For consistency with the data shown in figure 4, I kindly request the authors to provide data (including whole lung Masson's trichrome staining, Ashcroft score, and HP content) on X203 efficacy at 21 days post-bleomycin treatment. If this additional dataset confirms improved tissue regeneration at day 21, it will also assist in addressing points 14 and 15.
- 13-14) Damaged uninjured control lungs. Can author provide and explanation for that? Can author please provide additional images of uninjured control mice?
- 14) Improper image selection for X203 treatment supplementary figure 19c. Are author sure about and effect of X203 at the histological level?
- 15) Different number of animals shown for HP and Ashcroft in figure supp. 19c (4 animals/condition) and IF analysis Figure 5C (6 animals/condition). Can the author provide an explanation for this sample size selection? Is sample size supported by power analysis?

16) I thank the authors for the reply and for providing new images as reported in Fig R3.21

17) I thank the authors for their response and for presenting new data as shown in Fig R3.22. However, it remains challenging to comprehend why a reduction in Collagen I levels following X203 treatment (Figure 5j) is not mirrored by a corresponding decrease in HP levels, as observed in the Sftpc-CreER; Il11ra1^{fl/fl} mice. Could the authors kindly provide insight into this matter?

18) I thank the authors for rephrasing.

19) In my humble opinion this paper provides only evidence regarding the role of IL11 signaling in mouse ATII cells, and therefore I kindly ask the authors to rephrase accordingly.

Additional comments to the new version of the manuscript

1) Page 3 line 88: According to comments raised in point 14, please rephrase "These effects were similarly mirrored by anti-IL11 treatment" into "**These effects were only partially mirrored by anti-IL11 treatment**"

Point by point response to the reviewer's comments.

Reviewer #1 (Remarks to the Author):

The authors have very satisfactorily addressed all my concerns and are to be congratulated for an excellent study.

Response: We are grateful for the supportive and constructive comments that have greatly improved the manuscript.

Reviewer #2 (Remarks to the Author):

The authors have made significant progress in addressing previous concerns; however, additional refinement is necessary in the immunofluorescent and flow cytometric analyses to clarify the results.

The specific comments are as follows:

1. The fluorescent signals for the specified proteins are overly intense across several figures, which obscures the margins of cells and intracellular distribution of the indicated molecules in positive cells. It is advisable for the authors to modulate the fluorescent intensities to mitigate this issue.

Response: We thank the reviewer for his/her very constructive comments and acknowledge the reviewer's concern over the intensity of immunofluorescence markers presented in the manuscript. To reassure the reviewer of the appropriateness of the signal intensities of our IF images, we applied accepted principles to IF image acquisition, including the use of no primary antibody/isotype antibody controls and by capturing our images using the built-in automatically defined under-/overexposure settings on the Leica LAS software. And as standard practice, we applied the same image intensity settings for each particular set of markers between control and injured/treatment samples for each given set of experiments to ensure that the most appropriate representation of the phenotypes were observed.

In certain instances, for example Fig 2b, signal adjustments had to be boosted for PDPN across all groups to ensure that the AT1 cell marker was visible in BLM-injured lungs (Fig 2b) as compared to uninjured controls. The fluorescence signals for alveolar KRT8 expression in Fig 2e and Fig 4d,k and 5b were also appropriately based on the expressed amounts in the airways of uninjured lungs. In other instances, signal intensities of *tdT* had to be fine-tuned to highlight the extremely thin/flattened morphologies of differentiated lineage-traced AT2 cells that were very often undersaturated and poorly visualized, whilst minimizing oversaturated signals from native highly *tdT*-expressing AT2 cells within the same field of view (Fig 5b).

To better highlight the fine tuning of our imaging process, we share examples below of several oversaturated and what we deemed "optimal settings" from our lung IF imaging of *tdT*⁺ cells in our *Sftpc-tdT* reporter mice (Fig.R1A) and for intracellular GFP signal in the lung from IL11-EGFP reporter mice (Fig.R1B). We hope this addresses the reviewer's concerns.

Fig R2.1. Images for fluorescence intensity optimization. Images in the under-/over-saturation view are colored accordingly: oversaturated pixels (in blue); undersaturated pixels (in black); marker negative pixels (in green).

2. The discrimination of *Cldn4*^{high} and *Cldn4*^{low} cells in Figures 2g-i and Supplementary Figure 8 is unclear. The average intensities of *Cldn4* in lung epithelial cells from BLM-treated mice appeared to shift right side compared to those from untreated mice. The authors should compare the histograms of the expression of *Cldn4* in lung epithelial cells from both uninjured and BLM-treated mice. The authors should explain this issue in detail. Incorporating isotype control antibodies in Supplementary Figure 8c would be required. Moreover, It would be beneficial to increase the count of *EPCAM*⁺ cells shown on BLM day 10 in flow cytometry to distinguish GFP-positive cells more clearly.

Response: We apologize for the lack of clarity in distinguishing Cldn4-hi and -low cells and we acknowledge the reviewer's suggestions to improve upon our flow data. We have since re-evaluated our IL11-EGFP flow data and upon consultations with the technicians at our flow cytometry facility, we uncovered that our data had rather low signal/noise ratios of PE-tagged anti-GFP (previous Fig 2g and Supplementary Fig 8). This may have been caused by a faulty PE-channel detector/ laser set up in the particular flow instrument used during the profiling of IL11-EGFP cells, which was recently identified.

In the interest of best practice, we have thus re-performed our entire flow experiment on a new cohort of IL11-EGFP mice using a different flow cytometer at an alternative flow facility, which better delineates anti-GFP-PE signals. These revised results in Fig 2 and Supplementary Fig 8 also include a greater number of EpCAM+ cells to better distinguish GFP+ cells, as suggested by the reviewer. We also now included histogram plots of Cldn4 signals and appropriate Cldn4 FMO controls to better highlight our gating strategy for the delineation of Cldn4 subsets (Fig R2.2). Overall, we found that the majority of GFP+ expression in EpCAM+ cells was confined to the Cldn4-hi population in BLM-treated mice and almost absent in Cldn4-negative or -low EpCAM+ cells. In keeping with our original figure format, we have decided only to show the proportion of GFP+ Cldn4-hi cells in the main Fig. 2, as they represent IL11-expressing transitional cells.

Fig R2.2. (a) Histogram plot of Cldn4 signal. (b) FMO controls and gating strategy for delineating Cldn4 subsets. (c) Flow cytometry analysis of GFP expression in Cldn4-neg, Cldn4-low and Cldn4-hi cell populations from L11^{EGFP} reporter mice post-BLM injury. (d) Proportions of EpCAM⁺ Cldn4^{hi} cells or (e) the proportion of EpCAM⁺ GFP⁺ cells across Cldn4-negative, low and hi subsets in the lungs of uninjured or BLM-injured L11^{EGFP} reporter mice.

3. There is a discrepancy between FACS and IHC data. The authors should carefully calculate and interpret the results in Figure 2g and FigR1.1 regarding IL11-expressing epithelial cells. According to our calculation, Cldn4^{low} GFP⁺ cells and Cldn4^{high} GFP⁺ cells comprise 0.64% (0.87 x 0.73%) and 0.87% (0.129 x 6.73%) among all epithelial cells, respectively. Thus, the authors should reanalyze other data and create a new Figure to

include and compare accurate percentages of Cldn4^{high} GFP⁺ and Cldn4^{low} GFP⁺ cells among all epithelial cells. This reanalysis should address the inconsistencies and ensure the results are accurately presented in the context of the study.

Similarly, data from Figure 2e show that IL-11-GFP⁺ cells do not express KRT8, which further implies that a subset of EpCAM⁺Cldn4^{low} cells also exhibit GFP expression.

Response: We apologize for any discrepancies in our earlier flow data that may have arisen as a result of the equipment issues that are outlined in our response to point 2 above. As suggested by the reviewer, we have now included a graph comparing the percentages of GFP-expressing EpCAM⁺ cells in Cldn4^{hi}, Cldn4^{low} or Cldn4⁻ negative subsets from our new data (Fig R2.3). We have also included a revised plot of GFP⁺ EpCAM⁺ cells from the new data. We hope that our revised flow data now clearly shows that Cldn4^{hi} transitional epithelial cells are the predominant IL11-expressing epithelial cell type in the injured lung.

Fig R2.3. Flow cytometry analysis of EpCAM⁺ GFP⁺ cells in the lungs of uninjured or BLM-injured L11^{EGFP} reporter mice. (b) The total proportion of EpCAM⁺ cells constituted by the various GFP-expressing Cldn4 subsets.

With regards to the reviewer's comment that EpCAM⁺ Cldn4^{lo} cells express GFP, our new and revised flow data above do not support this case, we apologize for any confusion caused by our previous findings that must be viewed in the light of the equipment issues we identified. Additionally, the cellular identities of GFP⁺KRT8⁻ cell types cannot be directly inferred from the basis of KRT8 staining alone as additional GFP⁺KRT8⁻ signals may represent other abundant IL11-expressing cells such as fibroblasts. Nonetheless, we hope that our revised data now clearly indicates that both KRT8⁺ and Cldn4⁺ transitional cells are prominent sources of IL11 in the damaged alveolar epithelium.

4. The authors should provide a detailed method for how the protein intensity calculations were performed in Figures 3e and i, specifying whether they were conducted across different areas within a single well or multiple wells.

Response: We apologize for the lack of clarity in the methods description of the immunostaining quantification in Fig 3e and i. To elaborate: each biological experiment and each treatment condition was run in duplicate wells, and 7 to 14 fixed non-overlapping fields were imaged and analyzed per treatment group. Quantification of immunofluorescence in HPAEpiC and HSAEC experiments on the Operetta high throughput imaging system was performed using the built-in cell analysis tool on the Columbus software (version 2.7.2, PerkinElmer). To determine the fluorescence intensities for each cell, individual cells were denoted based on the DAPI nuclei staining and cell area were established based on total Alexa Fluor 488 signal. Fluorescence intensities of cytoplasmic Alexa Fluor 488 signals within each demarcated cell area were concurrently measured and fluorescence intensities for each cell were then further normalized to their respective area. Mean intensity/area for each analyzed field was then presented as one datapoint each.

To further elaborate, for Fig 3e, one representative dataset from three independent biological experiments are shown (mean fluorescence intensity/area for 14 non-overlapping fields per condition are shown). For Fig 3i, one representative dataset from two independent biological experiments are shown (mean fluorescence intensity/area for 7 non-overlapping fields per condition are shown). Similar results were obtained between biological experiments. We have now updated the figure legends accordingly.

5. It would be instructive for the authors to calculate the percentages of collagen I+ cells among Sfpcc+ cells under various conditions, and show in Supplementary Figures 11g and h.

Response: We appreciate the reviewer's suggestion and have now included new quantification of Collagen I staining intensity/area for mouse AT2 cells in Supplement Fig. 11.

Fig R2.4. Quantification of Collagen I immunostaining intensity over cell area in Sfpcc-tdT AT2 cells.

6. In Figures 5e and f, the Cldn4+ and PDPN+ cells do not form distinct populations. To resolve this, an increase in the number of analyzed EpCAM+ cells via flow cytometry is recommended to delineate these cell populations more effectively.

Response: We acknowledge the reviewer's point and have performed new experiments to analyze increased numbers of EpCAM+ cells by adding two extra mice per treatment group. We now also show our specific gating strategy for the delineation of Cldn4 subsets and of PDPN expressing cells based on FMO controls. Our new results are in line with our previous data and our overall interpretation of the results remains unchanged. We hope that our new plots (in revised Fig 5 and Supplementary Fig 20) more effectively delineate and showcase Cldn4-hi and PDPN+ cells in our therapeutic X203 study.

Fig R2.5. Flow cytometry analysis of *Cldn4* and *PDPN* expression in CD45- Cd31 EpCAM⁺ tdT⁺ cells from *Sftpc-tdT* mice treated with IgG/X203 antibodies after BLM-injury. **(a)** Representative fluorescence minus one (FMO) controls for gating of *Cldn4*⁺ signal. **(b)** Gating strategy for *Cldn4*^{hi}, *Cldn4*^{low} and *Cldn4*^{neg} cells based on *Cldn4* signal intensities from CD45-CD31-EpCAM⁺tdT⁺ cells from uninjured mice. Quantification of lung **(c)** EpCAM⁺ tdT⁺ *Cldn4*^{hi} cells and **(d)** EpCAM⁺ tdT⁺ *PDPN*⁺ cells. *n* = 4 - 5 mice / group. Data are represented as mean ± s.d. *P* values were determined by one way ANOVA (Tukey's test).

7. The expression patterns of *IL11ra1* in different types of epithelial cells in mice remain unclear. In addition, Figure 5g and Supplementary Figures 15 and 22 should include the characteristic genes of each identified cell cluster for better clarity.

Response: The reviewer makes an excellent suggestion here to include profiles of mouse *Il11ra1*. We now provide analysis of *Il11ra1* expression across all lung cell types across three separate publicly available mouse lung scRNA-seq datasets (Fig R2.6, revised Supplementary Fig. 13). Consistent with our human scRNA-seq analysis of *IL11RA*

in Supplementary Fig 3, our mouse scRNA-seq analysis similar shows that stromal cells such as fibroblasts and smooth muscle cells express the highest levels of *Il11ra1* and less abundantly expressed by alveolar epithelial cells and macrophages in the mouse lung. In general, immune cells express the least levels of *Il11ra1* which is again consistent with the human data. In addition, we have already presented a dot plot for the characteristic genes for each identified cell cluster for our mouse scRNA-seq experiments in Supplementary Figure 15b (Fig R2.7 below), which the reviewer may have missed.

Fig R2.6. Dot-plots showing the expression of *Il11ra1* across various mouse lung cell types in scRNA-seq data from Strunz et. al. (GSE141259), Ogawa et al. (GSE184854) and Joshi et al. (GSE127803) datasets.

Fig R2.7. Dot plot showing the expression of various cell identity markers as shown in Supplementary Figure 15b.

8. Moreover, a recent study reported that IL-11 induced by TGF β does not play a crucial role in TGF β -induced fibrosis (Tan et al., Front Immunol 2024). Thus, the authors should discuss this point in detail in the Discussion.

Response: We understand this suggestion to discuss the recent manuscript by Tan et. al. (2024) in Front. Immunol. We must say that we were surprised by their results which seem largely at odds with our group’s data and many others in the field both academic and commercial institutions, in particular with regards to their BLM and in vitro NHLF data. We would highlight that we do not think the authors used appropriate cells for their in vitro studies, as

we highlighted in a manuscript entitled “Critical Conditions for Studying Interleukin-11 Signaling In Vitro and Avoiding Experimental Artefacts” (PMID: 34570432).

To summarize, the Tan study found upregulated *IL11* RNA expression (but did not investigate protein expression) by in situ hybridisation and qPCR in tissues from patients with idiopathic pulmonary fibrosis (IPF), scleroderma, and inflammatory bowel disease (IBD), a finding that recapitulates previous studies performed by us for IL11 expression in IPF (our study; PMID: 31554736), any by others in skin and lung fibroblasts in systemic sclerosis (others; PMID: 23915349 and PMID: 29853453), and in human colon tissue with ulcerative colitis (others; PMID: 31348891). They also saw an elevation of secreted IL11, and concomitant STAT3 activation, from epithelial cells and fibroblasts by TGF-beta in vitro, which is in line with our studies and many others in the field. Taken together, in different human fibrotic disease states, it is convincing that IL11 gene and/or protein signature is upregulated which they and we can agree upon.

Where our results differ from theirs are outlined in 2 main areas:

(1) they saw no convincing trend in *Il11* gene upregulation in a mouse model of BLM-induced lung fibrosis (only a slight but significant increase at day 14 and no increase at day 21) despite a progressive increase in lung fibrosis as assessed only by Ashcroft scoring (with no HPA data or supporting lung histology images provided), no change in IL11 protein expression by ELISA on tissue homogenates (rather than performing complementary immunoblots which may be considered the gold standard for the robust detection of lowly expressed cytokines) (Figure 1D in PMID: 38455057), and therefore they decided to not proceed with anti-IL11 experiments in animals.

We also believe that we have provided a far more comprehensive analysis of IL11 changes in the BLM model to date. In our previous work (PMID: 31554736), BLM-induced lung fibrosis was assessed by two key established methods, Ashcroft scoring (subjective pathological assessment) and lung tissue HPA assay (unbiased surrogate assessment of tissue collagen). Progressive IL11 upregulation across day 7-21 in the BLM model was confirmed by immunoblots which was strongly linked to increased ECM proteins and lung fibrosis progression. Additionally, we have also established that *Il11* RNA (by bulk lung RNAseq which is unbiased as compared to selective qPCR) was significantly upregulated in the lung on the genome-wide level 21 days after BLM-injury (PMID: 31554736), which also contradicts their qPCR results.

We have also provided multiple sets of consistent evidence primarily from loss-of-function studies in vivo with global *Il1ral*-KO and *Il11*-KO mice (PMID: 34239012), fibroblast- (PMID:32656894) and AT2 cell-specific *Il1ral* deleted mice (this manuscript) and as well as with pharmacological studies with X203 that confirms a profibrotic role of IL11 in the BLM-model. More generally, we have also shown that X203 is also extremely effective in attenuating fibrosis across multiple organ systems as well (kidney, liver and heart-injury models), which they do not show for their candidate anti-IL11 antibodies.

Furthermore, and critically important, our work on IL11 in lung fibrosis have been replicated in studies by many other groups in the field (*siIL11* treatment in BLM model: PMID 35731866; PMID 36376885, PMID: 38679415; unspecified anti-IL11 (R&D systems) in BLM model: PMID: 36166195). Excluding a solely BLM-dependent effect on murine lung fibrosis, other IL-11 neutralizing antibodies, such as the less efficacious MAB418 (R&D systems) compared to X203, has also shown significant efficacy in the silica particles-induced lung inflammation and fibrosis model as well (conducted by others: PMID: 36503802). The authors could have used MAB418 as a comparator in their various assays, which would have greatly increased our confidence in their data.

Lastly, we and several other groups have also found that chronic recombinant mouse IL-11 administration subcutaneously to mice, as a gain-of-function model, was sufficient to promote lung fibrosis and pulmonary hypertension (PMID: 31554736, PMID: 36376885, PMID: 38679415). Therefore, we think that the lack of IL11 upregulation they observed in their BLM-model is likely a result of differing technicalities employed. The reasons

for their choice of performing ELISA on tissue homogenates rather than immunoblots is also unclear. There is also a lack of information regarding the source of the IL11 ELISA kit used in their study.

(2) Tan et al. saw no myofibroblast activation (by α SMA and COL1A1 immunostaining) following IL11 stimulation (Figure 5 in PMID: 38455057) and no inhibitory effects of their neutralizing IL11-antibody candidate (5A6.2) after TGF β stimulation in lung fibroblasts despite their evidence of pSTAT3 inhibition (Figure 6 in PMID: 38455057). This is in spite of the similar binding affinity to IL11 in Biocore kinetic assays as X203, at single digit nanomolar dissociation constant (K_D), against human and mouse IL11 as we have previously demonstrated for antibody development (PMID: 31554736; Fig S7 in Supplementary Materials).

There are several issues with their claims that their antibody are of similar efficacies:

1. Similar affinity binding capability does not necessarily equate to similar neutralizing functions. Affinity binding refers to the measure of strength of an antibody to the antigen. However, an antibody can bind strongly to an epitope but yet not necessarily neutralize the ability of the antigen to bind its cognate receptor and inhibit proper down-stream signaling for a particular biological effect. During antibody development, clones are often compared head-to-head for binding affinity and neutralization ability (usually by a series of in vitro methods) before selection of the best/better antibody clone for further development. This crucial information was not demonstrated here as to how their clones were selected for/against.
2. They showed that their antibody was able to neutralize pSTAT3 activation using the RAW264.7 (an immortalized macrophage cell type) pSTAT3 luciferase reporter assay, which at least suggest some neutralization capabilities of their clone for this particular pathway. However, we (and others) have previously shown that IL11 predominantly elicits its profibrotic effects via ERK signaling pathway (PMID: 29160304, PMID: 31959867, PMID: 36166195, PMID: 36052698), whereas STAT3 activity which is acutely triggered by IL11 (than ERK) predominantly drives inflammation responses in fibroblasts (shown in our recent study PMID: 36012165). We have also recently demonstrated in a series of in vitro assays and genetic *Il1ra1* null murine fibroblasts, that the profibrotic effects of IL11 are largely ERK-driven and at the post-transcriptional and protein translational level (shown in our recent separate study PMID: 34651016). It is currently unknown whether their antibody clones can inhibit ERK pathway.
3. With regards to their in vitro 5A6.2 antibody data on TGF-beta stimulated fibroblasts. They only showed data that 5A6.2 failed to reduce TGF-beta-induced secretion of TIMP1 and CTGF levels but ultimately failed to include any data on the two critical pathological myofibroblast markers α SMA and Collagen I. This is surprising and potentially misleading given that they had shown very strong TGF-beta responses for both of these key markers in a separate gain-of-function experiment (Fig 5).
4. There is also no detailed information on passage number of NHLF used in their in vitro assays. We have previously published on the critical culture conditions required for studying IL11 signaling pathways in vitro (PMID: 34570432). To note, the specific IL11RA is highly expressed on low passage fibroblasts but expressed at much lower levels when these cells are highly passaged through prolonged culture. For instance, we have found that IL11 does not stimulate the upregulation profibrotic markers such as α SMA, COL1A1, MMP2 and TIMP1 in high passage human lung fibroblasts (passage >5) to the same degree as in low passage cells (passage \leq 3). Besides, high passage lung fibroblasts also secrete much lower levels of IL11 following TGF-beta treatment as compared to low passage cells which can further mask physiologically relevant auto/paracrine effects.
5. As we have concluded in PMID: 34570432, “in vitro experiments with primary cell material that look at IL11-signaling, and with most cytokine-mediated effects in general, need to be planned and executed with great caution. Otherwise, physiologically relevant mechanisms may not be obtained and reproducible experimental artifacts can obscure our view of true cytokine biology”.

In light of the lack of information and differing technical aspects performed in the manuscript by Tan et. al., we do not think it scientifically correct to discuss their findings further as they differ greatly (likely due to technical artifacts, we would argue) from the general consensus and largely reproduced findings of IL11 biology in lung fibrosis in the current literature and in preclinical POC that underpins three phase 1 clinical trials for lung fibrosis. We hope the reviewer understands our points on this matter.

Reviewer #3 (Remarks to the Author):

Dear Authros,

please refer to the attached Pdf document that includes a point by point reply to the rebuttal letter.

General

I would like to express my gratitude to the authors for their thorough response to the comments and for the extensive addition of new data in the revised version of the paper leading to a significant overhaul of the manuscript, which includes two main figures that are largely new (Figures 3 and 4) as well as the new set of data on the Sftpc-CreER; Il11fl/fl mice reported in supplementary 12. However, there are still some aspects that, in my opinion, need further clarification.

The EPCAM+ IL-11+ cell population is exceedingly rare, and its biological relevance has been extensively evaluated in this study, yielding convincing results regarding the pathological role of IL11 in alveolar epithelial cells. The authors have provided extensive results from lineage-traced ATII cell experiments, demonstrating that ATII-restricted IL1ra1 knockout promotes ATII to ATI transdifferentiation, thereby supporting alveolar repair following bleomycin injury in mice. This effect is appreciable both histologically and biochemically (HP) in experiments utilizing the IL1ra1 knockout mouse model.

It is a common consensus in the field that IL11 represents a valuable target for the development of anti-IL11 therapies for IPF either using antibodies targeting IL11 or its receptor. Hence, the authors of this manuscript, who have extensive background in anti-IL11 treatment for cardiac fibrosis are understandably motivated to demonstrate the efficacy of the anti-IL11 treatments in other fibrotic diseases, such as IPF.

In my assessment, upon reviewing histological images and HP data, the efficacy of X203 appears less pronounced compared to the ATII-restricted IL1ra1 KO in the bleomycin-induced lung fibrosis model. This well fit the idea that genetic ablation of the receptor is more effective than pharmacological inhibition of the ligand.

What still remains counterintuitive, are the disparities in the impact of X203 on restoring ATII to ATI transdifferentiation, which, overall, seems convincing, yet demonstrates limited effectiveness in reducing lung fibrosis. Although there seems to be a reduction in lung fibrosis based on histological examination (only one whole slide shown), this reduction lacks significance in terms of HP contents, given the small sample size of only 4 animals per condition, which is a very limited number for this type of study. Moreover, lungs treated with X203 exhibit evident signs of tissue remodeling, including alveolar loss, when compared to the IL11 knockout condition (once again, only one slide shown) (Supplementary Figure 19). Considering the striking impact of X203 demonstrated in Figure 5b in promoting ATII to ATI transdifferentiation, I am apprehensive that the limited efficacy observed in Masson's trichrome staining and HP content may be attributed to the shorter time window (12 days) utilized for this series of experiments. It is plausible that evaluating lung tissue at 21 days post-bleomycin administration would have provided a more fully resolved phenotype as a long-waved consequence of the improved ATII to ATI trans-differentiation capacity.

Last, all the in vivo experiments in mice, despite the use of different genetic strains presents uninjured control lungs, that looks all but not uninjured, thus eliciting the possibility of concomitant lung infections or generally not very well controlled experimental conditions in the bleomycin-mouse model. It is recognized in the field that the bleomycin model of lung fibrosis needs a very careful optimization to be informative and reproducible as well as

adequate numbers of animal in each experimental group to balance the intrinsically high variability of the model.

Here below it follows a point-by-point discussion of authors' rebuttal letter, including additional request to the authors.

1) I appreciate their inclusion of flow cytometry analysis using an anti-GFP antibody instead of relying solely on the endogenously encoded EGFP (Figure R3.1). However, we have some concerns about the following sentence:

“With regards to the reviewers’ comment on performing IL11-targeted antibody IF experiments. We have already provided immunostaining of IL11 in AT2 lineage traced cells in Sftpc-tdT mice in original Fig 2h and Supplementary Fig 5c which the reviewer might have missed. We showed that anti-IL11 staining is co-localized to subsets of lineage-labeled (SftpcTdT) AT2 cells after BLM-injury, which indicates that activated / differentiating AT2 cells may be potential sources of IL11 in the injured lung. Unfortunately, none of the anti-IL11 antibodies that we have tested were specific for flow cytometry analysis and hence we are unable to provide further quantification of IL11+ cells in the injured lung.”

“Could authors please confirm their findings by showing an IF experiment using an antibody targeting IL-11 instead of the genetically encoded reporter gene? ”

Here my request was very simple, and I'm sorry if I wasn't clear on this point. I was actually asking for an IF image of Uninjured vs Bleo-injured lungs of IL11-EGFP mice stained for IL11/SFTPC/IL11-EGFP and DAPI (which is different from what authors included in Supplementary Figure 7g (EGFP-IL11) and h (SFTPCtdT-IL11-KRT8)). Could the author please provide the reviewer with this set of images?

I was and I am still concerned that the rarity of events shown in Fig 2B doesn't match with the abundance of cells reported in figure 2C and highlighted by “White arrowheads indicate marker positive IL11EGFP+ cells.” There are 10 highlighted objects (IL11/SFTPC) in a field of view with ≈200 total cells, out of which epithelial cells are a subpopulation inclusive of the SFTPC cells. Therefore, it results in a % of IL-11-positive lung epithelial cells that has to be higher than 4% (10/250), which is however 25 folds (4/0.150) more than the one showed in figure 2A by cytofluorimetry. For this specific reason we asked to stain tissue slides with anti IL-11 and anti SPC antibody to assess if the events labelled by the genetically encoded IL-11-EGFP were truly IL-11 expressing cells. It's a simple experiment and we would have really appreciated if the authors have included It in the supplementary figures.

Could author please either comment on this point, or include a better representative image for figure 2b?

Response: We greatly appreciate the reviewer's thorough assessment of our revised manuscript and for the additional constructive comments. To clarify on the above points, we first highlight that IL11-GFP expression is not observed in uninjured lungs and also almost entirely absent in non-fibrotic regions in BLM-treated mice. Since it is well established that BLM injury in mice causes patchy lung inflammation and fibrosis, we have chosen to focus our histology assessment and representative images of IL11-expressing cells only in injured regions in Fig 2, as showing its expression in non-injured regions would just be similar to uninjured controls. It is also expected that flow cytometry-based quantification of EpCAM+GFP+ cells in whole lung single cell suspension of injured mice, which consists of epithelial cells from both injured and uninjured regions, will inevitably show a much lower % of GFP+ cells as compared to the histology assessments.

We are also aware that the detection and distinction of endogenous EGFP-expressing cells by flow cytometry (Fig 2b) is highly challenging due to high noise/signal ratios of highly autofluorescent lung cell populations. We have since performed new flow cytometry analysis of fixed cells from IL11-EGFP mice using anti-GFP antibodies (Supplementary Fig 8) which revealed that ~1.7% of EpCAM+ cells express GFP after BLM-injury (Fig R3.1). Importantly, we also found that Cldn4-hi transitional epithelial cells are the predominant GFP-expressing epithelial cell subset in the injured lung (Fig R3.2). We hope that this now provides a better representation of the small but significantly increased proportion of IL11-expressing epithelial cells in the injured mouse lung. For these reasons, we have now moved the preliminary data shown in Fig 2b into the supplement to better focus Fig 2 on the main theme of IL11-expressing Krt8/Cldn4 transitional cells.

Fig R3.1. Flow cytometry analysis of EpCAM⁺ GFP⁺ cells in the lungs of uninjured or BLM-injured L11^{EGFP} reporter mice.

Fig R3.2. (a) Histogram plot of Cldn4 signal. (b) FMO controls and gating strategy for delineating Cldn4 subsets. (c) Flow cytometry analysis of GFP expression in Cldn4^{neg}, Cldn4^{low} and Cldn4^{hi} cell populations from L11^{EGFP} reporter mice post-BLM injury. (d) Proportions of EpCAM⁺ Cldn4^{hi} cells or (e) the proportion of EpCAM⁺ GFP⁺ cells across Cldn4-negative, low and hi subsets in the lungs of uninjured or BLM-injured L11^{EGFP} reporter mice.

With regards to the reviewer's request for IL11 + SFTPC + GFP co-staining in our IL11-EGFP reporter mice. We regret to inform the reviewer that this is not possible as we do not have the appropriate species-compatible multiplexing antibodies to do this effectively (i.e both anti-IL11 and anti-SFTPC are both rabbit primary antibodies). Hence we have only attempted to provide separate staining in the lungs of BLM-injured IL11-EGFP mice for either : 1) rabbit anti-IL11 + goat anti-GFP (which showed considerable overlap) or 2) rabbit anti-SFTPC + goat anti-GFP (which indicates IL11-expression in AT2-lineage cells). As a complement to these data, and described above, our new and improved flow cytometry analysis of IL11-EGFP lung epithelial cells demonstrated that Cldn4-hi expressing transitional cells are the predominant IL11-GFP expressing cell type in the BLM-injured lung epithelium.

We also further point out that alveolar KRT8 cells can retain SFTPC expression during early differentiation, as previously demonstrated by Tata's 2020 study (PMID: 32661339; Extended Fig 2), and also observed by us (Fig R3.3 below). This further supports our work and shows that SFTPC + GFP data in BLM-injured IL11-EGFP mice likely represent the upregulation of IL11 expression by a subset of activated / transitional AT2 cells, which we later show with KRT8 + GFP staining. Taken together, our histology and complementary flow data indicate that IL11 is specifically upregulated by activated AT2 cells and most profoundly expressed by Krt8+ and Cldn4-hi transitional cells in the injured mouse lung epithelium.

Fig R3.3. Image of immunostaining for KRT8 and SFTPC in lungs of Sftpc-tdT mice 7 days post-BLM injury. KRT8+SFTPC+tdT+ cells are indicated by yellow arrowheads.

2) I am ok with figure R3.2. and supplementary figure 7 regarding the stitched images.

3) I thank the authors for the reply and for providing additional evidence, and I'm ok with Fig R3.3.

4) I thank the authors for providing additional images as requested. I'm ok with the images reported in R3.4 and 3.5, and for incorporating R3.4 image in Figure 2.

5) I thank the authors for the reply and for providing additional evidence as requested. I agree on including R3.6 B in main figure 2 k.

Response: We thank the reviewer for his/her acknowledgments and agreement with our response for points 2-5.

6-8) I thank the authors for the reply. I align with the authors on the KRT8-hi presence in human lungs as reported by Kobayashi and Strunz (2 and 9 of rebuttal letter), and also Jiang- Zemans 2020 in AJRCCM "Ineffectual Type 2-to-Type 1 Alveolar Epithelial Cell Differentiation in Idiopathic Pulmonary Fibrosis:

Persistence of the KRT8hi Transitional State”. I’m ok with the new SFTPC staining, that now resembles much better SFTPC staining usually seen in primary ATII cells. I also agree on the rewording of “Cytopatic”.

In my humble opinion the use of HPAEpiC and HSAEC cells, simply doesn’t add any value to the story. Both the Strunz and Kobayashi’s papers, as well as this manuscript, largely model the KRT8+ cells in IPF pathology using mouse models, and therefore the obvious follow-up set of experiments, after figure 2, should have included only data on primary mouse ATII cells or human primary ATII cells, if the aim was to show efficacy in human Cells.

In alignment with our opinion, the data shown in Figure 3E on KRT8 (intensity/area) are quite different if compared with Figure 3I KRT8 (intensity/area). In particular, the difference between TGFb1+IgG and TGFb1+X203 are partially, if not completely, blunted in primary mouse ATII cells.

Consequently, I appreciated the decision of the authors to include some data on mouse primary ATII cells in Figure 3 J I K and supplementary Fig 11 F-H. However, I kindly ask to the authors to include in figure 3 the IF images present in Fig 11 F-H and a violin plot corresponding to the systematic quantification of Collagen1(intensity/area) in primary ATII cells, thus generating a similar set of data as the one shown in figure 3E-I for HPAEpiC and HSAEC respectively

Page 9 Line 262: “AT2 cell proliferation is crucial for alveolar repair after injury and we tested the effects of IL11 or TGFβ1 on AT2 cell proliferation” please either provide data on primary mouse or human AT2 cells or rephrase in “AT2 cell proliferation is crucial for alveolar repair after injury and we tested the effects of IL11 or TGFβ1 on proliferation of human alveolar epithelial cell”

Page 11 line 299: “gene regulation in AT2 cells” please either provide data on primary mouse or human AT2 cells or rephrase in “gene regulation in human alveolar epithelial cell”.

Page 11 line 311. Supplementary figure 11e is on HSAEC and not ATII, accordingly to the figure legend.

Last, we agree on the decision of the authors of removing all proliferation data from the main figure. Following up on our concerns in using HPAEpiC (point 6), we remain firmly convinced that study proliferation in PAEpiC, as a surrogate assay for studying the effect of IL11 on proliferation of primary mouse ATII cells, remains very stretched and poorly informative, if not misleading.

Response: Firstly, as discussed in our limitations previously, large scale high throughput IF studies of monocultures of mouse AT2 cells were not possible due to their inherent lack of attachment and viability issues. Hence, we have based our main in vitro findings on primary human cells (HPAEpiCs and HSAECs , which both do not have these culture issues) whilst attempted to also show, albeit in a more limited fashion, that IL11 consistently promotes EMT-like features in primary mouse AT2 cells.

With regards to the reviewer’s concerns over the differences in KRT8 expression in HPAEpiC and mouse AT2 cells. We do not fully understand the reviewer’s points here. We show that X203 significantly reduced TGF-beta-induced KRT8 expression in HPAEpiC (at 24 hours) and mouse AT2 cells (at 5 days) and that the differences in KRT8 intensities may have varied due to different assay conditions and timepoints.

As suggested by the reviewer, we have now included Collagen I intensity analysis for mouse AT2 cells (Fig R3.4) and have incorporated these data in the Supplementary Fig. 11. Further, we have also moved the images of Collagen I expression in mouse tdT+ AT2 cell into main Fig 3, in accordance with the reviewer’s suggestion.

Fig R3.4. Quantification of Collagen I immunostaining intensity over cell area in *Sftpc-tdT* AT2 cells.

We agree with the reviewer and have now rephrased lines 262 and 299 according, we thank the reviewer for his/her input:

“AT2 cell proliferation is crucial for alveolar repair after injury and we tested the effects of IL11 or TGFβ1 on proliferation of human alveolar epithelial cell”

“gene regulation in human alveolar epithelial cell”.

As for line 311, we have now reworded the sentence to better reflect on the actual results for each cell type:

“By immunostaining, we observed that IL11 and TGFβ1-treatment significantly increased the expression of Collagen I and fibronectin and secreted collagen by HSAEC and Collagen I expression in mouse AT2 cells”

7) I am ok with the new figure legend.

9) I thank the reviewer for providing references on the used protocol. According to our direct experience, primary mouse ATII cells culture in DMEM supplemented with 10% FBS, are poorly viable if compared to primary mouse ATII cells cultured in other specialized cell culture media, and therefore we still find these results surprising. However, these discrepancies could be partially explained by different isolation methods and therefore we trust the authors on their experimental setup.

We apologize for the confusion on the SFTPC tdT, we agree with the authors on this point. We are ok with new supplementary figure 12.

10) I thank the authors for the reply for providing additional evidence as reported in Fig R3.14

Response: We thank the reviewer for his/her acknowledgment for our response to point 7, 9 and 10.

11) I thank the authors for the reply for providing convincing additional evidence as reported in Fig R3.15-16. However, the image included in Figure 4k (4b in the original submission) it has not been changed. Again in this

figure, “The overlap between Sftpc-tdT and PDPN, in the selected image, is almost 100%, thus suggesting that after Tamoxifen treatment all ATI derives from ATII to ATI differentiation” Please include a more representative image such as the one reported in R3.15 animal #2 (Supplementary figure 17b), where there are a lot of PDPN+/ tdT- cells and many PDPN-/ tdT+ cells that still has the small cuboidal shape of ATII Cells.

Response: We acknowledge that our original enlarged images of *Sftpc-tdT;Il11ra1fl/fl* mice were composed mostly of flattened tdT+PDPN+ cells, which we believe was reflective of greatly enhanced regeneration. Nonetheless, we have now swapped the image in the main figure 4k with the ones in the supplementary and included enlarged images which now show the abundance of newly formed tdT+PDPN+ AT1 cells in close proximity to several native cuboidal shaped tdT+ AT2 cells within the same field of view (Fig R3.5). We hope that these new selections are now appropriate to the reviewer.

Fig R3.5. Replacement images of *Sftpc-tdT;Il11ra1fl/fl* mice after bleomycin injury in Fig 4k.

12) Regarding the use of different time points, we concur with the authors' explanation, but we remain puzzled about the rationale behind using day 21 for tissue assessment in the Sftpc- CreER; Il11ra1fl/fl experiment (Figure 4a) and day 12 in the Sftpc-tdT + IgG/X203 experiment. Considering the clear impact of X203 illustrated in Figure 5b on ATII to ATI transdifferentiation, I am concerned that the limited efficacy observed in Masson's trichrome staining and HP content may stem from the shorter time window (12 days) employed in these experiments. It is conceivable that evaluating lung tissue at 21 days post-bleomycin administration would have yielded a more comprehensive understanding of the phenotype as well as provide data consistency if compared with Figure 4. I kindly request the authors to provide data (including whole lung Masson's trichrome staining, Ashcroft score, and HP content) on X203 efficacy at 21 days post-bleomycin treatment. If this additional dataset confirms improved tissue regeneration at day 21, it will also assist in addressing points 14 and 15.

Response: We acknowledge and appreciate the reviewer's comments here. However, we have already published extensively on the effects of X203-treatment on fibrosis in the BLM model at longer 21 and 28 day time points in two of our previous manuscripts (PMID: 31554736, PMID: 32656894). Hence, we do not believe that additional data of fibrosis readouts at these later time points would greatly benefit the current study which primarily focuses on AT2 differentiation phenotypes that are best captured early. Also, as mentioned in the previous rebuttal, we focused on optimal early time points (<14 days) across our histology and scRNA-seq experiments to best capture transitional epithelial cell phenotypes, and also provided supplemental fibrosis data at these earlier time points (day 12) in supplementary fig 14 and 19. And as previously mentioned, we included Day 21 time point for *Sftpc-CreER;Il11ra1fl/fl* experiments to assess fibrosis outcomes in this novel genetic strain and as part of our routine

survival studies which ends at day 21. To improve clarity on this matter, we have now included a sentence in the text (for Fig 5) to introduce these previous findings:

“In our previous therapeutic studies, we showed that X203-treatment significantly diminished lung inflammation and reversed established lung fibrosis in BLM-injured mice”

Nonetheless, for completeness, we have conducted a small day 21 study on our remaining but limited numbers of *Sftpc-tdT* mice in our colony ($n = 2$ mice / group). We treated these mice with X203/IgG every 2 days starting 7 days post-BLM and again focused on epithelial phenotypes by staining the lungs for PDPN and KRT8 (Fig R3.6). As anticipated, we observed greatly enhanced AT2-AT1 differentiation (PDPN+tdT+ cells) and markedly reduced lineage traced transitional cells (KRT8+tdT+ cells) and reduced fibrotic lesion sizes (as indicated by regions of dense DAPI accumulation) in X203 as compared to IgG treated mice (Fig R3.6). Overall, tissue injury is markedly reduced and alveolar regeneration is greatly enhanced by X203 in the day 21 model which are largely consistent with the phenotypes observed at day 12.

Lastly, we regret to inform the reviewer that we do not currently have sufficient *Sftpc-tdT* animals available to us to expand on these observations for statistical quantitative readouts of regeneration and we have decided to include these new data in a new Supplementary fig. 22 as an extension of our findings in Fig 5.

Fig R3.6. (a) Schematic showing the administration time points of BLM and X203 or IgG antibodies in *Sftpc-tdT* mice. Lung tissues were assessed 21 days post-BLM challenge. (b) Representative images of immunostaining for KRT8 and PDPN in injured regions of lungs from BLM-injured *Sftpc-tdT* mice treated with X203 or IgG antibodies at day 21. Scale bars: 100 μ m. (c) The proportions of KRT8+ tdT+ cells or PDPN+ tdT+ cells divided by the number of tdT+ cells in the injured lung regions ($n = 2$ mice / group).

13) I appreciate the author for incorporating whole slide histological images in supplementary 13C, which now provide a clearer depiction of the overall histology of these lungs. The *Il11ra1f1/fl* lungs appear significantly damaged under basal uninjured conditions, with regions distinctly positive for collagen (blue). Could the authors kindly provide comments on this observation?

*A similar concern arises regarding the *Sftpc-CreER;IL11+/+* uninjured control mice in Supplementary Figure 18. The uninjured controls exhibit an HP content that is twice as high as the uninjured control in Figure 4c. Could the authors please comment on this discrepancy?*

Response: Regarding the histology images of uninjured *Sftpc-CreER;Il11ra1fl/fl* lung in Supplementary 13C that may have appeared damaged. The reviewer has in fact highlighted (in red) regions that showed compressed alveolar structures as potential artifacts from tissue fixation / histology processing, and not of overt damage/pathology. The light blue stainings represent baseline lung collagen levels in the cohort, which we further discuss in point 14 below. To be clear, we do not inflate the lungs of *Sftpc-CreER;Il11ra1fl/fl* mice with formalin during the fixation process as we routinely collect other lung lobes from the same mouse for other analyses. This may have caused regions of collapsed architecture especially in proximal zones during fixation, and other macro-scaled lung images from this uninjured cohort contained similar artifacts and were not suitable as replacement images. Additionally, as a precaution, histology analyses (ashcroft) were also not performed on these affected regions with clear artifacts.

To further clarify on HPA data in Supplementary Fig. 18, the entire right lung of *Sftpc-CreER;Il11fl/fl* mice was used for HPA content determination as compared to just the right caudal lobes for *Sftpc-CreER;Il11ra1fl/fl* mice. This explains the overall increase in HPA content for *Sftpc-CreER;Il11fl/fl* lungs, which is expected given the increased amount of tissue analyzed. These information have been detailed in the respective figure legends, which we now also restate in the methods section accordingly, and we apologize for any confusion caused.

14) I appreciate the authors' response and their provision of additional data, as depicted in Figure R3.19. However, the uninjured lung appears to exhibit signs of pathological tissue remodeling, rather than appearing truly uninjured (as per image below). Furthermore, the HP levels are higher than those observed in uninjured mice in R3.14 or Supplementary 13c, which already exhibited pathological features (as mentioned previously).

Regarding the effect of X203, the Masson's trichrome staining indicates a comparable loss of alveoli in the BLM+IgG group compared to the BLM+X203 group. Additionally, the authors probed a specific region (highlighted in red) in the BLM+IgG group, which is evidently not representative of the entire lung (as evidenced by regions in green).

Overall, it is possible to appreciate a mild reduction in fibrosis at the histological level (one slides only), reflected by a non-significant trend in HP reduction (n=4), as also confirmed by the authors at line 511 page 18.

Last, authors included only 4 animals per condition, which is by far a very limited number in this type of study. Is sample size supported by power analysis?

Response: We acknowledge and agree with the reviewer's concerns that uninjured lungs of our inhouse *Sftpc-tdT* strain does not appear to look completely healthy at baseline and that this has already been accounted for and reflected in the slightly elevated ashcroft scores as compared to our other strains. The very subtle baseline changes may be due to the housing of our mice in the specific SPF conditions of our institution's vivarium. Importantly, we reemphasize that all in vivo experiments in the study were conducted with the most appropriate control animals. For this case, age matched *Sftpc-tdT* mice were all similarly injected with tamoxifen prior to BLM/Saline/antibody treatment and housed under the exact same conditions during the course of the study. Despite these, key alveolar epithelial and fibrosis phenotypes at baseline did not appear to be affected. Critically, these mice ultimately showed very robust alveolar responses to BLM-injury, displayed upregulation of our target IL11-protein in lineage traced cells (Fig 2k) and exhibited expected alveolar epithelial differentiation programs and increase in ECM proteins associated with severe BLM-induced lung injury (increased alveolar KRT8+ and ColI+ cells; Fig R3.7 below). Last but not least, this strain also responded to X203 treatment in an expected fashion (described below for our HPA

data). Hence, we were not overly concerned over the baseline phenotype of these mice and we now hope the reviewer understands.

Fig R3.7. Images of immunostaining for KRT8 and Collagen I in lungs from uninjured or BLM-injured Sftpc-tdT mice 12 days post-BLM-injury.

As for the choice of images in the same in vivo antibody study, it is our intention to focus on displaying images of severely injured regions between treatment groups to emphasize that fibrotic lesion sizes were profoundly reduced by X203 as compared to IgG. Furthermore, we have also provided unbiased HPA analysis and Ashcroft scoring, which assesses fibrosis across entire lung tissue, as additional independent readouts of fibrosis in the model. Nevertheless, we have replaced the enlarged images of BLM+IgG to a less fibrotic area to better mirror the corresponding tissue level quantitative HPA and Ashcroft data (Fig R3.8).

Fig R3.8. Replacement enlarged image of Masson's trichrome staining of lungs from Sftpc-tdT mice treated with IgG antibodies 12 days post-BLM-injury.

Additionally, we have compiled Masson's trichrome images of the other 3 mice in each antibody treatment group for the reviewer's consideration (Fig R3.9 ; imaged at similar specific locations of the left lung as above). These images consistently capture the profound reduction of fibrotic lesion number and lesion size across large regions of the lung tissue in X203-treated mice as compared to IgG-treated mice. We hope the reviewer can now better appreciate these effects.

Fig R3.9. Masson's trichrome staining of individual lungs from *Sftpc-tdT* mice treated with either X203 or IgG antibodies 12 days post-BLM-injury.

With regards to the effects of X203-treatment on HPA content in our *Sftpc-tdT* strain, the reviewer comments that he/she saw a non-significant trend in HPA reduction following X203 treatment in lines 511, which is inconsistent with the data presented and from our text description of these results. Our data in fact showed that HPA was reduced by ~59% in X203 as compared to IgG and that this was statistically significant by one-way ANOVA analysis ($P=.038$). Importantly, the effects of X203 shown are consistent and comparable to the effects seen in our previous studies of X203-treatment in wild-type C57BL/6J mice (same background as *Sftpc-tdT* mice) in the 21 day BLM-model; in our previous therapeutic studies using this antibody, we documented a ~53% reduction ($n=18$; $P=6.86e-4$) in HPA content in X203 as compared to IgG in (PMID: 31554736) and ~51% reduction ($n=10$; $P=.006$) in (PMID:32656894). As such, we are very confident of our current data on the reduction of fibrosis and associated epithelial phenotypes with X203-treatment.

As for the reviewer's comment on power analysis, we refer the reviewer to our response below.

15) Please refer to point 11 regarding the AT2 - AT1 trans-differentiation in injured *Sftpc-tdT*. Regarding the potential impact of X203 on specific subclasses of pathological fibroblasts, the provided images in R3.20 are quite intriguing, and I appreciate the author for including them. However, this secondary effect of X203 on fibroblasts should ideally translate into an even stronger anti-fibrotic effect of X203 on bleomycin-induced lung fibrosis. On the contrary X203 appears less effective, in terms of Masson's trichrome and HP levels, if compared with ATII-restricted *IL1ra1* KO model. It's worth noting that the ATII-restricted *IL1ra1* KO model should be considered pure model of ATII-restricted IL11 signaling and excluding by definition any effect of IL11 on fibroblasts. Could the authors kindly provide insight into this matter?

Response: We generally agree with the reviewer's comments here that our genetic model (100% cell specific IL11-signaling inhibition) appeared more effective than X203-treatment (likely less than 100% inhibition over the time course). Firstly, it is acceptable to expect certain degree of differences in fibrosis outcomes between the two models as they were designed to investigate different aspects of IL11 biology and were also performed on separate mouse strains: 1) *Sftpc-Il11ra1fl/fl* model was used as proof-of-concept for the direct role of IL11-signaling in AT2 cells for aberrant *Krt8* transitional cell differentiation and fibrosis; 2) whereas our therapeutic X203 study utilized *Sftpc-tdT* mice to test the potential of anti-IL11 antibodies for the improvement of alveolar epithelial regeneration after injury. Furthermore, given that we do not currently have additional data on fibroblast activities in both *Sftpc-Il11ra1fl/fl* and X203-treatment models, we unfortunately cannot provide further insights from those already discussed in our previous rebuttal.

12-15) In summary the set of histological and biochemical data produced for the X203 experiment are not convincing and needs the following major revisions:

- 12) For consistency with the data shown in figure 4, I kindly request the authors to provide data (including whole lung Masson's trichrome staining, Ashcroft score, and HP content) on X203 efficacy at 21 days post-bleomycin treatment. If this additional dataset confirms improved tissue regeneration at day 21, it will also assist in addressing points 14 and 15.

Response: The reviewer makes a repeat comment about our X203 in vivo data, which we have now addressed in point 12 above.

- 13-14) Damaged uninjured control lungs. Can author provide and explanation for that? Can author please provide additional images of uninjured control mice?

Response: As described in our response to point 14 above, the lung images for uninjured *Sftpc-III1ra1fl/fl* and -WT mice are representative of our cohort of saline-treated uninjured controls.

- 14) Improper image selection for X203 treatment supplementary figure 19c. Are author sure about and effect of X203 at the histological level?

Response: As described in our response to point 14 above, we reaffirm that the images for BLM+X203 treated mice are representative of the fibrosis phenotype observed and are also reflective of our tissue level HPA and Ashcroft analyses.

- 15) Different number of animals shown for HP and Ashcroft in figure supp. 19c (4 animals/condition) and IF analysis Figure 5C (6 animals/condition). Can the author provide an explanation for this sample size selection? Is sample size supported by power analysis?

Response: We acknowledge the reviewer's concerns over the sample sizes presented in Fig S19 and Fig 5C. We have indeed conducted *a priori* power calculations before performing these in vivo experiments. To elaborate, we performed power calculations on G.power software based on the large effect size ($d = 2.92$) of previous Ashcroft data from our very large therapeutic X203 study of $n = 18$ mice at day 21 post-BLM time point (Ashcroft scores: BLM+IgG = 5.55 ± 0.93 vs. BLM+X203 = 3.025 ± 0.79) (PMID: 31554736). Our analysis showed that a high statistical power of ~90% was achieved with just 4 mice / group ($\alpha = 0.05$, power $1 - \beta = 0.8$). Hence, we posit that our experiment with ≥ 4 mice / group here is adequately powered to detect meaningful differences in fibrosis outcomes and of associated alveolar phenotypes between our two antibody treatment groups.

16) I thank the authors for the reply and for providing new images as reported in Fig R3.21

17) I thank the authors for their response and for presenting new data as shown in Fig R3.22. However, it remains challenging to comprehend why a reduction in Collagen1 levels following X203 treatment (Figure 5j) is not mirrored by a corresponding decrease in HP levels, as observed in the *Sftpc-CreER; III1ra1fl/fl* mice. Could the authors kindly provide insight into this matter?

Response: We acknowledge that HPA levels were slightly but not statistically significantly elevated in X203-treated mice as compared to uninjured controls ($P=0.1938$) and that this may seem unexpected given the large effects on Collagen I expression. We reason that, since HPA is a universal component of fibrillar collagen of all types, the slightly elevated levels of HPA may represent the elevated expression of other types of collagen, and/or could be related to the therapeutic instead of prophylactic antibody-treatment approach taken. We also reiterate that we do not see complete normalization of HPA content to levels of uninjured controls in our previous studies of therapeutic X203-treatment in the BLM-model. Hence, our results here are expected and largely consistent with our previous findings.

Importantly, the expression of Collagen I was specifically investigated here to validate our new scRNA-seq findings of profibrotic *Coll1a1*-expressing *Krt8+* transitional cell states, and should also be appreciated in this key context.

18) I thank the authors for rephrasing.

19) In my humble opinion this paper provides only evidence regarding the role of IL11 signaling in mouse ATII cells, and therefore I kindly ask the authors to rephrase accordingly.

Response: We respectfully disagree with the reviewer's comment here as we have provided extensive human scRNA-seq evidence to support the case that *IL11* is specifically upregulated in *IIIIRA*-expressing disease specific epithelial and stromal cell types and strongly correlated with EMT-like signatures in disease specific aberrant epithelial cells in human PF. Our extensive in vitro experiments similarly demonstrated that the IL11-ERK axis directly triggers EMT-related changes in both primary human alveolar and distal lung epithelial cells.

Additional comments to the new version of the manuscript

1) Page 3 line 88: According to comments raised in point 14, please rephrase "These effects were similarly mirrored by anti-IL11 treatment" into "These effects were only partially mirrored by anti-IL11 treatment"

Response: We thank the reviewer for this comment but we do not agree with the reviewer's interpretation of our X203 in vivo fibrosis data as discussed above. We have instead rephrased the sentence to better reflect the key qualitative similarities in epithelial phenotypes between our anti-IL11 and genetic models. We hope that this now provides better clarity.

"We further show that therapeutic administration of anti-IL11 antibodies in the bleomycin model similarly prevents the accumulation of profibrotic *Krt8+* transitional cells and enhances regeneration of the injured lung epithelium."

REVIEWER COMMENTS

Reviewer #2 (Remarks to the Author):

Although the authors have addressed most concerns raised by the reviewer, some data added in the revised manuscript appear inconsistent. Therefore, the authors should clarify the following points before publication.

Major Points:

1. Although the authors analyzed the same cell populations representing IL-11+ expressing epithelial cells in the lung following BLM injection, the percentages of these cells show a striking difference between the results of Supplementary Figures 6 and 8. In Supplementary Figure 6b, the percentages of EGFP+ epithelial cells were approximately 0.1 to 0.2% in BLM-treated mice on Day 10. In contrast, in Supplementary Figure 8 and related Figures 2f to 2h, the percentages of EGFP-PE+ cells were about 1 to 3% in mice with the same treatment. This indicates a more than 10-fold difference between the two results. The authors should thoroughly explain these inconsistent results.

2. In Supplementary Figure 8 and related Figures 2f to 2h, please explain what Cldn4 Alexa Fluor 488 means. Does this indicate that the rabbit anti-Cldn4 antibody (Invitrogen, 36-4800) was visualized with donkey Alexa488-conjugated anti-rabbit antibody? If so, the Alexa488-conjugated antibody would react with PE-conjugated anti-GFP (Abcam, ab303588) antibodies raised in rabbits. The authors need to clarify this point, including the detailed method.

Minor Points:

1. In Lines 102-103 and Figure 1, given that cells other than epithelial cells, including stromal cells, also express Il11, the title in Figure 1 should be changed to a more appropriate one.
2. In Lines 224 and 226, Supplementary Fig. 6 should be corrected to Supplementary Fig. 7.

Reviewer #3 (Remarks to the Author):

General

The authors have satisfactorily addressed the majority of my concerns by providing either convincing explanation, previously published data, or new data supporting their findings.

However, I kindly ask the authors to comment these two additional points:

1) Main text Line 561-565 and Supplementary figure 22: I appreciate the authors' efforts in generating these data. However, it is regrettable that they currently lack a sufficient number of Sftpc-tdT animals. Including data with an N=2 is concerning and in my humble opinion it falls below the standards expected for a NatCom paper. Therefore, I suggest removing lines 561-565 and Supplementary Figure 22."

2) Rebuttal letter point. 13:

"To further clarify on HPA data in Supplementary Fig. 18, the entire right lung of Sftpc-CreER;Il11fl/fl mice was used for HPA content determination as compared to just the right caudal lobes for Sftpc-CreER;Il11ra1fl/fl mice. This explains the overall increase in HPA content for Sftpc-CreER;Il11fl/fl lungs, which is expected given the increased amount of tissue analyzed. This information has been detailed in the respective figure legends, which we now also restate in the methods section accordingly, and we apologize for any confusion caused"

The Y-axis legend in Fig. 4C, Supp. Fig. 14C, Supp. Fig. 18G, and Supp. Fig. 19E consistently reads ' $\mu\text{g}/\text{Right lung}$.' How did the authors calculate these values? The typical output of an HP quantification assay is a concentration, such as μg of HP per μl of homogenate, obtained by interpolating readings with a standard curve. The specific portion of the lung used is irrelevant, as the same amount of tissue should be used to ensure representativeness of the entire lung.

If different amounts of tissue were digested in different experiments using the same buffer volume (thereby altering the concentration), the data should be expressed as μg per mg of digested tissue. This would ensure comparability across experiments and eliminate bias due to varying starting material amounts. Could the authors please clarify this point?

Point by point rebuttal

Reviewer #2 (Remarks to the Author):

Although the authors have addressed most concerns raised by the reviewer, some data added in the revised manuscript appear inconsistent. Therefore, the authors should clarify the following points before publication.

Major Points:

1. Although the authors analyzed the same cell populations representing IL-11+ expressing epithelial cells in the lung following BLM injection, the percentages of these cells show a striking difference between the results of Supplementary Figures 6 and 8. In Supplementary Figure 6b, the percentages of EGFP+ epithelial cells were approximately 0.1 to 0.2% in BLM-treated mice on Day 10. In contrast, in Supplementary Figure 8 and related Figures 2f to 2h, the percentages of EGFP-PE+ cells were about 1 to 3% in mice with the same treatment. This indicates a more than 10-fold difference between the two results. The authors should thoroughly explain these inconsistent results.

Response: We wish to thank the reviewer for his/her very constructive assessment of our revised work.

We acknowledge and agree with the reviewer's comment that the proportion of endogenous EGFP expressing epithelial cells in Supplementary Fig. 6 differ slightly from anti-GFP staining in Supplementary Fig. 8 and that there may be various reasons as to why these data do not completely align. Firstly, these data are based on two completely different sets of experiments and conducted on separate cohorts of mice. We first provided a preliminary characterisation of the model by quantifying IL11-expressing lung cells based on endogenous EGFP levels in live lung cells. We mentioned that endogenous EGFP+ cells were challenging to delineate given the highly auto fluorescent nature of lung cells and that endogenous EGFP+ signals were largely predetermined based on auto-fluorescence levels of non-EGFP expressing IL11^{+/+} cells, which we believe are the most appropriate (genetic) controls for this experiment. Second, we provided further refinement of the model on a separate cohort of mice by quantifying anti-GFP and Cldn4 stained cells, which provided a much clearer delineation of IL11-EGFP expressing cells (i.e. more robust GFP signatures) in the epithelial compartment.

It is also worth mentioning that these two separate cohorts of IL11-EGFP mice were bred and housed independently in two separate animal facilities in our academic institution. Hence, there may also be subtle differences in IL11-expression between the cohorts in this regard. Nonetheless, and most importantly, our data from our various methodologies (including histology) consistently showed that BLM-injury results in robust upregulation of IL11 expression in lung alveolar epithelial cells. We hope that the reviewer now understands that the differences in IL11-EGFP+ data between experiments are not inconsistencies but instead are reflective of differences in methodologies used.

2. In Supplementary Figure 8 and related Figures 2f to 2h, please explain what Cldn4 Alexa Fluor 488 means. Does this indicate that the rabbit anti-Cldn4 antibody (Invitrogen, 36-4800) was visualized with donkey Alexa488-conjugated anti-rabbit antibody? If so, the Alexa488-conjugated antibody would react with PE-conjugated anti-GFP (Abcam, ab303588) antibodies raised in rabbits. The authors need to clarify this point, including the detailed method.

Response: We acknowledge the reviewer's concern. Firstly, AF488 secondary antibody used was to visualize Cldn4 staining and we were aware that both AF488 secondary and anti-GFP primary antibodies were raised in rabbit and there was no good alternative set up at hand to detect both proteins together. However, we did not mention in the previous revision that in order to minimize any potential cross reactivity of these two antibodies, we had simply employed sequential staining by first staining the lung cells with anti-Cldn4 primary followed by AF488 secondary, and only after these 2 antibodies were incorporated and the cells thoroughly washed, that anti-GFP-PE was then applied as a last staining step before analysis.

As an indication of the appropriateness and robustness of this staining approach, we did not observe profound overlap of AF488 and anti-GFP signals across various cell populations. For example, amongst the various Cldn4 subsets, we only detected GFP+ in Cldn4 high populations and not in Cldn4 low populations. We also share additional plots below (Fig R2.1) showing the lack of AF488 signal in GFP+ stromal (CD45-CD31-EpCAM-) compartment in injured IL11-EGFP mice, which is in contrast to the abundant AF488 signature in GFP+ EpCAM+ cells - which are indicative of GFP+ transitional cells that we describe in Fig 2f-h. We apologize for not mentioning these steps previously and have adjusted the methods accordingly, we hope that this clarifies the reviewer's concern.

Fig R2.1. Flow cytometry analysis showing the presence or absence of Alexa Fluor 488 signal in the GFP⁺ epithelial or stromal compartment of IL11-EGFP lung cells after BLM-injury.

Minor Points:

1. In Lines 102-103 and Figure 1, given that cells other than epithelial cells, including stromal cells, also express Il11, the title in Figure 1 should be changed to a more appropriate one.

2. In Lines 224 and 226, Supplementary Fig. 6 should be corrected to Supplementary Fig. 7.

Response:

Minor Point 1) We disagree with the reviewer to retitl Fig. 1 as the main figure only contains data of IL11-expression in alveolar epithelial cells.

Instead, we have included IF data on IL11-expressing fibroblasts in Supplementary Fig. 7 and have titled the figure appropriately. We hope that the reviewer agrees with this.

“Supplementary Fig. 7. IL11 is upregulated in alveolar epithelial cells, fibroblasts and CD45⁺ cells after bleomycin-induced lung injury in mice.”

Minor Point 2) We thank the reviewer for highlighting these typographical errors, which we have now corrected.

Reviewer #3 (Remarks to the Author):

General

The authors have satisfactorily addressed the majority of my concerns by providing either convincing explanation, previously published data, or new data supporting their findings.

However, I kindly ask the authors to comment these two additional points:

1) Main text Line 561-565 and Supplementary figure 22: I appreciate the authors' efforts in generating these data. However, it is regrettable that they currently lack a sufficient number of Sftpc-tdT animals. Including data with an N=2 is concerning and in my humble opinion it falls below the standards expected for a NatCom paper. Therefore, I suggest removing lines 561-565 and Supplementary Figure 22."

Response: We greatly thank the reviewer for his/her very constructive comments of our revised work. We acknowledge that we lack sufficient numbers for the experiment in Supplementary Fig. 22 and will therefore remove the data from the manuscript as suggested by the reviewer.

2) Rebuttal letter point. 13:

"To further clarify on HPA data in Supplementary Fig. 18, the entire right lung of Sftpc-CreER;I111fl/fl mice was used for HPA content determination as compared to just the right caudal lobes for Sftpc-CreER;I111ra1fl/fl mice. This explains the overall increase in HPA content for Sftpc-CreER;I111fl/fl lungs, which is expected given the increased amount of tissue analyzed. These information have been detailed in the respective figure legends, which we now also restate in the methods section accordingly, and we apologize for any confusion caused"

The Y-axis legend in Fig. 4C, Supp. Fig. 14C, Supp. Fig. 18G, and Supp. Fig. 19E consistently reads 'µg/Right lung.' How did the authors calculate these values? The typical output of an HP quantification assay is a concentration, such as µg of HP per µl of homogenate, obtained by interpolating readings with a standard curve. The specific portion of the lung used is irrelevant, as the same amount of tissue should be used to ensure representativeness of the entire lung.

If different amounts of tissue were digested in different experiments using the same buffer volume (thereby altering the concentration), the data should be expressed as μg per mg of digested tissue. This would ensure comparability across experiments and eliminate bias due to varying starting material amounts. Could the authors please clarify this point?

Response: The reviewer is correct to state that the typical output of HPA quantification is μg of HPA per μl of homogenate which can then be expressed as $\mu\text{g}/\text{mg}$ tissue digested after factoring in the respective dilutions. However, we have observed that the representation of HPA data in $\mu\text{g}/\text{mg}$ is inappropriate for the model as severe inflammation/oedema associated with BLM-injury profoundly increases tissue weights which may therefore further dilute HPA content per unit of tissue mass. In this case, we do not find that HPA concentrations ($\mu\text{g}/\text{mg}$) of BLM-treated lungs statistically increased from that of uninjured controls, which in itself is counterintuitive and ultimately does not reflect on the expected and overall fibrotic lung phenotypes observed when presented this way. To elaborate, we give examples below of lung weights and HPA concentration ($\mu\text{g}/\text{mg}$) of day 21 data of *Sftpc-CreER;Il11ralfl/fl* mice (Fig R3.1). This indicates a non-significant increase in HPA concentration between uninjured and injured controls, despite a very significant increase in wet tissue weights following BLM-injury - which is expected of the model. These results are in contrast to our presented data ($\mu\text{g}/\text{lung}$), which is derived simply by multiplying the $\mu\text{g}/\text{mg}$ data with the total wet tissue weight assessed to obtain the total HPA content in the sample.

To further clarify, and to eliminate any sampling bias, we hydrolysed each sample in equal volumes (1 mL) of 6N HCL and our data is expressed as μg of HPA in the entire 1 mL of tissue hydrolysate. As mentioned previously, we also quantified either the entire right lung or right caudal lobes accordingly and not parts thereof. We have adopted this convention across all of our previous lung fibrosis studies and following similar and previous reports of lung HPA content ($\mu\text{g}/\text{lung}$) in the bleomycin model by others in the field (PMID: 29967351, PMID: 29058717, PMID: 29200204 and very recent PMID:38987592). It is for these reasons that we have chosen to express our data as the total amount of HPA in the entire tissue homogenate as we believe it to be the most appropriate for the model and hope this clarifies the reviewers' concerns.

To provide better clarity, we have now relabeled the y-axis of Fig. 4c and Supplementary Fig. 14c and 19e to $\mu\text{g}/\text{R.lobe}$.

Fig R3.1. (a) Right caudal lobe wet tissue weight and (b) hydroxyproline concentration of right caudal lobes in uninjured and BLM-treated *Sftpc-CreER;Il11ra1^{fl/fl}* mice 21 days post injury. Statistics by one-way ANOVA.

REVIEWERS' COMMENTS

Reviewer #2 (Remarks to the Author):

The authors have responded to the comments by the reviewers, and now the manuscript will be suitable for publication in Nature Communications.

Reviewer #3 (Remarks to the Author):

I would like to express my gratitude to the authors for their response to the comments.

I appreciate the decision to remove Supplementary Figure 22 due to the insufficient number of animals in each experimental group.

However, regarding the HP quantification experiments, I disagree with the explanation provided:

"HPA data in $\mu\text{g}/\text{mg}$ is inappropriate for the model as severe inflammation/oedema associated with BLM-injury profoundly increases tissue weights, which may further dilute HPA content per unit of tissue mass."

The bleo-treated lungs are undergoing an inflammatory response, meaning that you are amplifying the quantification of HPA by multiplying it by a mass that represents inflammatory exudate and inflammatory cells rather than fibrotic tissue. This is conceptually incorrect and could have affected the measurements performed at day 12 post-BLM.

Conversely, this should not be the case for measurements performed on day 21, when inflammation is largely replaced by fibrosis.

In my opinion, the correct explanation is as follows: A fibrotic lung is more dense than a normal lung because alveoli are replaced by fibrotic tissue. Therefore, if you weigh the same amount of material for both tissues, you are digesting less material for the bleo-treated lung compared to the control, which could blunt the differences if you express the values as HPA $\mu\text{g}/\text{mg}$ of digested tissue. For this reason, and not the one reported by the authors, it is acceptable to express the values as HPA in the total lung (or any other type of piece you used), thus multiplying your HPA $\mu\text{g}/\text{mg}$ by the weight of the used portion. A lung that is more fibrotic is heavier because it is denser and therefore contains more ECM in the same volume of digested lung.

Lastly, it would have been advisable to use the same region of tissue for all measurements,

making the measurements comparable across different experiments. However, this issue is well balanced by the MT and Ashcroft score quantification, which are consistent with HPA quantifications. It remains debatable if the quantifications at day12 are appropriate in terms of absolute values, given the fact that inflammation could bias tissue weight.

Point by point rebuttal.

Reviewer #2 (Remarks to the Author):

The authors have responded to the comments by the reviewers, and now the manuscript will be suitable for publication in Nature Communications.

Response: We greatly appreciate the reviewer for his/her comments and input in improving the quality and clarity of our data and manuscript.

Reviewer #3 (Remarks to the Author):

I would like to express my gratitude to the authors for their response to the comments. I appreciate the decision to remove Supplementary Figure 22 due to the insufficient number of animals in each experimental group. However, regarding the HP quantification experiments, I disagree with the explanation provided:

"HPA data in $\mu\text{g}/\text{mg}$ is inappropriate for the model as severe inflammation/oedema associated with BLM-injury profoundly increases tissue weights, which may further dilute HPA content per unit of tissue mass."

The bleo-treated lungs are undergoing an inflammatory response, meaning that you are amplifying the quantification of HPA by multiplying it by a mass that represents inflammatory exudate and inflammatory cells rather than fibrotic tissue. This is conceptually incorrect and could have affected the measurements performed at day 12 post-BLM.

Conversely, this should not be the case for measurements performed on day 21, when inflammation is largely replaced by fibrosis.

In my opinion, the correct explanation is as follows: A fibrotic lung is more dense than a normal lung because alveoli are replaced by fibrotic tissue. Therefore, if you weigh the same amount of material for both tissues, you are digesting less material for the bleo-treated lung compared to the control, which could blunt the differences if you express the values as HPA $\mu\text{g}/\text{mg}$ of digested tissue. For this reason, and not the one reported by the authors, it is acceptable to express the values as HPA in the total lung (or any other type of piece you used), thus multiplying your HPA $\mu\text{g}/\text{mg}$ by the weight of the used portion. A lung that is more fibrotic is heavier because it is denser and therefore contains more ECM in the same volume of digested lung.

Lastly, it would have been advisable to use the same region of tissue for all measurements, making the measurements comparable across different experiments. However, this issue is well balanced by the MT and Ashcroft score quantification, which are consistent with HPA

quantifications. It remains debatable if the quantifications at day 12 are appropriate in terms of absolute values, given the fact that inflammation could bias tissue weight.

Response:

We genuinely thank the reviewer once again for his/her feedback in greatly improving our manuscript. We fully agree with the reviewer's opinion and excellent explanation for our day 21 data, and greatly appreciate the reviewer's insights into this matter. However, we believe that our day 12 data is also appropriate as we observed similar issues at day 12 as described for day 21 before. HPA concentrations ($\mu\text{g}/\text{mg}$) at day 12 were not significantly different between uninjured and injured controls, despite huge increases in lung tissue weights (Fig. R3.1). In this regard, we have similarly chosen to represent our day 12 data as absolute HPA content per lobe as these results best align with the overall fibrotic phenotype at this time point.

Lastly, we apologize for not standardizing our measurements of either R.lobes or R.lungs across our different strains, which could have made cross comparisons of HPA content much more straightforward.

Fig R3.1. (a) Hydroxyproline concentration and (b) right caudal lobe weights of *Sftpc-tdT* mice treated with BLM + IgG/X203 12 days 12 post-BLM challenge.